# DeformableTST: Transformer for Time Series Forecasting without Over-reliance on Patching

**Donghao Luo,  Xue Wang**

Department of Precision Instrument, Tsinghua University, Beijing 100084, China

`ldh21@mails.tsinghua.edu.cn, wangxue@mail.tsinghua.edu.cn`

## Abstract

With the proposal of patching technique in time series forecasting, Transformer-based models have achieved compelling performance and gained great interest from the time series community. But at the same time, we observe a new problem that the recent Transformer-based models are overly reliant on patching to achieve ideal performance, which limits their applicability to some forecasting tasks unsuitable for patching. In this paper, we intent to handle this emerging issue. Through diving into the relationship between patching and full attention (the core mechanism in Transformer-based models), we further find out the reason behind this issue is that full attention relies overly on the guidance of patching to focus on the important time points and learn non-trivial temporal representation. Based on this finding, we propose **DeformableTST** as an effective solution to this emerging issue. Specifically, we propose deformable attention, a sparse attention mechanism that can better focus on the important time points by itself, to get rid of the need of patching. And we also adopt a hierarchical structure to alleviate the efficiency issue caused by the removal of patching. Experimentally, our DeformableTST achieves the consistent state-of-the-art performance in a broader range of time series tasks, especially achieving promising performance in forecasting tasks unsuitable for patching, therefore successfully reducing the reliance on patching and broadening the applicability of Transformer-based models. Code is available at this repository: `https://github.com/luodhhh/DeformableTST`.

## 1  Introduction

Time series forecasting is widely used in real-world applications, such as transportation management [39, 5], economic planning [41, 30, 31], energy planning [42, 34] and weather forecasting [45]. Because of the immense practical value, time series forecasting has received great attention and has grown tremendously in recent years [44, 19, 33, 2, 28, 32, 25].

But looking back at the development of time series forecasting, **Transformer-based models, who have sparked the boom of time series forecasting** [55, 47, 57], **are constantly being challenged**. In particular, some recent studies [52, 18, 22] have questioned that attention mechanism is not suitable for modeling the temporal dependency in time series. As the early strike back of Transformer-based models, PatchTST [35] proposes that attention mechanism can work better in temporal modeling with the help of large size patching technique. Afterwards, equipped with the growing patch size and increasing input length, the advanced Transformer-based models [53, 58, 51, 6] gain great performance improvement and successfully win back the championship in time series forecasting.

However, with large size patching becoming a must-have technique for the following Transformer-based models, a new problem occurs: **patched-based Transformers have to work with a very long input length and a very large patch size to achieve ideal performance** [35, 53, 22]. But large size patching cannot be apply to all kinds of time series forecasting tasks. For example, some forecasting

(a) Not using patching (Input length: 336, Token number: 336)
Focusing on nearly all input time points

(b) Divided into 42 patches (Input length: 336, Token number: 42)
Focusing on nearly 42 important input time points

Figure 1: The Effective Receptive Field (ERF) of PatchTST. A brighter area means that these time points are focused by the model when extracting temporal representation. The results show that PatchTST highly relies on the guidance of patching to focus on the important time points. This phenomenon is also present in multiple advanced patch-based Transformer forecasters (Appendix E).

tasks are with limited input lengths [29, 30, 31], which are not sufficient to be divided into patches. In such condition, the advanced Transformer-based models suffer from severe performance degradation due to the lack of patching [58, 51], limiting their applicability to a wider range of forecasting tasks.

**To broaden the applicability of the Transformer-based model, we need to design an attention mechanism that is less reliant on patching (e.g., can work well with a small patch size or can work well even without patching)**. To this end, we first analyze exactly why attention must work with patching and why patching can help attention better model the temporal dependency in time series forecasting? We visualize the effective receptive fields (ERFs) of PatchTST [35] in Figure 1. And the ERFs can indicate which parts of the time points in input series are focused by the model when extracting temporal representations. A surprising finding is shown in Figure 1 (left). If without patching, nearly all time points in input series are equally focused by the model and the model performs worse (MSE 0.385), exposing the problem of distracted attention. This finding means that attention has not learned to distinguish the importance of each time point in input series, leading to trivial representation. Note that the time points in a time series are very redundant or even noisy [35, 56, 55, 7, 53], focusing on the trivial part of them will influence the predictions. Thus, an ideal time series forecaster should mainly focus on a small number of important time points which make contribution to better performance and reflect the property of time series. In Figure 1 (right), when using patching, the model focuses on some selected time points and achieve better performance (MSE 0.367), indicating that the model has successfully focused on the important time points. And in terms of why patching can guide the model to learn a non-trivial representation, we find that the pattern of ERF is also divided by patches, which means that patching can force the model to only focus on a small number of important time points based on the patch partition. As a conclusion of above discussion on Figure 1, **since full attention is unable to focus on the important time points by itself, it highly relies on the guidance of patching to focus on the important time points and learn non-trivial representation. This is the reason why full attention must work with patching to achieve ideal performance**.

Therefore, if we can find another way to help attention focus on the important time points, we can get rid of over-reliance on patching. Since full attention is hard to focus due to the redundancy in time series data [35, 56, 55, 7, 53], replacing it with sparse attention can be a natural idea. There are some previous prior-based sparse attentions in time series community [55, 47, 57]. But due to the diverse pattern in different time series, their priors are hard to match all kinds of inputs, resulting in their inferior performance. Different from them, we introduce a data-driven sparse attention called deformable attention under the inspiration of deformable operations [8, 60, 48]. It can sample a subset of important time points from the input series based on the learnable offsets and only calculate attention with these selected important time points. These learnable offsets are learned from each input sample, therefore being more flexible to the diverse property in different time series.

Based on the above motivations, we intend to broaden the applicability of Transformer-based models. To accomplish this goal, **we propose DeformableTST, a Transformer-based model that is less reliant on patching**. Technically, the patching process in our method is optional. We remove the patching process in most cases. Only when the input length is very long, we will use a small size patching for better efficiency. Since the removal of patching will cause severe memory usage in previous plain architecture, we adopt a hierarchical architecture to alleviate this efficiency issue. And we further introduce deformable attention, a data-driven sparse attention that can better focus on the important time points by itself, to achieve excellent performance without patching. Experimentally, DeformableTST achieves the consistent state-of-the-art performance in a wider range of time series tasks, especially in tasks unsuitable for patching, thus successfully reducing the reliance on patching and broadening the applicability of Transformer-based models. **Our contribution are as follows**:

- We dive into the relationship between patching and attention. We point out a new problem that recent advanced Transformer-based models are too reliant on patching. And we further

find out the reason behind this problem is that full attention relies overly on the guidance of patching to focus on important time points and learn non-trivial temporal representation.

- To get rid of the over-reliance on patching, we propose DeformableTST and achieve the consistent state-of-the-art performance in a wider range of time series forecasting tasks. Experimental results show that our deformable attention can better model the temporal dependency in time series without reliance on patching.

- We successfully broaden the applicability of Transformer-based models in time series tasks. Our DeformableTST can flexibly adapt to multiple input lengths and achieve excellent performance in tasks unsuitable for patching, which is a great improvement than previous Transformer-based models.

## 2 Related Work

### 2.1 Tranformers for Time Series Forecasting

Transformer-based models mainly use attention mechanism to model the temporal dependency in time series [55, 47, 57]. In 2020s, they achieve excellent performance in time series forecasting for the first time and bring great attention to time series forecasting tasks [26, 9, 56, 20, 21, 7]. But their validity is questioned by [52, 18] with the finding that a simple linear layer can outperform complicated attention mechanisms. It's until the proposal of patching that Transformer-based models win back the championship in time series forecasting [35]. Based on patching technique, Pathformer [6] adopts a multi-scale patches structure. Crossformer [53] and CARD [51] further propose to additionally apply attention on variate and feature dimensions rather than only on temporal dimension. Sageformer [54] combines the graph methods with patch-based Transformer forecasters. And GPT4TS [58] also transfers pre-trained large language models to time series with the help of patching. But the question of whether attention is suitable for modeling the temporal dependency in time series still remains. For example, although adopting a Transformer architecture, iTransformer [22] still suggests that linear layers are more appropriate for temporal modeling. Meanwhile, the proposal of patching also comes with a new question that advanced Transformer-based models are too reliant on patching. Therefore, further research about Transformer-based forecasters are still needed, especially on the question of how to better use attention in temporal modeling without over-reliance on patching.

### 2.2 Sparse Attention

Sparse attention used to be popular in time series forecasting. Early Transformer-based models usually adopt prior-based sparse attention mechanisms. Informer [55] adopt ProbSparse attention to model the temporal dependency. Autoformer and FEDformer [47, 57] further combine the signal processing technique with the attention mechanisms and select the top-k sparse representation in time domain or frequency domain respectively. But due to the diverse pattern in different time series, these priors are hard to match all kind of inputs, resulting in their inferior performance. As a comparison, data-driven sparse attention, also called deformable attention, is more flexible to diverse inputs. Similar idea has been explored in Computer Vision (CV). Inspired by deformable convolution [8, 59], deformable DERT [60] proposes multi-scale deformable attention for object detection tasks. And [48, 49] further improve it and make it suitable for general CV tasks. In this work, we propose a deformable attention for time series forecasting to break through the bottleneck faced by previous attention mechanism in modeling temporal dependency.

## 3 DeformableTST

Given an observed multivariate or univariate time series as input, time series forecasting aims to predict the length-$T$ future series based on the length-$I$ input series. In real-world scenarios, the input length $I$ varies from a wide range and is not always sufficient for patching technique, leading to the limited applicability of previous patch-based Transformer forecasters. To tackle this problem, we propose DeformableTST. And we introduce details of DeformableTST in following subsections.

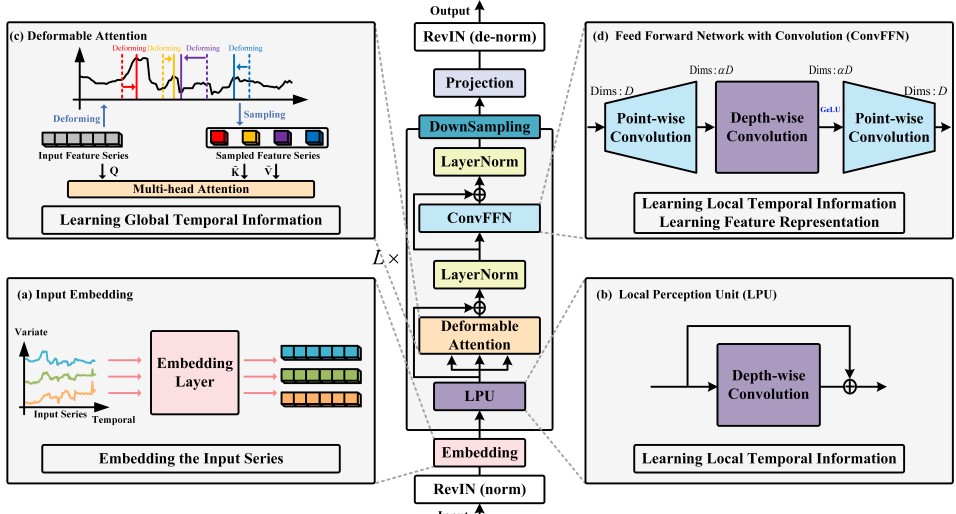

Figure 2: Structure overview of DeformableTST. (a) The input time series is embedded variate-independently. (b) The local perception unit (LPU) is used to learn the local temporal information. (c) The proposed deformable attention is adopted to learn the global temporal information. (d) The feed-forward network injected with a depth-wise convolution (ConvFFN) is used to learn the local temporal information and the new feature representation.

## 3.1 Structure Overview

As shown in Figure 2, our DeformableTST adopts the encoder-only architecture of Transformer [43], including the input embedding layer, hierarchical Transformer backbone and prediction head. And following the recent Transformer-based models, we adopt RevIN [15] to mitigate the distribution shift between the training and testing data.

**Input Embedding Layer** Denoted $\mathbf{X}_{in} \in \mathbb{R}^{M \times I}$ as the $M$ variates input time series of length $I$, it will be divided into $N_0$ non-overlapping patches and then embedded variate-independently into $D_0$-dimensional embeddings:

$$\mathbf{X}_0 = \text{Embedding}(\mathbf{X}_{in}) \tag{1}$$

$\mathbf{X}_0 \in \mathbb{R}^{M \times D_0 \times N_0}$ is the input embedding. It is worth noting that DeformableTST is less reliant on patching and thus the patching process is optional. We only adopt patching when the input length is very long for efficiency reasons. And we also adopt a much smaller patch size than recent Transformer-based models, making it more adaptable to diverse input lengths.

**Hierarchical Transformer Backbone** The backbone is stacked by $L$ Transformer blocks and utilizes a hierarchical structure. The forward process in the $i$-th block is simply formulated as follows:

$$\mathbf{X}_i^{local} = \text{LPU}(\mathbf{X}_{i-1}) \tag{2}$$

$$\mathbf{X}_i^{global} = \text{LayerNorm}\left(\mathbf{X}_i^{local} + \text{DeformableAttention}(\mathbf{X}_i^{local})\right) \tag{3}$$

$$\mathbf{X}_i = \text{LayerNorm}\left(\mathbf{X}_i^{global} + \text{ConvFFN}(\mathbf{X}_i^{global})\right) \tag{4}$$

$\mathbf{X}_i \in \mathbb{R}^{M \times D_i \times N_i}$ is the output feature series of the $i$-th block, $i \in \{1, ..., L\}$. And $D_i$ and $N_i$ are the sizes of its feature and temporal dimensions. DeformableAttention is the core component to better cpature the global temporal dependency, which will be introudced in Section3.2. LPU and ConvFFN are local enhancement modules (Figure 2 (b) and (d)). LPU is the local perception unit, a depth-wise convolution with residual connection [10]. And ConvFFN is a feed-forward network injected with a depth-wise convolution [50]. These two modules are adopted to improve the local temporal modeling ability. And a GELU activation [11] is adopted in ConvFFN to provide nonlinearity when learning the new feature representation. Meanwhile, to construct a hierarchical structure, a downsampling convolution layer [24] with kernel size 2 and stride 2 is adopted between two blocks, which will halve the series' temporal dimension and double the feature dimension.

**Prediction Head** We first flatten the final representation from the backbone $\mathbf{X}_L \in \mathbb{R}^{M \times D_L \times N_L}$ into $\mathbb{R}^{M \times (D_L \times N_L)}$. Then we obtain the prediction through a linear projection layer:

$$\widehat{\mathbf{Y}} = \text{Projection}(\mathbf{X}_L) \tag{5}$$

Where $\widehat{\mathbf{Y}} \in \mathbb{R}^{M \times T}$ is the prediction of length $T$ with $M$ variates.

## 3.2 Deformable Attention

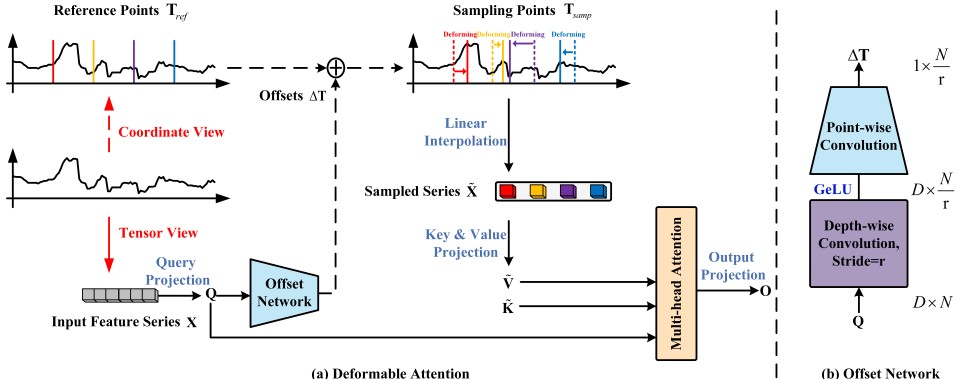

Figure 3: Deformable Attention. (a) The process of deformable attention from the tensor view and coordinate view. (b) The structure of the offset network, marked with the size of feature series.

Figure 3 introduces the detailed process of our deformable attention. In each attention module, it first samples a few important time points from the input feature series $\mathbf{X}$ based on the learnable offsets. Then the sampled important time points are fed to the key and value projections to get the sampled key and value tokens $\widetilde{\mathbf{K}}, \widetilde{\mathbf{V}}$. Meanwhile, the input feature series $\mathbf{X}$ is also projected into queries $\mathbf{Q}$. Finally, standard multi-head attention [43] is applied to $\mathbf{Q}, \widetilde{\mathbf{K}}, \widetilde{\mathbf{V}}$ to obtain the attention output $\mathbf{O}$.

**Sample the Important Time Points** As shown in Figure 3 (a), we sample the important time points based on a set of learnable coordinates called sampling points. Specifically, the sampling points are calculated by a set of uniformly sparse reference points and their learnable offsets.

Given a length-$N$ feature series $\mathbf{X} \in \mathbb{R}^{M \times D \times N}$, we first generate the sparse reference points $\mathbf{T}_{ref} \in \mathbb{R}^{M \times 1 \times N_{samp}}$ from a 1D uniform grid. The grid size $N_{samp} = N/r$ is downsampled from the input series length $N$ with a downsampling factor $r$ to provide sparsity. The reference points indicate the 1D coordinates of some time points uniformly distributed in the feature series $\mathbf{X}$ with interval $r$. These coordinate values are normalized to $[-1, +1]$, where $-1$ indicates the start of the series and $+1$ means the end of the series. And these reference points serve as the initial coordinates for the following deforming process.

Then we obtain the offsets for each reference point by offset sub-network (Figure 3 (b)). It contains two convolution layers. The first layer is a depth-wise convolution, which can take the local neighbors into consideration when generating the offsets [48]. It takes the query tokens $\mathbf{Q}$ as input, where $\mathbf{Q}$ is the linear projection of the feature series $\mathbf{X}$. After a nonlinear activation, the output from the first layer is passed into a point-wise convolution layer to generate the offsets $\Delta\mathbf{T} \in \mathbb{R}^{M \times 1 \times N_{samp}}$.

Adding up the reference points with the learnable offsets, we obtain $N_{samp}$ sampling points, which can serve as the final coordinates to sample the important time points from the feature series $\mathbf{X}$. In practice, we follow [48, 60] and calculate the values of these important time points by linear interpolation $\phi(\cdot; \cdot)$ to make this sampling process differentiable. The overall process is as follows:

$$\Delta\mathbf{T} = \text{Offset-Network}(\mathbf{Q}) \tag{6}$$

$$\mathbf{T}_{samp} = \mathbf{T}_{ref} + \Delta\mathbf{T} \tag{7}$$

$$\widetilde{\mathbf{X}} = \phi(\mathbf{X}; \mathbf{T}_{samp}) \tag{8}$$

where $\tilde{\mathbf{X}} \in \mathbb{R}^{M \times D \times N_{samp}}$ is the sampled feature series consisting of the important time points. And the implementation of linear interpolation $\phi(\cdot ; \cdot)$ is in Appendix I.1. And we clip $\mathbf{T}_{samp}$ by $-1$ and $+1$ to avoid sampling outside the feature series.

**Calculate Attention Output**    In above sampling process, we have got the query tokens $\mathbf{Q}$. After the sampling process, we can get the sampled key and value tokens $\tilde{\mathbf{K}}$, $\tilde{\mathbf{V}}$ after two linear projections of the sampled feature series $\tilde{\mathbf{X}}$. Then we calculate the multi-head self-attention with $H$ heads as:

$$\mathbf{O}^{(h)} = \mathrm{Softmax}\left(\mathbf{Q}^{(h)} \tilde{\mathbf{K}}^{(h)\top} / \sqrt{d} + \mathbf{B}\right) \tilde{\mathbf{V}}^{(h)}, h = 1, \dots, H \tag{9}$$

$$\mathbf{O} = \mathrm{OutputProjection}(\mathrm{Concat}\left(\mathbf{O}^{(1)}, \dots, \mathbf{O}^{(H)}\right)) \tag{10}$$

where $d = D/H$ is the dimension of each head. The upper index $^{(h)}$ denotes the $h$-th attention head. After concatenating the output embedding from each attention head $\mathbf{O}^{(h)}$ together, we obtain the output of the DeformableAttention module $\mathbf{O} \in \mathbb{R}^{M \times D \times N}$ through a linear projection. $\mathbf{B}$ is the deformable relative position bias to provide the positional information into the attention map and its implementation is introduced in Appendix I.2.

To conclude, this subsection introduces the detailed process of DeformableAttention (Eq.(3)). And for the $i$-th block, $\mathbf{X}$ in this subsection corresponds to $\mathbf{X}_i^{local}$ in Eq.(3) and $\mathbf{O}$ corresponds to $\mathbf{X}_i^{global}$.

## 4   Experiments

We thoroughly evaluate our DeformableTST on a wide range of time series forecasting tasks, including long-term forecasting tasks with various input lengths, as well as multivariate and univariate short-term forecasting tasks that are unsuitable for patching, to verify the performance and applicability of our DeformableTST.

**Baselines**    We extensively include the latest and advanced models in time series community as strong baselines, including patch-based Transformer models: Pathformer [6], CARD [51], GPT4TS [58], PatchTST [35]; non patch-based Transformer models: iTransformer [22], FEDformer [57], Autoformer [47]; other non Transformer-based models: RLinear [18], TiDE [9], TimesNet [46], DLinear [52] and SCINet [20]. We also include the state-of-the-art models in each specific task as additional baselines for a comprehensive comparison.

**Main Result**    As shown in Figure 4, **our DeformableTST achieves consistent state-of-the-art performance in a broader range of time series tasks**. In details, DeformableTST can flexibly adapt to multiple input lengths and especially achieve excellent performance in tasks unsuitable for patching, which is a great improvement than previous Transformer-based models, **proving that our DeformableTST can successfully reduce the reliance on patching and broaden the applicability of Transformer-based models**. Experiment details and result discussions of each task are provided in following subsections. In each table, the best results are in **bold** and the second best are underlined.

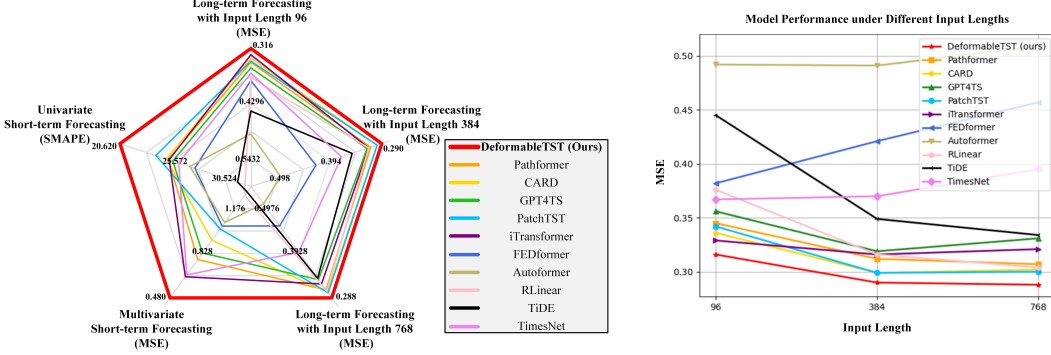

Figure 4: Model performance comparison (left) and performance under different input lengths (right).

## 4.1 Long-term Forecasting

**Setups**    We conduct long-term forecasting experiments on 8 popular real-world benchmarks, including Weather [45], Traffic [39], ECL [42], Solar-Energy [34] and 4 ETT datasets [55]. **In this paper, we refine the evaluation approach for a comprehensive comparision of the models.** Different from previous settings that use a fixed short input length (e.g., 96) [55, 47, 22]. We fix three different input lengths $\{96, 384, 768\}$ and calculate the averaged results to adequately reflect model's adaptability to multiple input lengths. These input lengths covers a variety of real-world application scenarios, i.e., shorter than prediction lengths, within the prediction lengths' interval and longer than prediction lengths. Following the previous settings, we set prediction lengths as $\{96, 192, 336, 720\}$ and calculate the MSE and MAE of multivariate time series forecasting as metrics.

**Results**    Table 1 shows the excellent performance of DeformableTST in long-term forecasting. Concretely, DeformableTST gains the best performance in most cases, surpassing extensive state-of-the-art Transformer-based models.  As shown in Figure 4 (right), DeformableTST achieves the consistent state-of-the-art performance in all input lengths and gains continuous performance improvement with the increasing input length, validating its adaptability to multiple input lengths and its effectiveness in extracting useful information from longer history. For comparison, the non patch-based Transformer baselines suffer from performance degradation with increasing input length due to the distracted attention on the prolonging input. And the patch-based Transformer baselines can not work well with a short input length (e.g., 96) because leveraging patching on the short time series leads to very few tokens, limiting attention's ability in long-term modeling.

Table 1: Multivariate long-term forecasting results.  A lower MSE or MAE indicates a better performance. Results are averaged from three input lengths $I \in \{96, 384, 768\}$ and four prediction lengths $T \in \{96, 192, 336, 720\}$. See Table 8, 9, 10 for full results with more baselines.

| Models | DeformableTST (Ours) | | Pathformer [6] | | CARD [51] | | GPT4TS [58] | | PatchTST [35] | | iTransformer [22] | | FEDformer [57] | | Autoformer [47] | | RLinear [18] | | TiDE [9] | | TimesNet [46] | |
|---|---|---|---|---|---|---|---|---|---|---|---|---|---|---|---|---|---|---|---|---|---|---|
| Metric | MSE | MAE | MSE | MAE | MSE | MAE | MSE | MAE | MSE | MAE | MSE | MAE | MSE | MAE | MSE | MAE | MSE | MAE | MSE | MAE | MSE | MAE |
| ETTh1 | **0.413** | **0.430** | 0.439 | 0.446 | 0.430 | 0.438 | 0.479 | 0.459 | 0.438 | 0.444 | 0.461 | 0.463 | 0.505 | 0.495 | 0.483 | 0.488 | 0.429 | 0.434 | 0.505 | 0.492 | 0.524 | 0.485 |
| ETTh2 | **0.336** | **0.381** | 0.361 | 0.401 | 0.355 | 0.391 | 0.399 | 0.425 | 0.356 | 0.394 | 0.390 | 0.417 | 0.441 | 0.466 | 0.437 | 0.466 | 0.358 | 0.396 | 0.499 | 0.505 | 0.436 | 0.450 |
| ETTm1 | **0.358** | **0.386** | 0.375 | 0.394 | 0.368 | **0.386** | 0.373 | 0.393 | 0.365 | 0.391 | 0.386 | 0.405 | 0.456 | 0.462 | 0.576 | 0.523 | 0.379 | 0.390 | 0.383 | 0.397 | 0.412 | 0.414 |
| ETTm2 | 0.267 | 0.321 | 0.277 | 0.329 | 0.268 | 0.321 | 0.286 | 0.331 | 0.273 | 0.327 | 0.281 | 0.335 | 0.335 | 0.376 | 0.347 | 0.389 | **0.266** | **0.320** | 0.310 | 0.373 | 0.311 | 0.351 |
| Weather | **0.233** | **0.266** | 0.239 | 0.269 | 0.247 | 0.278 | 0.249 | 0.280 | 0.236 | 0.269 | 0.244 | 0.276 | 0.317 | 0.360 | 0.327 | 0.366 | 0.253 | 0.283 | 0.254 | 0.295 | 0.271 | 0.299 |
| Solar-Energy | **0.199** | **0.255** | 0.248 | 0.300 | 0.228 | 0.282 | 0.259 | 0.318 | 0.234 | 0.303 | 0.214 | 0.266 | 0.394 | 0.460 | 0.866 | 0.703 | 0.279 | 0.320 | 0.277 | 0.351 | 0.261 | 0.319 |
| ECL | **0.169** | **0.267** | 0.176 | 0.272 | 0.174 | 0.269 | 0.182 | 0.271 | 0.177 | 0.271 | 0.175 | 0.271 | 0.227 | 0.342 | 0.218 | 0.321 | 0.189 | 0.280 | 0.201 | 0.300 | 0.213 | 0.315 |
| Traffic | **0.410** | **0.280** | 0.454 | 0.312 | 0.426 | 0.291 | 0.453 | 0.309 | 0.430 | 0.290 | 0.424 | 0.306 | 0.685 | 0.423 | 0.725 | 0.433 | 0.502 | 0.330 | 0.579 | 0.401 | 0.593 | 0.318 |

## 4.2 Short-term Forecasting

Time series community are currently focusing on long-term forecasting tasks, where the input length and prediction length are adequate for patching technique. However, the short-term forecasting tasks, where the input length and prediction length are limited, are also of extensive practical value in real-world appilications. The study on short-term forecasting tasks has been stagnant in recent years and the advanced patching technique proposed in long-term forecasting is hard to apply to short-term forecasting due to the limited input length. To validate the applicability of our DeformableTST in short-term forecasting, we extensively conduct experiments in following two kinds of tasks.

**Setups of Multivariate Short-term Forecasting**    We conduct multivariate short-term forecasting experiments on 8 popular real-world benchmarks, including Exchange [17], ILI [3], 2 ETTh [55] and 4 PEMS datasets [5]. We set prediction lengths as $\{6, 12, 18\}$ and set the input length to be 2 times of the prediction length, which precisely meets the definition of limited input lengths in short-term forecasting. We calculate the MSE and MAE of multivariate time series forecasting as metrics.

**Setups of Univariate Short-term Forecasting**    The study on univariate short-term forecasting tasks used to be popular in the early time series community [29, 31, 37] but has been stagnant in recent years. In this paper, we bring back this classic tasks and conduct experiments on following datasets: M1 [29], M3 [30], M4 [31], Tourism [1], NN5 [41], Hospital [12] and KDD Cup [13]. Following the classic settings [29, 37], we calculate the SMAPE as metric. Specially for M4 datasets, we follow the

rules of M4 competition [31] and use MASE and OWA as additional metrics. The prediction lengths are from 2 to 48 and the input length is 2 times of the prediction length.

We emphasize the difference between multivariate and univariate tasks as follows. In multivariate tasks, all input samples are obtained by sliding window from the same multivariate long series. Therefore, there is a high degree of similarity among the input samples in multivariate tasks. In contrast, the input samples in univariate tasks are collected from many different univariate time series sources. As a result, the input samples in univariate tasks may differ from each other and have quite different temporal property, making the univariate short-term forecasting tasks much more difficult.

**Results** As shown in Table 2 and 3. DeformableTST performs excellently in short-term forecasting. Compared with the second best model, it achieves averaged 14.1% SMAPE promotion in univariate tasks and averaged 25.6% MSE promotion in multivariate tasks. As a comparison, the limited input length in short-term forecasting poses a dilemma for patch-based Transformer forecasters. Using patching will lead to very few tokens, making it unable to fully utilize the long-term modeling ability in attention. Not using patching will lead to the distracted attention on all input time points, making it hard to extract non-trivial temporal information. As a result, previous patch-based Transformer forecasters fail in many cases of short-term forecasting tasks. And in multivariate short-term forecasting, the variate correlation plays an important role to the final results for the temporal information is limited due to the limited input length. Therefore, CARD and iTransformer, which can learn the variate correlation, achieve ideal performance in PEMS datasets whose variate number is very large. As a variate-independent method, our DeformableTST still competes favorably with these cross-variate methods, further demonstrating its excellent temporal modeling ability to extract useful information even from the limited inputs. It will be our future work to study how to capture the multivariate correlation in our model, which will further improve the performance. Meanwhile, the great diversity in univariate samples makes it more difficult to learn temporal representation in univariate short-term forecasting. Therefore, the linear baselines and iTransformer, which adopt linear layers on the temporal dimension, suffer from inferior performance due to the insufficient representation capability in linear layers. By contrast, thanks to the better representation capability in attention mechanism and the better focusing capacity in the proposed deformable attention, our DeformableTST is particularly good at short-term forecasting tasks, which is a great improvement than previous Transformer-based models, therefore successfully broadening the applicability of Transformer-based models.

Table 2: Multivariate short-term forecasting results. A lower MSE or MAE indicates a better performance. Results are averaged from three prediction lengths $T \in \{6, 12, 18\}$. And the PEMS results are further averaged by four subsets. Full results and more baselines are listed in Table 11.

| Models | DeformableTST (Ours) | | Pathformer [6] | | CARD [51] | | GPT4TS [58] | | PatchTST [35] | | iTransformer [22] | | FEDformer [57] | | Autoformer [47] | | RLinear [18] | | TiDE [9] | | TimesNet [46] | |
|---|---|---|---|---|---|---|---|---|---|---|---|---|---|---|---|---|---|---|---|---|---|
| Metric | MSE | MAE | MSE | MAE | MSE | MAE | MSE | MAE | MSE | MAE | MSE | MAE | MSE | MAE | MSE | MAE | MSE | MAE | MSE | MAE | MSE | MAE |
| ETTh1 | **0.373** | **0.381** | 0.456 | 0.424 | 0.490 | 0.433 | 0.475 | 0.427 | 0.551 | 0.467 | 0.435 | 0.412 | 0.468 | 0.445 | 0.494 | 0.459 | 0.743 | 0.543 | 0.699 | 0.528 | 0.440 | 0.419 |
| ETTh2 | **0.142** | **0.239** | 0.159 | 0.257 | 0.163 | 0.262 | 0.164 | 0.263 | 0.172 | 0.269 | 0.164 | 0.262 | 0.176 | 0.279 | 0.174 | 0.278 | 0.211 | 0.303 | 0.192 | 0.289 | 0.171 | 0.266 |
| ILI | **1.767** | **0.798** | 3.130 | 1.160 | 3.872 | 1.371 | 3.365 | 1.227 | 4.217 | 1.435 | 2.510 | 1.009 | 4.407 | 1.509 | 4.500 | 1.523 | 5.262 | 1.611 | 5.444 | 1.648 | 2.544 | 0.930 |
| Exchange | **0.013** | **0.070** | 0.015 | 0.077 | 0.016 | 0.083 | 0.015 | 0.077 | 0.016 | 0.085 | 0.014 | 0.073 | 0.028 | 0.119 | 0.026 | 0.117 | 0.022 | 0.100 | 0.019 | 0.093 | 0.016 | 0.081 |
| PEMS (Avg) | 0.104 | **0.208** | 0.137 | 0.235 | 0.108 | 0.215 | 0.133 | 0.239 | 0.140 | 0.254 | **0.102** | **0.208** | 0.122 | 0.239 | 0.147 | 0.268 | 0.165 | 0.272 | 0.188 | 0.291 | 0.130 | 0.237 |

Table 3: Univariate short-term forecasting results. Lower metrics indicate better performance. *Weight Avg* and *Avg* means the results are (weighted) averaged by subdatasets. We only report the SMAPE as metric here. Full results with more metrics for M4 are provided in Table 12 and 13. *. in the Transformers indicates the name of *former.

| Models | DeformableTST (Ours) | Path. [6] | CARD [51] | GPT4TS [58] | Cross. [53] | PatchTST [35] | iTransformer [22] | FED. [57] | Auto. [47] | RLinear [18] | TiDE [9] | TimesNet [46] | DLinear [52] | SCINet [20] | N-HiTS [4] | N-BEATS [37] |
|---|---|---|---|---|---|---|---|---|---|---|---|---|---|---|---|---|
| M1 (Avg) | **15.250** | 18.684 | 18.001 | 19.771 | 20.055 | 18.662 | 21.626 | 20.056 | 21.066 | 27.709 | 27.119 | 19.126 | 24.422 | 26.910 | 25.466 | 19.810 |
| M3 (Avg) | **11.747** | 17.747 | 16.381 | 19.082 | 14.234 | 14.721 | 17.035 | 18.988 | 17.681 | 34.291 | 20.876 | 18.770 | 20.588 | 24.226 | 20.869 | 17.823 |
| M4 (Weight Avg) | **11.688** | 12.001 | 11.815 | 12.008 | 13.475 | 11.952 | 11.878 | 12.840 | 12.909 | 13.398 | 13.711 | 11.829 | 13.639 | 12.699 | 11.927 | 11.851 |
| Tourism (Avg) | **21.502** | 31.345 | 23.349 | 35.018 | 23.618 | 23.442 | 24.429 | 37.295 | 35.793 | 39.756 | 44.504 | 33.279 | 37.363 | 37.739 | 35.934 | 42.832 |
| NN5 | **14.372** | 18.186 | 22.491 | 16.012 | 17.672 | 17.717 | 20.449 | 17.072 | 19.650 | 22.868 | 23.424 | 22.355 | 23.781 | 26.486 | 16.645 | 23.282 |
| Hospital | **19.088** | 21.551 | 21.392 | 22.587 | 20.907 | 22.123 | 20.785 | 28.646 | 25.749 | 26.184 | 29.103 | 21.182 | 23.015 | 28.807 | 25.309 | 23.594 |
| KDD Cup | **50.694** | 56.740 | 61.653 | 54.713 | 58.472 | 59.449 | 60.615 | 59.013 | 57.617 | 62.922 | 63.447 | 56.618 | 59.523 | 63.358 | 54.855 | 62.305 |

# 5 Model Analysis

## 5.1 Ablation Study

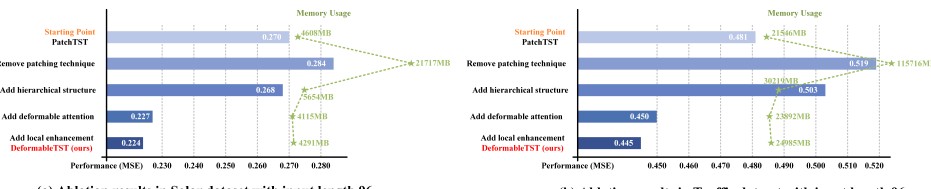

(a) Ablation results in Solar dataset with input length 96    (b) Ablation results in Traffic dataset with input length 96

Figure 5: Ablation study in long-term forecasting tasks. From the top to the bottom, each row means one design that we add on PatchTST to modify it into our DeformableTST. We report the averaged MSE of four prediction lengths. The memory usage is recorded under input-96-predict-96 setting with the same batch size. A lower MSE (a shorter blue bar) and a smaller memory usage (a green star closer to the vertical axis) means better performance and efficiency. More results are in Appendix F.1.

In order to validate the effectiveness of our designs, we start from a PatchTST [35] and gradually update it into our DeformableTST by adding our designs step by step. And we provide the trajectory going from a PatchTST to a DeformableTST in Figure 5. To get rid of the over-reliance on patching, we remove the patching design in PatchTST. After this step, PatchTST suffers from degradation in both performance and efficiency. To address these issues, more designs are adopted in DeformableTST.

First, the removal of patching leads to a larger number of tokens processed in the attention computation, resulting in heavier memory usage. Responding to the issue, we adopt a hierarchical structure to gradually reduce the number of tokens, therefore alleviating the efficiency problem. Secondly, full attention is hard to focus on the important time points after removing the patching design, leading to trivial temporal representation and performance degradation. To help attention better focus without patching, we propose deformable attention as a complementary. Thanks to the better focusing ability in deformable attention, we observe great performance improvement after adding this design. Meanwhile, we also adopt some local enhancement modules in our design since locality is also important in time series [20]. And this step also brings performance improvement. After equipped with all our designs, DeformableTST shows great performance and efficiency superiority than the baseline PatchTST, which proves the necessity and effectiveness of our designs.

## 5.2 Compared Deformable Attention with Prior-based Sparse Attention

In Section 1, we propose that sparse attention can help attention to better foucs without patching and further argue that data-drivien sparse attention is more appropriate than prior-based ones. To validate the necessity and effectiveness of using data-drivien sparse attention, we compare our deformable attention with some prior-based sparse attentions in time series community, that is ProbSparse Attention in Informer [55], AutoCorrelation in Autoformer [47] and FourierAttention in FEDformer [57]. We also include the local window attention in Swin Transformer [23] and the vanilla full attention [43] adopted in most baselines [22, 53, 35] for a comprehensive comparison.

As shown in Table 4, our deformable attention surpasses other prior-based competitors in all benchmarks. This is because the priors are hard to match all kind of inputs due to the diverse pattern in different time series, resulting in the inferior performance of prior-based sparse attentions. Different from them, our deformable attention is a data-driven sparse attention that can learn from the input time series, therefore is more flexible to the diverse property in different time series.

Our deformable attention also surpasses the full attention by a large margin for it can better focus on the important time points to learn non-trivial temporal representation, while the full attention suffers from the distracted attention and trivial temporal representation due to the lack of patching. Although window attention can help attention avoid being distracted in a global range by limiting the attention computation into local windows, its performance still decreases due to the lack of long-term modeling ability, which is an important ability a time series forecaster should have.

Table 4: Comparison of our Deformable Attention with other prior-based Sparse Attentions. We replace our Deformable Attention with other prior-based Sparse Attentions for comparison. We conduct the experiment in long-term forecasting tasks with input length 96 and list the averaged MSE/MAE of four different prediction lengths. More results are in Appendix F.2.

| Dataset | ETTh1 | | ETTh2 | | ETTm1 | | ETTm2 | | Weather | | Solar | | ECL | | Traffic | |
|---|---|---|---|---|---|---|---|---|---|---|---|---|---|---|---|---|
| Metric | MSE | MAE | MSE | MAE | MSE | MAE | MSE | MAE | MSE | MAE | MSE | MAE | MSE | MAE | MSE | MAE |
| **Deformable Attention** | **0.425** | **0.428** | **0.346** | **0.382** | **0.373** | **0.395** | **0.283** | **0.327** | **0.251** | **0.276** | **0.224** | **0.256** | **0.183** | **0.278** | **0.445** | **0.285** |
| ProbSparse Attention | 0.431 | 0.434 | 0.379 | 0.407 | 0.390 | 0.404 | 0.295 | 0.338 | 0.267 | 0.287 | 0.270 | 0.291 | 0.201 | 0.293 | 0.483 | 0.327 |
| AutoCorrelation | 0.449 | 0.443 | 0.375 | 0.401 | 0.393 | 0.405 | 0.292 | 0.334 | 0.263 | 0.286 | 0.275 | 0.298 | 0.215 | 0.302 | 0.561 | 0.373 |
| FourierAttention | 0.453 | 0.443 | 0.374 | 0.401 | 0.384 | 0.402 | 0.290 | 0.334 | 0.266 | 0.287 | 0.265 | 0.285 | 0.208 | 0.299 | 0.493 | 0.333 |
| Full Attention | 0.447 | 0.444 | 0.373 | 0.400 | 0.385 | 0.403 | 0.293 | 0.337 | 0.260 | 0.286 | 0.247 | 0.273 | 0.201 | 0.292 | 0.469 | 0.316 |
| Window Attention | 0.441 | 0.437 | 0.370 | 0.398 | 0.380 | 0.397 | 0.291 | 0.334 | 0.263 | 0.285 | 0.255 | 0.280 | 0.204 | 0.295 | 0.488 | 0.327 |

# 6 Conclusion and Future Work

In this paper, we expose an emerging issue faced by advanced Transformer-based models that they have limited applicability in time series forecasting tasks due to their over-reliance on patching. And we further find out the reason behind this problem is that full attention relies overly on the guidance of patching to focus on the important time points and learn non-trivial temporal representation. To tackle this problem, we propose DeformableTST as an effective solution, which equips with deformable attention that can better focus on the important time points by itself to get rid of the over-reliance on patching. Experimentally, DeformableTST achieves the consistent state-of-the-art performance in a broader range of time series forecasting tasks, especially achieving promising performance in tasks unsuitable for patching, therefore successfully reducing the reliance on patching and broadening the applicability of Transformer-based models. And we hope our findings can prompt people to rethink the relationship between Transformer-based models and patching technique, thereby designing more powerful Transformer-based forecasters with a wider range of applicability.

# Acknowledgment

This work was supported by the Hebei Innovation Plan (20540301D).

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

# A  Datasets

Table 5: Detailed descriptions of multivariate datasets. The Dataset Size denotes the total number of time points in (Train, Validation, Test) split respectively.

| Tasks | Dataset | Variates | Prediction Length | Dataset Size | Frequency | Information |
|---|---|---|---|---|---|---|
| Long-term Forecasting | ETTh1 | 7 | {96, 192, 336, 720} | (8545, 2881, 2881) | Hourly | Electricity |
| | ETTh2 | 7 | {96, 192, 336, 720} | (8545, 2881, 2881) | Hourly | Electricity |
| | ETTm1 | 7 | {96, 192, 336, 720} | (34465, 11521, 11521) | 15min | Electricity |
| | ETTm2 | 7 | {96, 192, 336, 720} | (34465, 11521, 11521) | 15min | Electricity |
| | Weather | 21 | {96, 192, 336, 720} | (36792, 5271, 10540) | 10min | Weather |
| | Solar-Energy | 137 | {96, 192, 336, 720} | (36601, 5161, 10417) | 10min | Energy |
| | ECL | 321 | {96, 192, 336, 720} | (18317, 2633, 5261) | Hourly | Electricity |
| | Traffic | 862 | {96, 192, 336, 720} | (12185, 1757, 3509) | Hourly | Transportation |
| Short-term Forecasting | PEMS03 | 358 | {6, 12, 18} | (15617, 5135, 5135) | 5min | Transportation |
| | PEMS04 | 307 | {6, 12, 18} | (10172, 3375, 3375) | 5min | Transportation |
| | PEMS07 | 883 | {6, 12, 18} | (16911, 5622, 5622) | 5min | Transportation |
| | PEMS08 | 170 | {6, 12, 18} | (10690, 3548, 3548) | 5min | Transportation |
| | Exchange | 8 | {6, 12, 18} | (5120, 665, 1422) | Daily | Illness |
| | ILI | 7 | {6, 12, 18} | (676, 96, 194) | Weekly | Economy |
| | ETTh1 | 7 | {6, 12, 18} | (8545, 2881, 2881) | Hourly | Electricity |
| | ETTh2 | 7 | {6, 12, 18} | (8545, 2881, 2881) | Hourly | Electricity |

Table 6: Datasets and mapping details of univariate short-term forecasting datasets.

| Dataset | Sample Numbers (train set,test set) | Variate Numbers | Prediction Length |
|---|---|---|---|
| M1 Yearly | (181, 181) | 1 | 2 |
| M1 Quarterly | (203, 203) | 1 | 3 |
| M1 Monthly | (617 , 617 ) | 1 | 8 |
| M3 Yearly | (645, 645) | 1 | 3 |
| M3 Quarterly | (756, 756) | 1 | 4 |
| M3 Monthly | (1428, 1428) | 1 | 10 |
| M3 Other | (174 , 174 ) | 1 | 10 |
| M4 Yearly | (23000, 23000) | 1 | 6 |
| M4 Quarterly | (24000, 24000) | 1 | 8 |
| M4 Monthly | (48000, 48000) | 1 | 18 |
| M4 Weekly | (359, 359) | 1 | 13 |
| M4 Daily | (4227, 4227) | 1 | 14 |
| M4 Hourly | (414, 414) | 1 | 48 |
| Tourism Quarterly | (427, 427) | 1 | 5 |
| Tourism Monthly | (366 , 366 ) | 1 | 15 |
| NN5 Weekly | (111, 111 ) | 1 | 15 |
| Hospital Monthly | (767 , 767 ) | 1 | 10 |
| KDD Cup Hourly | (270 , 270) | 1 | 48 |

## A.1  Multivariate Long-term and Short-term Forecasting Datasets

We evaluate the multivariate long-term forecasting performance on 8 popular real-world datasets, including Weather, Traffic, ECL, Solar-energy and 4 ETT datasets (ETTh1, ETTh2, ETTm1, ETTm2). And for multivariate short-term forecasting tasks, we choose Exchange, ILI, 2 ETTh datasets and 4 PEMS datasets for benchmarking. These datasets have been extensively utilized for benchmarking and cover many aspects of life.

The variate number, dataset size and sampling frequency of each dataset are summarized in Table 5 . We follow standard protocol [55] and split all datasets into training, validation and test set in chronological order by the ratio of 6:2:2 for the ETT and PEMS dataset and 7:1:2 for the other datasets. And training, validation and test sets are zero-mean normalized with the mean and standard deviation of training set. Each of above datasets only contains one continuous long time series, and we obtain samples by sliding window.

More introduction of the datasets are as follow:

1) **Weather**[1] contains 21 meteorological indicators of Germany in 2020.

2) **Traffic**[2] contains the road occupancy rates measured by 862 different sensors on San Francisco Bay area freeways in 2 years.

3) **ECL**(Electricity)[3] contains hourly electricity consumption of 321 clients from 2012 to 2014.

4) **ETT**(Electricity Transformer Temperature)[4] contains the data collected from two different electricity transformers with 2 different resolutions (15 minutes and 1 hour) by 7 sensors.

5) **Solar**(Solar-Eneryg)[5] contains 137 time series representing the solar power production in Alabama state in 2006.

6) **PEMS**[6] is collected from California freeway and contains 4 subsets.

7) **Exchange**[7] the daily exchange rates of eight different countries ranging from 1990 to 2016.

8) **ILI**(Influenza-Like Illness)[8] contains 7 indicators of influenza-like illness (ILI) patients in the United States between 2002 and 2021.

### A.2 Univariate Short-term Forecasting Datasets

We conduct univariate short-term forecasting experiments on 7 popular datasets, including M1, M3, M4, Tourism, NN5, Hospital and KDD Cup. We emphasize the difference between multivariate and univariate tasks as follows. In multivariate tasks, all input samples are obtained by sliding window from the same multivariate long series. Therefore, there is a high degree of similarity between the input samples in multivariate tasks. In contrast, the input samples in univariate tasks are collected from many different univariate time series sources. As a result, the input samples in univariate tasks may differ from each other and have quite different temporal property, making the univariate short-term forecasting tasks much more difficult.

Table 6 summarizes details of statistics of univariate short-term forecasting datasets. And more introduction of the datasets are as follow:

1) **M1**[9] contains 3 subsets with different frequency: Yearly, Quarterly and Monthly. The series are belonging to 7 different domains: macro 1, macro 2, micro 1, micro 2, micro 3, industry and demographic.

2) **M3**[10] contains 4 subsets with different frequency: Yearly, Quarterly, Monthly and Others. The series are belonging to 6 different domains: demographic, micro, macro, industry, finance and other.

3) **M4**[11] contains 6 subsets with different frequency: Yearly, Quarterly, Monthly, Weekly, Daily and Hourly. The series are belonging to a wide range of economic, industrial, financial and demographic areas.

---

[1] https://www.bgc-jena.mpg.de/wetter/

[2] https://pems.dot.ca.gov/

[3] https://archive.ics.uci.edu/ml/datasets/ElectricityLoadDiagrams20112014

[4] https://github.com/zhouhaoyi/ETDataset

[5] https://www.nrel.gov/grid/solar-power-data.html

[6] http://pems.dot.ca.gov/

[7] https://github.com/laiguokun/multivariate-time-series-data

[8] https://gis.cdc.gov/grasp/fluview/fluportaldashboard.html

[9] https://doi.org/10.2307/2345077

[10] https://doi.org/10.1016/S0169-2070(00)00057-1

[11] https://doi.org/10.1016/j.ijforecast.2019.04.014

4) **Tourism**[12] contains 3 subsets with different frequency (Yearly, Quarterly and Monthly) used in the Kaggle Tourism forecasting competition. Considering the dataset size, we only use Tourism Quarterly and Tourism Monthly.

5) **NN5**[13] contains weekly time series from the banking domain.

6) **Hospital Dataset**[14] contains 767 monthly time series that represent the patient counts related to medical products from January 2000 to December 2006.

7) **KDD Cup**[15] contains 270 hourly time series representing the air quality levels by multiple measurements such as PM2.5, PM10, NO2, CO, O3 and SO2.

## B Experiment details

### B.1 Long-term Forecasting

**Implementation Details** Our method is trained with the L2 loss, using the ADAM [16] optimizer with an initial learning rate in $\{10^{-3}, 5 \times 10^{-4}, 10^{-4}\}$. The default training process is 50 epochs with proper early stopping. The mean square error (MSE) and mean absolute error (MAE) are used as metrics. All the experiments are repeated 5 times with different seeds and the means of the metrics are reported as the final results. All the deep learning networks are implemented in PyTorch[38] and conducted on NVIDIA A100 40GB GPU.

**Model Parameter** By default, DeformableTST contains 4 Transfomer blocks. And we adopt downsampling layers between two blocks, which will halve the series' length and double the model dimension. The dimension $D$ of the first block is set as 16. The expansion $\alpha$ is set as 4. The number of important time points $N_{samp}$ is set as 12. We optionally adopt non-overlap patching depended on the input lengths. When input length is 96, we do not adopt patching. When input length is 384, the patch size is 4. When input length is 768, the patch size is 8. For baseline models, if the original papers conduct long-term forecasting experiments on the dataset we use, we follow the official codes with the recommended model parameters in the original papers, including the number of blocks, model dimension, etc. Otherwise, their model parameters are searched from following searching space: number of blocks $L$ from $\{2, 4, 6\}$, model dimension $D$ from $\{64, 128, 256\}$ and FFN expansion $\alpha$ from $\{1, 2, 4, 8\}$.

**Metric** We adopt the mean square error (MSE) and mean absolute error (MAE) of multivariate time series forecasting as metrics.

$$\text{MSE} = \frac{1}{T} \sum_{i=0}^{T} (\widehat{\mathbf{Y}}_i - \mathbf{Y}_i)^2$$

$$\text{MAE} = \frac{1}{T} \sum_{i=0}^{T} \left| \widehat{\mathbf{Y}}_i - \mathbf{Y}_i \right|$$

where $\widehat{\mathbf{Y}}, \mathbf{Y} \in \mathbb{R}^{T \times M}$ are the $M$ variates prediction results of length $T$ and corresponding ground truth. $\widehat{\mathbf{Y}}_i$ means the $i$-th time point in the prediction result.

### B.2 Multi-variate Short-term Forecasting

**Implementation Details** Our method is trained with the L2 loss, using the ADAM [16] optimizer with an initial learning rate in $\{10^{-3}, 5 \times 10^{-4}, 10^{-4}\}$. The default training process is 50 epochs with proper early stopping. The mean square error (MSE) and mean absolute error (MAE) are used as metrics. All the experiments are repeated 5 times with different seeds and the means of the metrics are reported as the final results. All the deep learning networks are implemented in PyTorch[38] and conducted on NVIDIA A100 40GB GPU.

---

[12]https://cran.r-project.org/web/packages/Tcomp
[13]http://www.neural-forecasting-competition.com/NN5/
[14]https://cran.r-project.org/package=expsmooth
[15]https://www.kdd.org/kdd2018/kdd-cup

**Model Parameter**  By default, DeformableTST contains 6 Transformer blocks with the model dimension $D = 256$ and FFN expansion $\alpha = 4$. The number of important time points $N_{samp}$ is in the range of 1 to 12 in short-term forecasting tasks, which is depended on the different input lengths. Due to the limited input lengths, we do not adopt patching and downsampling layers in short-term forecasting tasks. For baseline models, we follow the official codes with the recommended model parameters and some of their model parameters are re-searched from following searching space: number of blocks $L$ from $\{2, 4, 6\}$, model dimension $D$ from $\{64, 128, 256\}$ and FFN expansion $\alpha$ from $\{1, 2, 4, 8\}$.

**Metric**  We adopt the mean square error (MSE) and mean absolute error (MAE) of multivariate time series forecasting as metrics.

$$\text{MSE} = \frac{1}{T} \sum_{i=0}^{T} (\widehat{\mathbf{Y}}_i - \mathbf{Y}_i)^2$$

$$\text{MAE} = \frac{1}{T} \sum_{i=0}^{T} \left| \widehat{\mathbf{Y}}_i - \mathbf{Y}_i \right|$$

where $\widehat{\mathbf{Y}}, \mathbf{Y} \in \mathbb{R}^{T \times M}$ are the $M$ variates prediction results of length $T$ and corresponding ground truth. $\widehat{\mathbf{Y}}_i$ means the $i$-th time point in the prediction result.

### B.3  Univariate Short-term Forecasting

**Implementation Details**  Our method is trained with the SMAPE loss, using the ADAM [16] optimizer with an initial learning rate of $5 \times 10^{-4}$. The default training process is 50 epochs with proper early stopping. Following [46], we fix the input length to be 2 times of prediction length for all models. All the experiments are repeated 5 times with different seeds and the means of the metrics are reported as the final results.

**Model Parameter**  By default, DeformableTST contains 4 Transformer blocks with the model dimension $D = 256$ and FFN expansion $\alpha = 4$. Due to the limited input lengths, we do not adopt patching and downsampling layers in short-term forecasting tasks.

**Metric**  For the M4 datasets, following [31, 37], we adopt the symmetric mean absolute percentage error (SMAPE), mean absolute scaled error (MASE) and overall weighted average (OWA) as the metrics, which can be calculated as follows:

$$\text{SMAPE} = \frac{200}{T} \sum_{i=1}^{T} \frac{|\mathbf{Y}_i - \widehat{\mathbf{Y}}_i|}{|\mathbf{Y}_i| + |\widehat{\mathbf{Y}}_i|}, \qquad \text{MAPE} = \frac{100}{T} \sum_{i=1}^{T} \frac{|\mathbf{Y}_i - \widehat{\mathbf{Y}}_i|}{|\mathbf{Y}_i|},$$

$$\text{MASE} = \frac{1}{T} \sum_{i=1}^{T} \frac{|\mathbf{Y}_i - \widehat{\mathbf{Y}}_i|}{\frac{1}{T-p} \sum_{j=p+1}^{T} |\mathbf{Y}_j - \mathbf{Y}_{j-p}|}, \qquad \text{OWA} = \frac{1}{2} \left[ \frac{\text{SMAPE}}{\text{SMAPE}_{\text{Naïve2}}} + \frac{\text{MASE}}{\text{MASE}_{\text{Naïve2}}} \right],$$

where $p$ is the periodicity of the data. $\widehat{\mathbf{Y}}, \mathbf{Y} \in \mathbb{R}^{T \times M}$ are the $M$ variates prediction results of length $T$ and corresponding ground truth. $\widehat{\mathbf{Y}}_i$ means the $i$-th time point in the prediction result.

For other datasts, we adopt the symmetric mean absolute percentage error (SMAPE) as the metric, which can be calculated as follows:

$$\text{SMAPE} = \frac{200}{T} \sum_{i=1}^{T} \frac{|\mathbf{Y}_i - \widehat{\mathbf{Y}}_i|}{|\mathbf{Y}_i| + |\widehat{\mathbf{Y}}_i|},$$

where $\widehat{\mathbf{Y}}, \mathbf{Y} \in \mathbb{R}^{T \times M}$ are the $M$ variates prediction results of length $T$ and corresponding ground truth. $\widehat{\mathbf{Y}}_i$ means the $i$-th time point in the prediction result.

## C  Pseudo-code of DeformableTST

We provide the pseudo-code of DeformableTST in Algorithm 1.

**Algorithm 1** DeformableTST - Overall Architecture.

**Require:** Input time series $\mathbf{X}_{in} \in \mathbb{R}^{B \times I \times M}$; batch size $B$; variates number $M$; input length $I$; prediction length $T$; DeformableTST block number $L$; feature series's embedding dimension in the $i$-th block $D_i$; feature series's length in the $i$-th block $N_i$. Boolean flag to indicate using downsampling layers or not `Use_Downsampling`.

1: $\mathbf{X}_{in} = \texttt{RevIN}(\mathbf{X}_{in}, \text{mode=norm})$ ▷ $\mathbf{X} \in \mathbb{R}^{B \times I \times M}$

2: $\mathbf{X}_{in} = \mathbf{X}_{in}.\texttt{transpose}$ ▷ $\mathbf{X} \in \mathbb{R}^{B \times M \times I}$

3: $\mathbf{X}_{in} = \mathbf{X}_{in}.\texttt{reshape}$ ▷ $\mathbf{X}_{in} \in \mathbb{R}^{(BM) \times 1 \times I}$

4: ▷ Embedding the input time series variate-independently with optional patching.

5: $\mathbf{X}_0 = \texttt{Embedding}(\mathbf{X}_{in})$ ▷ $\mathbf{X}_0 \in \mathbb{R}^{(BM) \times D_0 \times N_0}$

6: **for** $i$ **in** $\{1, \ldots, L\}$**:** ▷ Run through DeformableTST blocks.

7:     ▷ The local perception unit (LPU) is used to learn the local temporal information.

8:     $\mathbf{X}_i^{local} = \texttt{LPU}(\mathbf{X}_{i-1})$ ▷ $\mathbf{X}_i^{local} \in \mathbb{R}^{(BM) \times D_i \times N_i}$

9:     ▷ The deformable attention is adopted to learn the global temporal information.

10:     $\mathbf{X}_i^{global} = \texttt{LayerNorm}\big(\mathbf{X}_i^{local} + \texttt{DeformAttention}(\mathbf{X}_i^{local})\big)$ ▷ $\mathbf{X}_i^{global} \in \mathbb{R}^{(BM) \times D_i \times N_i}$

11:     ▷ The feed-forward network injected with a depth-wise convolution (ConvFFN) is used to learn the local temporal information and the new feature representation.

12:     $\mathbf{X}_i = \texttt{LayerNorm}\big(\mathbf{X}_i^{global} + \texttt{ConvFFN}(\mathbf{X}_i^{global})\big)$ ▷ $\mathbf{X}_i \in \mathbb{R}^{(BM) \times D_i \times N_i}$

13:     ▷ Adopting the optional downsampling layer between two blocks.

14:     **if** $i < L$ **and** `Use_Downsampling` **is True:**

15:         $\mathbf{X}_i = \texttt{DownSampling}(\mathbf{X}_i)$ ▷ $\mathbf{X}_i \in \mathbb{R}^{(BM) \times (2D_i) \times (N_i/2)}$

16: **End for**

17: $\mathbf{X}_L = \mathbf{X}_L.\texttt{reshape}$ ▷ $\mathbf{X}_L \in \mathbb{R}^{B \times M \times (D_L N_L)}$

18: $\hat{\mathbf{Y}} = \texttt{Projection}(\mathbf{X}_L)$ ▷ Obtaining the forecasting series with Projection, $\hat{\mathbf{Y}} \in \mathbb{R}^{B \times M \times T}$

19: $\hat{\mathbf{Y}} = \hat{\mathbf{Y}}.\texttt{transpose}$ ▷ $\hat{\mathbf{Y}} \in \mathbb{R}^{B \times T \times M}$

20: $\hat{\mathbf{Y}} = \texttt{RevIN}(\hat{\mathbf{Y}}, \text{mode=denorm})$ ▷ $\hat{\mathbf{Y}} \in \mathbb{R}^{B \times T \times M}$

21: **Return** $\hat{\mathbf{Y}}$ ▷ Return the prediction result $\hat{\mathbf{Y}}$

## D  Parameter Sensitivity

To evaluate the parameter sensitivity of our DeformableTST, we perform experiments with varying model parameters, including number of blocks ranging from $L = \{2, 4, 6\}$, model dimension ranging from $D = \{16, 32, 64\}$, FFN expansion ranging from $\alpha = \{1, 2, 4, 8\}$, number of important time points ranging from $N_{samp} = \{6, 12, 24\}$ and learning rate ranging from $lr = \{10^{-3}, 5 \times 10^{-4}, 10^{-4}\}$.

The results are shown in Figure 6. In general, our model is robust to the choice of model parameters. Compared with the default block number $L = 4$, stacking more blocks will bring further performance improvement. Considering both performance and efficiency, we recommend to fix the block number as 4 in long term forecasting tasks.

We also compare our model sensitivity to patch size with PatchTST's. As shown in Figure 6, our model is less sensitive to the choice of patch size, therefore successfully getting rid of the over-reliance of patching.

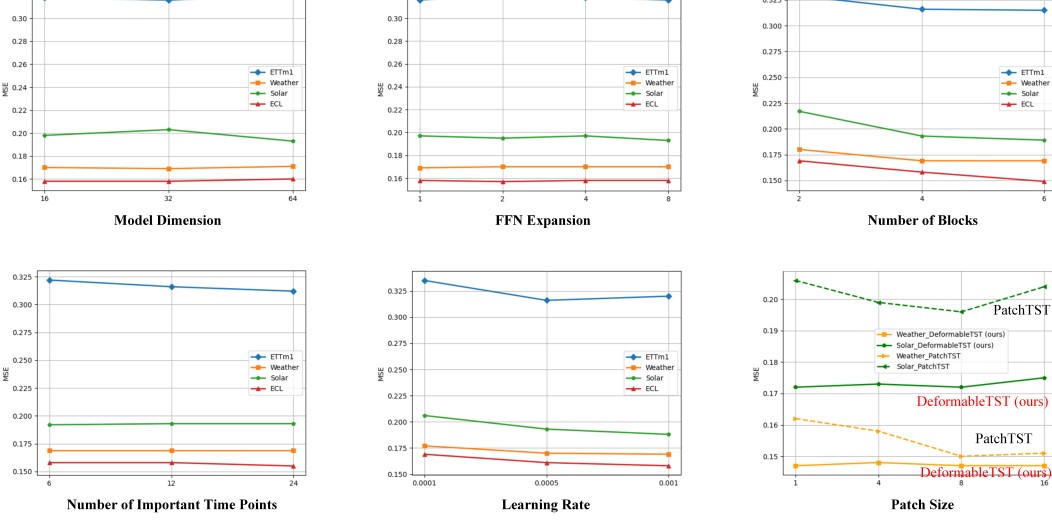

Figure 6: Parameter sensitivity. For patch size, we conduct experiments under input-384-predict-96 settings and adopt PatchTST as comparison. For other parameters, we conduct experiments under input-96-predict-96 settings.

# E More ERF Results

In Section 1, we visualize the effective receptive fields (ERFs) of PatchTST [35] based on [27, 14] to see which parts of the time points in input series are focused by the model when extracting temporal representations. Here we provide more ERF visualization results with other patch-based Transformer forecasters. In a ERF figure, a brighter area means that the model tends to focus on these time points when extracting temporal representation, and thus these time points will contribute more to the middle point of the final representation.

As shown in Figure 7, the ERFs of the patch-based Transformer forecasters highly rely on the guidance of patching. If without patching, nearly all time points in input series are equally focused by the model and the model performs worse, exposing the problem of distracted attention. This finding means that attention has not learned to distinguish the importance of each time point in input series, leading to trivial representation. After patching, these Transformer forecasters tend to foucs on some important time points based on a patch partition. This phenomenon means that: although patching can help attention better focus on important time points, the guidance of patching tend to distribute the focused points evenly among all patches. But in real-world scenarios, it is possible that the important time points are not evenly distributed among all patches, which is inconsistent with the tendency in the guidance of patching, leading to the inferior performance of these patch-based Transformer forecasters in some cases. There is also some difference among the ERFs of these patch-based Transformer forecasters due to their different specific designs. PatchTST [35] tends to only focus on a very few time points in each patch and ignore others. But CARD [51] can focus on more time points in one patch thanks to its additional attention across feature dimension, which aligns the information within patch. And the ERF of GPT4TS [58] also reveals the autoregressive property in GPT [40] backbone, making it only focus on the time points before the the middle point.

By contrast, we provide the ERF of our DeformableTST in Figure 8 for comparison. By default, we set $N_{samp} = 12$. Under input-96 settings, our DeformableTST does not adopt patching technique but can still focus on a small number of important time points, proving that our DeformableTST can foucs well by itself without the need of patching. Under input-384 settings, we adopt a small size patching and divide the input series into 96 patches to alleviate the efficiency issue. Under such condition, our DeformableTST still does not focus based on a patch partition. Although adopting patching technique, the focused time points in our DeformableTST is not evenly distributed among all 96 patches, therefore being less reliant on patching.

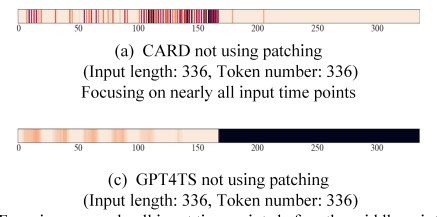

(a) CARD not using patching
(Input length: 336, Token number: 336)
Focusing on nearly all input time points

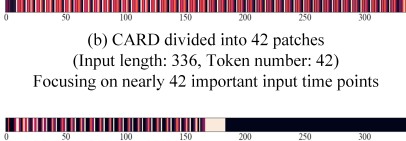

(b) CARD divided into 42 patches
(Input length: 336, Token number: 42)
Focusing on nearly 42 important input time points

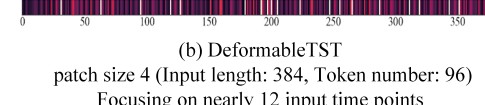

(c) GPT4TS not using patching
(Input length: 336, Token number: 336)
Focusing on nearly all input time points before the middle point

(d) GPT4TS divided into 42 patches
(Input length: 336, Token number: 42)
Focusing on nearly 21 important input time points before the middle point

Figure 7: The Effective Receptive Field (ERF) of more patch-based Transformer forecasters. A brighter area means that these time points are focused by the model when extracting temporal representation.

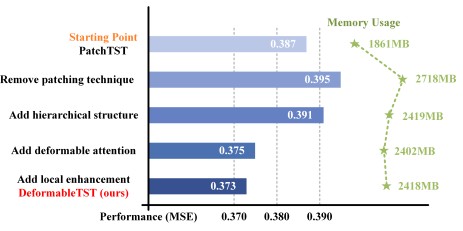

(a) DeformableTST
Not using patching (Input length: 96, Token number: 96)
Focusing on nearly 12 input time points

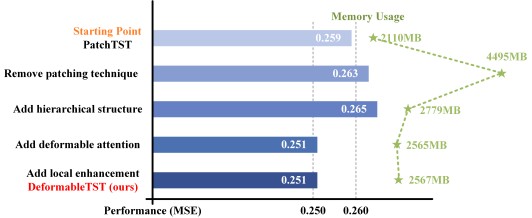

(b) DeformableTST
patch size 4 (Input length: 384, Token number: 96)
Focusing on nearly 12 input time points

Figure 8: The Effective Receptive Field (ERF) of DeformableTST. A brighter area means that these time points are focused by the model when extracting temporal representation.

# F  More Results of Model Analysis in Section 5

## F.1  More Ablation Results in Section 5.1

More ablation results are provided in Figure 9.

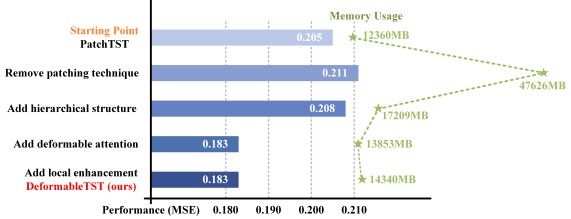

(a) Ablation results in ETTm1 dataset with input length 96

(b) Ablation results in Weather dataset with input length 96

(c) Ablation results in ECL dataset with input length 96

Figure 9: More ablation study results in long-term forecasting tasks. From the top to the bottom, each row means one design we add on PatchTST to modify it into our DeformableTST. We report the averaged MSE of four prediction lengths. The memory usage is recorded under input-96-predict-96 setting with the same batch size. A lower MSE (a shorter blue bar) and a smaller memory usage (a green star closer to the vertical axis) means better performance and efficiency.

## F.2  More Comparison Results in Section 5.2

**Case Study**    We provide the visualization of learned important time points as an intuitive comparision for some popular sparse attentions in time series community. As shown in Figure 10, the keys of window attention are restricted within the local window. And the important keys for ProbSparse

attention tend to gather in a small area around the time point that most fits the prior. This property makes window attention and ProbSparse attention only focus on a small local area and fail to find the important time points in a global range, leading to the loss of information. Meanwhile, imporatant time points refer to the time points that reflect the property of time series and make contribution to better performance. And the types of imporatant time points are varied (e.g., time point in the similar changing stage, the inflexion point, the extremal point and so on). But AutoCorrelation can only foucs on the time points in the similar changing stage due to its prior, resulting in the lack of diversity. By contrast, our deformable attention can find the important time points in a global range for the reference points in the sampling process are uniformly distributed throughout the whole input time series. And our deformable attention can also learn different types of important time points because it is less reliant on specific priors and can foucs on any appropriate important time points based on the learnable offsets learned from the input series. Therefore, our deformable attention can find more types of important time points in a wider range, therefore being more flexible to the diverse property in different time series and performing better than other prior-based sparse attentions.

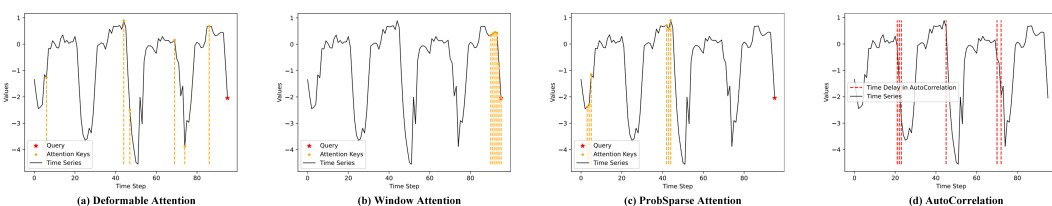

Figure 10: Visualization of learned important time points. For clearness, we show the top-6 keys with respect to the last query for attentions and show the top-6 time delays for AutoCorrelation.

# G   Model Efficiency

We comprehensively compare the forecasting performance, training speed, and memory usage of some advanced Transformer-based models. And we compare the efficiency under two representative conditions: (1) the dataset is of a large variate number, (2) the experiment setting is of a long input length and prediction length. And the results are shown in Figure 11. Considering both performance and efficiency, our DeformableTST shows great superiority than other Transformer-based competitors, therefore being an ideal choice in time series forecasting.

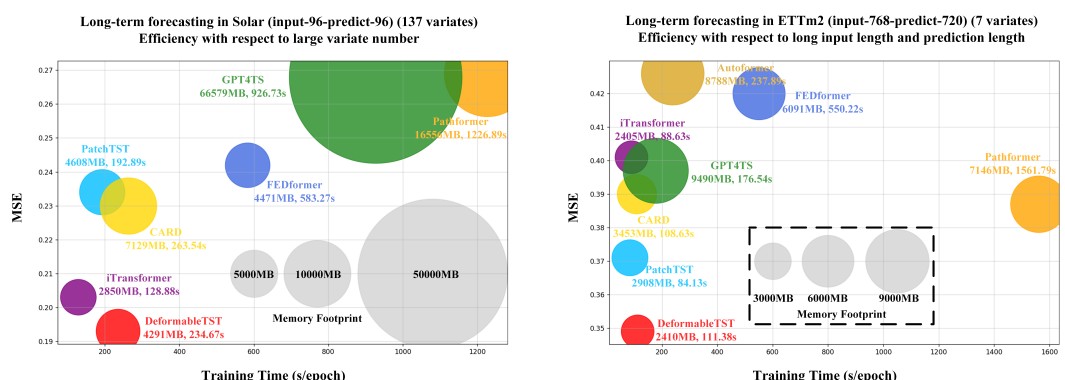

Figure 11: Model efficiency comparison.

# H   Error Bar

We report the standard deviation of DeformableTST performance under five runs with different random seeds in Table 7, which exhibits that the performance of DeformableTST is stable.

Table 7: Error bar of DeformableTST.

| Dataset | ETTh1 | | ETTh2 | | ETTm1 | | ETTm2 | |
|---|---|---|---|---|---|---|---|---|
| Horizon | MSE | MAE | MSE | MAE | MSE | MAE | MSE | MAE |
| 96 | 0.373±0.002 | 0.396±0.001 | 0.281±0.001 | 0.334±0.001 | 0.316±0.001 | 0.358±0.001 | 0.178±0.000 | 0.262±0.000 |
| 192 | 0.427±0.001 | 0.427±0.000 | 0.353±0.002 | 0.382±0.001 | 0.354±0.001 | 0.380±0.001 | 0.243±0.002 | 0.301±0.002 |
| 336 | 0.437±0.003 | 0.426±0.002 | 0.341±0.000 | 0.379±0.000 | 0.379±0.002 | 0.405±0.002 | 0.310±0.001 | 0.348±0.001 |
| 720 | 0.464±0.003 | 0.462±0.003 | 0.410±0.004 | 0.431±0.003 | 0.443±0.004 | 0.438±0.003 | 0.400±0.002 | 0.398±0.002 |

| Dataset | Weather | | Solar-Energy | | ECL | | Traffic | |
|---|---|---|---|---|---|---|---|---|
| Horizon | MSE | MAE | MSE | MAE | MSE | MAE | MSE | MAE |
| 96 | 0.169±0.000 | 0.213±0.000 | 0.193±0.001 | 0.232±0.001 | 0.158±0.002 | 0.255±0.002 | 0.418±0.002 | 0.271±0.002 |
| 192 | 0.216±0.002 | 0.255±0.001 | 0.225±0.002 | 0.261±0.001 | 0.168±0.004 | 0.265±0.004 | 0.437±0.004 | 0.276±0.003 |
| 336 | 0.271±0.001 | 0.294±0.001 | 0.239±0.000 | 0.267±0.000 | 0.183±0.003 | 0.280±0.004 | 0.449±0.005 | 0.290±0.005 |
| 720 | 0.347±0.003 | 0.344±0.002 | 0.238±0.000 | 0.265±0.000 | 0.223±0.004 | 0.313±0.004 | 0.477±0.004 | 0.303±0.004 |

# I More Implementation Details about Deformable Attention

## I.1 Implementation of Linear Interpolation

As mentioned in Section 3.2, we sample the important time points based on a set of learnable coordinates called sampling points $\mathbf{T}_{samp}$. In practice, after obtaining the sampling points $\mathbf{T}_{samp}$, we achieve this sampling process via linear interpolation following [48, 60]. In details, we calculate the values of these important time points by linear interpolation $\phi(\cdot;\cdot)$ to make this sampling process differentiable. And the linear interpolation $\phi(\cdot;\cdot)$ is calculated as follows:

$$\phi\left(\mathbf{X}; \mathbf{T}_{samp}\right) = \sum_{t=0}^{N-1} g(t, \text{De-normalize}(\mathbf{T}_{samp}))\mathbf{X}[:, t, :], \tag{11}$$

where $g(a, b) = \max(0, 1 - |a - b|)$ and $t$ indexes all the coordinates on $\mathbf{X} \in \mathbb{R}^{M \times D \times N}$. The value of $\mathbf{T}_{samp}$ is de-normalized back to the range of $[0, N-1]$ before passed into $g$. As $g$ would be non-zero only on the 2 integral coordinates closest to $\mathbf{T}_{samp}$, it simplifies Eq.(11) to calculate the value of each important time point as the weighted average of its only 2 closest time points.

## I.2 Deformable Relative Positional Bias

Due to the deforming process and hierarchical representation, the fixed absolute positional embedding is not suitable for our design. Instead, we adopt a relative position bias to encode the position information into the attention map [23], which represents the relative positional information between the query token series $\mathbf{Q}$ and the key token series $\tilde{\mathbf{K}}$.

Considering the input feature series $\mathbf{X} \in \mathbb{R}^{M \times D \times N}$ of the attention module, the relative coordinate displacements of this length-$N$ series contain $(2N - 1)$ different values and lie in the range of $[-N, N]$. Then we will maintain a learnable parameterized bias table $\hat{\mathbf{B}} \in \mathbb{R}^{H \times (2N-1) \times 1}$, in which each element represents one of the above $(2N - 1)$ relative coordinate displacements.

Meanwhile, to represent the relative positional information between the query token series $\mathbf{Q}$ and the key token series $\tilde{\mathbf{K}}$, we also need to denote the coordinates of $\mathbf{Q}$ and $\tilde{\mathbf{K}}$. The coordinates of $\mathbf{Q}$ are generated from a 1D uniform grid with size $N$. It indicates the 1D coordinates of all $N$ time points in the query series $\mathbf{Q}$. These coordinate values are normalized to $[-1, +1]$, where $-1$ indicates the start of the series and $+1$ means the end of the series. And the coordinates of $\tilde{\mathbf{K}}$ are the sampling point $\mathbf{T}_{samp}$. Then the relative position $\mathbf{R} \in \mathbb{R}^{N \times N_{samp} \times 1}$ is calculated as follows:

$$\mathbf{R} = \mathbf{Q}_{coord} - \text{Transpose}(\mathbf{T}_{samp}) \tag{12}$$

Then we clip $\mathbf{R}$ by $-1$ and $+1$ and obtain deformable relative positional bias $\mathbf{B}$ by linear interpolation to the learnable parameterized bias table $\hat{\mathbf{B}}$ with the continuous relative displacements $\mathbf{R}$ as follows:

$$\mathbf{B} = \phi(\hat{\mathbf{B}}; \mathbf{R}) \tag{13}$$

# J More Discussions on Patching and Transformer-based Models

## J.1 Discussions on Patching

**Impact on Performance**   As shown in Figure 1 and 7, although patching can help attention better focus on important time points, the guidance of patching tend to distribute the focused points evenly among all patches. But in real-world scenarios, it is possible that the important time points are not evenly distributed among all patches. In some cases, most of the important time points are only distributed in a few patches while other patches only contain unimportant time points. Such cases are inconsistent with the tendency in the guidance of patching, leading to the inferior performance of the patch-based Transformer forecasters in these cases. And patching technique must work with a very long input length because leveraging patching on short time series leads to very few tokens, limiting attention's ability in long-term modeling. These drawbacks limit the performance and applicability of the previous patch-based Transformer forecasters.

**Impact on Efficiency**   In addition to helpling attention focus better, patching can also improve the efficiency by reducing the number of tokens. When facing a very long input length (e.g., 384 and 768), we still adopt a small patch size in our design for better efficiency. Therefore this paper is not a call to completely abandon patching technique in all cases, but an extension to scenarios where patching technique is not suitable.

## J.2 Compared with Typical Transformer-based Models

To highlight how we innovate and upgrade the Transformer-based model, we compare our DeformableTST with some milestone Transformer-based models in time series community.

**Compared with PatchTST [35]**   PatchTST is a milestone Transformer-based model and DeformableTST can be seen as an improvement of it to solve the problem of over-reliance on patching. PatchTST highly relies on patching for patching can force the attention to focus on only a few important time points within each patch and ignore other time points within the patch. But the choose of patch size is a dilemma in practice: a too large patch size could lead to many other time points within the patch being ignored, resulting in the risk of neglecting other priorities. And a too small patch size will lead to a huge amount of tokens, making it hard to focus. This dilemma cannot be resolved by PatchTST itself. To address this issue, we propose deformable attention, an attention mechanism that can focus well by itself, to get rid of the need of patching. Since deformable attention enjoys great focusing ability, we can use a small patch size to mitigate the problem of some priorities being ignored and free to worry about attention being hard to focus on important time points, successfully reducing the reliance on patching and resolving the dilemma.

**Compared with iTransformer [22]**   DeformableTST and iTransformer are the lastest Transformer-based forecasters. Both methods focus on the underperformance issue of previous attention mechanism in temporal modeling and devote to designing a more powerful Transformer-based forecaster. But they hold different opinions. Although iTransformer still adopts a Transformer architecture, it insists that attention is not suitable for temporal modeling while linear layers are more appropriate. This design makes its performance more similar to linear-based forecasters rather than Transformer-based ones, being less competitive in difficult tasks (e.g., univariate short-term forecasting tasks with lower similarity between samples). Different from iTransformer, our DeformableTST follows the tradition of Transformer-based models to still uses attention for temporal modeling. Specifically, in this paper, through experiments and analysis, we further attribute the underperformance issue of previous attention mechanism to their over-reliance on patching and thus propose deformable attention to solve this problem, successfully designing a more powerful Transformer-based forecaster and broadening the applicability of Transformer-based models.

# K Limitations and Future Work

In this paper, we mainly focus on how to better use attention in temporal modeling and do not consider the multivariate correlating. It will be our future work to study how to further capture the multivariate correlation in our model, which can hopefully improve the performance, especially in datasets with

large number of variates. Besides, we will further explore the potential of our DeformableTST in more time series analysis tasks and further develop its performance by large-scale pre-training in the future.

## L   Ethics Statement and Broader Impact

Our work only focuses on the time series forecasting problem, so there is no potential ethical risk.

Our model achieves the state-of-the-art performance on a wider range of time series forecasting tasks, covering a large amount of real-world scenarios. Therefore, the proposed model makes it promising to tackle real-world forecasting problem, helping our society make better decisions and prevent risks in advance. And we hope our findings can prompt people to rethink the relationship between Transformer-based models and patching technique, thereby designing more powerful Transformer-based forecasters with a wider range of applicability.

Our paper mainly focuses on scientific research and has no obvious negative social impact.

## M   Reproducibility Statement

In the main text, we have strictly formalized the model architecture with equations. All the implementation details are included in the Appendix, including dataset descriptions, metrics, model, and experiment configurations. Code is available at this repository: `https://github.com/luodhhh/DeformableTST`.

## N   Full Results

Due to the space limitation of the main text, we place the full results of all experiments and results of more baselines in the following subsections. And we also provide the showcases in Appendix O.

### N.1   Long-term Forecasting with Input Length 96

The full results of long-term forecasting with input length 96 are provided in Table 8. And more results with more baselines are also included.

### N.2   Long-term Forecasting with Input Length 384

The full results of long-term forecasting with input length 384 are provided in Table 9. And more results with more baselines are also included.

### N.3   Long-term Forecasting with Input Length 768

The full results of long-term forecasting with input length 768 are provided in 10. And more results with more baselines are also included.

### N.4   Multivariate Short-term Forecasting

The full results of multivariate short-term forecasting tasks are provided in Table 11. And more results with more baselines are also included.

### N.5   Univariate Short-term Forecasting

The full results of M4 datasets are provided in Table 12. And the full results of other datasets are provided in Table 13.

Table 8: Full results of the **long-term forecasting tasks** with input length 96. We compare extensive competitive models under different prediction lengths. The input sequence length is set to 96 for all baselines. *Avg* means the average results from all four prediction lengths.

| Models | | DeformableTST (Ours) | | Pathformer [6] | | CARD [51] | | GPT4TS [58] | | PatchTST [35] | | iTransformer [22] | | FEDformer [57] | | Autoformer [47] | | RLinear [18] | | TiDE [9] | | TimesNet [46] | | DLinear [52] | | SCINet [20] | |
|---|---|---|---|---|---|---|---|---|---|---|---|---|---|---|---|---|---|---|---|---|---|---|---|---|---|---|---|
| Metric | | MSE | MAE | MSE | MAE | MSE | MAE | MSE | MAE | MSE | MAE | MSE | MAE | MSE | MAE | MSE | MAE | MSE | MAE | MSE | MAE | MSE | MAE | MSE | MAE | MSE | MAE |
| ETTh1 | 96 | 0.373 | 0.396 | 0.396 | 0.409 | 0.398 | 0.405 | 0.394 | 0.404 | 0.414 | 0.419 | 0.386 | 0.405 | 0.376 | 0.419 | 0.449 | 0.459 | 0.386 | 0.395 | 0.479 | 0.464 | 0.384 | 0.402 | 0.386 | 0.400 | 0.654 | 0.599 |
| | 192 | 0.427 | 0.427 | 0.449 | 0.434 | 0.448 | 0.433 | 0.463 | 0.442 | 0.460 | 0.445 | 0.441 | 0.436 | 0.420 | 0.448 | 0.500 | 0.482 | 0.437 | 0.424 | 0.525 | 0.492 | 0.436 | 0.429 | 0.437 | 0.432 | 0.719 | 0.631 |
| | 336 | 0.437 | 0.426 | 0.476 | 0.447 | 0.494 | 0.457 | 0.487 | 0.452 | 0.501 | 0.466 | 0.487 | 0.458 | 0.459 | 0.465 | 0.521 | 0.496 | 0.479 | 0.446 | 0.565 | 0.515 | 0.491 | 0.469 | 0.481 | 0.459 | 0.778 | 0.659 |
| | 720 | 0.464 | 0.462 | 0.489 | 0.474 | 0.483 | 0.472 | 0.496 | 0.472 | 0.500 | 0.488 | 0.503 | 0.491 | 0.506 | 0.507 | 0.514 | 0.512 | 0.481 | 0.470 | 0.594 | 0.558 | 0.521 | 0.500 | 0.519 | 0.516 | 0.836 | 0.699 |
| | Avg | 0.425 | 0.428 | 0.453 | 0.441 | 0.456 | 0.442 | 0.460 | 0.443 | 0.469 | 0.454 | 0.454 | 0.447 | 0.440 | 0.460 | 0.496 | 0.487 | 0.446 | 0.434 | 0.541 | 0.507 | 0.458 | 0.450 | 0.456 | 0.452 | 0.747 | 0.647 |
| ETTh2 | 96 | 0.281 | 0.334 | 0.289 | 0.342 | 0.294 | 0.341 | 0.308 | 0.361 | 0.302 | 0.348 | 0.297 | 0.349 | 0.358 | 0.397 | 0.346 | 0.388 | 0.288 | 0.338 | 0.400 | 0.440 | 0.340 | 0.374 | 0.333 | 0.387 | 0.707 | 0.621 |
| | 192 | 0.353 | 0.382 | 0.377 | 0.401 | 0.375 | 0.391 | 0.390 | 0.409 | 0.388 | 0.400 | 0.380 | 0.400 | 0.429 | 0.439 | 0.456 | 0.452 | 0.374 | 0.390 | 0.528 | 0.509 | 0.402 | 0.414 | 0.477 | 0.476 | 0.860 | 0.689 |
| | 336 | 0.341 | 0.379 | 0.375 | 0.399 | 0.419 | 0.426 | 0.421 | 0.438 | 0.426 | 0.433 | 0.428 | 0.432 | 0.496 | 0.487 | 0.482 | 0.486 | 0.415 | 0.426 | 0.643 | 0.571 | 0.452 | 0.452 | 0.594 | 0.541 | 1.000 | 0.744 |
| | 720 | 0.410 | 0.431 | 0.426 | 0.443 | 0.422 | 0.438 | 0.428 | 0.452 | 0.431 | 0.446 | 0.427 | 0.445 | 0.463 | 0.474 | 0.515 | 0.511 | 0.420 | 0.440 | 0.874 | 0.679 | 0.462 | 0.468 | 0.831 | 0.657 | 1.249 | 0.838 |
| | Avg | 0.346 | 0.382 | 0.367 | 0.396 | 0.378 | 0.399 | 0.387 | 0.415 | 0.387 | 0.407 | 0.383 | 0.407 | 0.437 | 0.449 | 0.450 | 0.459 | 0.374 | 0.398 | 0.611 | 0.550 | 0.414 | 0.427 | 0.559 | 0.515 | 0.954 | 0.723 |
| ETTm1 | 96 | 0.316 | 0.358 | 0.329 | 0.361 | 0.327 | 0.359 | 0.334 | 0.370 | 0.329 | 0.367 | 0.334 | 0.368 | 0.379 | 0.419 | 0.505 | 0.475 | 0.355 | 0.376 | 0.364 | 0.387 | 0.338 | 0.375 | 0.345 | 0.372 | 0.418 | 0.438 |
| | 192 | 0.354 | 0.380 | 0.374 | 0.390 | 0.372 | 0.381 | 0.377 | 0.389 | 0.367 | 0.385 | 0.377 | 0.391 | 0.426 | 0.441 | 0.553 | 0.496 | 0.391 | 0.392 | 0.398 | 0.404 | 0.374 | 0.387 | 0.380 | 0.389 | 0.439 | 0.450 |
| | 336 | 0.379 | 0.405 | 0.400 | 0.401 | 0.403 | 0.401 | 0.410 | 0.409 | 0.399 | 0.410 | 0.426 | 0.420 | 0.445 | 0.459 | 0.621 | 0.537 | 0.424 | 0.415 | 0.428 | 0.425 | 0.410 | 0.411 | 0.413 | 0.413 | 0.490 | 0.485 |
| | 720 | 0.443 | 0.438 | 0.467 | 0.439 | 0.467 | 0.438 | 0.471 | 0.442 | 0.454 | 0.439 | 0.491 | 0.459 | 0.543 | 0.490 | 0.671 | 0.561 | 0.487 | 0.450 | 0.487 | 0.461 | 0.478 | 0.450 | 0.474 | 0.453 | 0.595 | 0.550 |
| | Avg | 0.373 | 0.395 | 0.393 | 0.398 | 0.392 | 0.395 | 0.398 | 0.402 | 0.387 | 0.400 | 0.407 | 0.410 | 0.448 | 0.452 | 0.588 | 0.517 | 0.414 | 0.407 | 0.419 | 0.419 | 0.400 | 0.406 | 0.403 | 0.407 | 0.485 | 0.481 |
| ETTm2 | 96 | 0.178 | 0.262 | 0.176 | 0.260 | 0.176 | 0.259 | 0.179 | 0.263 | 0.175 | 0.259 | 0.180 | 0.264 | 0.203 | 0.287 | 0.255 | 0.339 | 0.182 | 0.265 | 0.207 | 0.305 | 0.187 | 0.267 | 0.193 | 0.292 | 0.286 | 0.377 |
| | 192 | 0.243 | 0.301 | 0.245 | 0.306 | 0.246 | 0.306 | 0.247 | 0.308 | 0.241 | 0.302 | 0.250 | 0.309 | 0.269 | 0.328 | 0.281 | 0.340 | 0.246 | 0.304 | 0.290 | 0.364 | 0.249 | 0.309 | 0.284 | 0.362 | 0.399 | 0.445 |
| | 336 | 0.310 | 0.348 | 0.308 | 0.343 | 0.302 | 0.343 | 0.313 | 0.350 | 0.305 | 0.343 | 0.311 | 0.348 | 0.325 | 0.366 | 0.339 | 0.372 | 0.307 | 0.342 | 0.377 | 0.422 | 0.321 | 0.351 | 0.369 | 0.427 | 0.637 | 0.591 |
| | 720 | 0.400 | 0.398 | 0.406 | 0.401 | 0.400 | 0.404 | 0.409 | 0.403 | 0.402 | 0.400 | 0.412 | 0.407 | 0.421 | 0.415 | 0.433 | 0.432 | 0.407 | 0.398 | 0.558 | 0.524 | 0.408 | 0.403 | 0.554 | 0.522 | 0.960 | 0.735 |
| | Avg | 0.283 | 0.327 | 0.284 | 0.328 | 0.281 | 0.328 | 0.287 | 0.331 | 0.281 | 0.326 | 0.288 | 0.332 | 0.305 | 0.349 | 0.327 | 0.371 | 0.286 | 0.327 | 0.358 | 0.404 | 0.291 | 0.333 | 0.350 | 0.401 | 0.571 | 0.537 |
| Weather | 96 | 0.169 | 0.213 | 0.165 | 0.206 | 0.160 | 0.208 | 0.191 | 0.231 | 0.177 | 0.218 | 0.174 | 0.214 | 0.217 | 0.296 | 0.266 | 0.336 | 0.192 | 0.232 | 0.202 | 0.261 | 0.172 | 0.220 | 0.196 | 0.255 | 0.221 | 0.306 |
| | 192 | 0.216 | 0.255 | 0.218 | 0.259 | 0.207 | 0.250 | 0.236 | 0.267 | 0.225 | 0.259 | 0.221 | 0.254 | 0.276 | 0.336 | 0.307 | 0.367 | 0.240 | 0.271 | 0.242 | 0.298 | 0.219 | 0.261 | 0.237 | 0.296 | 0.261 | 0.340 |
| | 336 | 0.271 | 0.294 | 0.266 | 0.292 | 0.265 | 0.292 | 0.287 | 0.303 | 0.278 | 0.297 | 0.278 | 0.296 | 0.339 | 0.380 | 0.359 | 0.395 | 0.292 | 0.307 | 0.287 | 0.335 | 0.280 | 0.306 | 0.283 | 0.335 | 0.309 | 0.378 |
| | 720 | 0.347 | 0.344 | 0.346 | 0.342 | 0.346 | 0.346 | 0.363 | 0.352 | 0.354 | 0.348 | 0.358 | 0.347 | 0.403 | 0.428 | 0.419 | 0.428 | 0.364 | 0.353 | 0.351 | 0.386 | 0.365 | 0.359 | 0.345 | 0.381 | 0.377 | 0.427 |
| | Avg | 0.251 | 0.276 | 0.249 | 0.275 | 0.245 | 0.274 | 0.269 | 0.288 | 0.259 | 0.281 | 0.258 | 0.278 | 0.309 | 0.360 | 0.338 | 0.382 | 0.272 | 0.291 | 0.271 | 0.320 | 0.259 | 0.287 | 0.265 | 0.317 | 0.292 | 0.363 |
| Solar-Energy | 96 | 0.193 | 0.232 | 0.269 | 0.300 | 0.230 | 0.281 | 0.254 | 0.295 | 0.234 | 0.286 | 0.203 | 0.237 | 0.242 | 0.342 | 0.884 | 0.711 | 0.322 | 0.339 | 0.312 | 0.399 | 0.250 | 0.292 | 0.290 | 0.378 | 0.237 | 0.344 |
| | 192 | 0.225 | 0.261 | 0.296 | 0.314 | 0.267 | 0.308 | 0.268 | 0.312 | 0.267 | 0.310 | 0.233 | 0.261 | 0.285 | 0.380 | 0.834 | 0.692 | 0.359 | 0.356 | 0.339 | 0.416 | 0.296 | 0.318 | 0.320 | 0.398 | 0.280 | 0.380 |
| | 336 | 0.239 | 0.267 | 0.320 | 0.334 | 0.289 | 0.319 | 0.327 | 0.336 | 0.290 | 0.315 | 0.248 | 0.273 | 0.290 | 0.296 | 0.941 | 0.723 | 0.397 | 0.369 | 0.368 | 0.430 | 0.319 | 0.330 | 0.353 | 0.415 | 0.304 | 0.389 |
| | 720 | 0.238 | 0.265 | 0.334 | 0.337 | 0.294 | 0.327 | 0.333 | 0.347 | 0.289 | 0.317 | 0.249 | 0.275 | 0.357 | 0.427 | 0.882 | 0.717 | 0.397 | 0.356 | 0.370 | 0.425 | 0.338 | 0.337 | 0.356 | 0.413 | 0.308 | 0.388 |
| | Avg | 0.224 | 0.256 | 0.305 | 0.321 | 0.270 | 0.309 | 0.296 | 0.323 | 0.270 | 0.307 | 0.233 | 0.262 | 0.291 | 0.381 | 0.885 | 0.711 | 0.369 | 0.356 | 0.347 | 0.417 | 0.301 | 0.319 | 0.330 | 0.401 | 0.282 | 0.375 |
| ECL | 96 | 0.158 | 0.255 | 0.161 | 0.256 | 0.155 | 0.246 | 0.189 | 0.268 | 0.181 | 0.270 | 0.148 | 0.240 | 0.193 | 0.308 | 0.201 | 0.317 | 0.201 | 0.281 | 0.237 | 0.329 | 0.168 | 0.272 | 0.197 | 0.282 | 0.247 | 0.345 |
| | 192 | 0.168 | 0.265 | 0.173 | 0.269 | 0.170 | 0.259 | 0.198 | 0.280 | 0.188 | 0.274 | 0.162 | 0.253 | 0.201 | 0.315 | 0.222 | 0.334 | 0.201 | 0.283 | 0.236 | 0.330 | 0.184 | 0.289 | 0.196 | 0.285 | 0.257 | 0.355 |
| | 336 | 0.183 | 0.280 | 0.189 | 0.283 | 0.194 | 0.285 | 0.205 | 0.283 | 0.204 | 0.293 | 0.178 | 0.269 | 0.214 | 0.329 | 0.231 | 0.338 | 0.215 | 0.298 | 0.249 | 0.344 | 0.198 | 0.300 | 0.209 | 0.301 | 0.269 | 0.369 |
| | 720 | 0.223 | 0.313 | 0.233 | 0.319 | 0.229 | 0.316 | 0.246 | 0.316 | 0.246 | 0.324 | 0.225 | 0.317 | 0.246 | 0.355 | 0.254 | 0.361 | 0.257 | 0.331 | 0.284 | 0.373 | 0.220 | 0.320 | 0.245 | 0.333 | 0.299 | 0.390 |
| | Avg | 0.183 | 0.278 | 0.189 | 0.282 | 0.187 | 0.277 | 0.210 | 0.287 | 0.205 | 0.290 | 0.178 | 0.270 | 0.214 | 0.327 | 0.227 | 0.338 | 0.219 | 0.298 | 0.251 | 0.344 | 0.192 | 0.295 | 0.212 | 0.300 | 0.268 | 0.365 |
| Traffic | 96 | 0.418 | 0.271 | 0.490 | 0.314 | 0.448 | 0.300 | 0.531 | 0.349 | 0.462 | 0.295 | 0.395 | 0.268 | 0.587 | 0.366 | 0.613 | 0.388 | 0.649 | 0.389 | 0.805 | 0.493 | 0.593 | 0.321 | 0.650 | 0.396 | 0.788 | 0.499 |
| | 192 | 0.437 | 0.276 | 0.495 | 0.319 | 0.465 | 0.303 | 0.535 | 0.350 | 0.466 | 0.296 | 0.417 | 0.276 | 0.604 | 0.373 | 0.616 | 0.382 | 0.601 | 0.366 | 0.756 | 0.474 | 0.617 | 0.336 | 0.598 | 0.370 | 0.789 | 0.505 |
| | 336 | 0.449 | 0.290 | 0.523 | 0.342 | 0.477 | 0.306 | 0.547 | 0.353 | 0.482 | 0.304 | 0.433 | 0.283 | 0.621 | 0.383 | 0.622 | 0.337 | 0.609 | 0.369 | 0.762 | 0.477 | 0.629 | 0.336 | 0.605 | 0.373 | 0.797 | 0.508 |
| | 720 | 0.477 | 0.303 | 0.559 | 0.354 | 0.509 | 0.323 | 0.554 | 0.359 | 0.514 | 0.322 | 0.467 | 0.302 | 0.626 | 0.382 | 0.660 | 0.408 | 0.647 | 0.387 | 0.719 | 0.449 | 0.640 | 0.350 | 0.645 | 0.394 | 0.841 | 0.523 |
| | Avg | 0.445 | 0.285 | 0.517 | 0.332 | 0.475 | 0.308 | 0.542 | 0.353 | 0.481 | 0.304 | 0.428 | 0.282 | 0.610 | 0.376 | 0.628 | 0.379 | 0.626 | 0.378 | 0.760 | 0.473 | 0.620 | 0.336 | 0.625 | 0.383 | 0.804 | 0.509 |
| 1st Count | | 16 | 17 | 0 | 4 | 6 | 5 | 0 | 0 | 2 | 1 | 7 | 7 | 1 | 0 | 0 | 0 | 0 | 4 | 0 | 0 | 1 | 0 | 1 | 0 | 0 | 0 |

Table 9: Full results of the **long-term forecasting tasks** with input length 384. We compare extensive competitive models under different prediction lengths. The input sequence length is set to 384 for all baselines. *Avg* means the average results from all four prediction lengths.

| Models | | DeformableTST (Ours) | | Pathformer [6] | | CARD [51] | | GPT4TS [58] | | PatchTST [35] | | iTransformer [22] | | FEDformer [57] | | Autoformer [47] | | RLinear [18] | | TiDE [9] | | TimesNet [46] | | DLinear [52] | | SCINet [20] | |
|---|---|---|---|---|---|---|---|---|---|---|---|---|---|---|---|---|---|---|---|---|---|---|---|---|---|---|---|
| Metric | | MSE | MAE | MSE | MAE | MSE | MAE | MSE | MAE | MSE | MAE | MSE | MAE | MSE | MAE | MSE | MAE | MSE | MAE | MSE | MAE | MSE | MAE | MSE | MAE | MSE | MAE |
| ETTh1 | 96 | **0.369** | **0.396** | 0.386 | 0.413 | 0.377 | 0.402 | 0.415 | 0.419 | 0.376 | 0.401 | 0.413 | 0.427 | 0.398 | 0.439 | 0.446 | 0.448 | 0.372 | 0.397 | 0.451 | 0.455 | 0.414 | 0.428 | 0.373 | 0.399 | 0.419 | 0.435 |
| | 192 | 0.410 | 0.417 | 0.413 | 0.425 | **0.404** | 0.418 | 0.447 | 0.447 | 0.410 | 0.421 | 0.449 | 0.450 | 0.454 | 0.471 | 0.425 | 0.448 | 0.411 | 0.421 | 0.482 | 0.473 | 0.469 | 0.467 | 0.407 | 0.421 | 0.471 | 0.469 |
| | 336 | 0.391 | 0.414 | 0.437 | 0.438 | 0.428 | 0.435 | 0.457 | 0.449 | 0.434 | 0.439 | 0.455 | 0.458 | 0.469 | 0.476 | 0.517 | 0.499 | 0.425 | 0.429 | 0.501 | 0.486 | 0.502 | 0.480 | 0.435 | 0.445 | 0.515 | 0.497 |
| | 720 | 0.447 | 0.464 | 0.463 | 0.472 | **0.437** | **0.459** | 0.471 | 0.462 | 0.451 | 0.471 | 0.528 | 0.521 | 0.630 | 0.559 | 0.557 | 0.536 | 0.450 | 0.465 | 0.510 | 0.509 | 0.645 | 0.542 | 0.479 | 0.501 | 0.529 | 0.515 |
| | Avg | **0.404** | **0.423** | 0.425 | 0.448 | 0.412 | 0.429 | 0.448 | 0.444 | 0.418 | 0.433 | 0.461 | 0.464 | 0.488 | 0.486 | 0.486 | 0.483 | 0.415 | 0.428 | 0.486 | 0.481 | 0.508 | 0.479 | 0.424 | 0.442 | 0.483 | 0.479 |
| ETTh2 | 96 | **0.272** | **0.334** | 0.283 | 0.350 | 0.275 | 0.336 | 0.319 | 0.371 | 0.275 | 0.337 | 0.306 | 0.363 | 0.426 | 0.469 | 0.395 | 0.444 | 0.278 | 0.338 | 0.370 | 0.431 | 0.347 | 0.395 | 0.305 | 0.368 | 0.327 | 0.383 |
| | 192 | **0.325** | **0.369** | 0.341 | 0.389 | 0.339 | 0.376 | 0.395 | 0.419 | 0.338 | 0.375 | 0.366 | 0.402 | 0.393 | 0.432 | 0.396 | 0.439 | 0.356 | 0.393 | 0.441 | 0.474 | 0.403 | 0.430 | 0.399 | 0.428 | 0.375 | 0.412 |
| | 336 | **0.319** | **0.373** | 0.382 | 0.421 | 0.360 | 0.397 | 0.414 | 0.439 | 0.352 | 0.391 | 0.400 | 0.427 | 0.390 | 0.436 | 0.397 | 0.441 | 0.364 | 0.408 | 0.469 | 0.499 | 0.441 | 0.457 | 0.471 | 0.476 | 0.438 | 0.453 |
| | 720 | 0.395 | 0.433 | 0.426 | 0.452 | **0.390** | **0.425** | 0.450 | 0.470 | 0.398 | 0.435 | 0.435 | 0.454 | 0.513 | 0.520 | 0.463 | 0.484 | 0.419 | 0.447 | 0.504 | 0.530 | 0.471 | 0.483 | 0.764 | 0.621 | 0.439 | 0.461 |
| | Avg | **0.328** | **0.377** | 0.358 | 0.403 | 0.341 | 0.384 | 0.394 | 0.425 | 0.341 | 0.385 | 0.377 | 0.412 | 0.431 | 0.464 | 0.413 | 0.452 | 0.354 | 0.397 | 0.446 | 0.484 | 0.416 | 0.441 | 0.485 | 0.473 | 0.395 | 0.427 |
| ETTm1 | 96 | **0.292** | **0.344** | 0.306 | 0.353 | 0.295 | 0.346 | 0.298 | 0.352 | **0.292** | **0.344** | 0.314 | 0.366 | 0.339 | 0.408 | 0.528 | 0.504 | 0.316 | 0.355 | 0.311 | 0.353 | 0.338 | 0.377 | 0.302 | 0.345 | 0.316 | 0.363 |
| | 192 | **0.335** | 0.371 | 0.337 | 0.378 | 0.339 | 0.371 | 0.345 | 0.379 | 0.340 | 0.374 | 0.353 | 0.389 | 0.397 | 0.442 | 0.583 | 0.534 | 0.337 | **0.366** | 0.344 | 0.372 | 0.363 | 0.386 | 0.337 | 0.368 | 0.360 | 0.390 |
| | 336 | **0.365** | 0.392 | 0.371 | 0.396 | 0.373 | 0.392 | 0.370 | 0.391 | 0.367 | 0.394 | 0.383 | 0.405 | 0.549 | 0.501 | 0.593 | 0.539 | 0.375 | 0.389 | 0.378 | 0.391 | 0.389 | 0.405 | 0.368 | **0.384** | 0.397 | 0.409 |
| | 720 | **0.417** | 0.418 | 0.437 | 0.433 | 0.420 | **0.414** | 0.428 | 0.426 | 0.422 | 0.427 | 0.437 | 0.435 | 0.470 | 0.472 | 0.618 | 0.553 | 0.426 | 0.415 | 0.433 | 0.422 | 0.514 | 0.465 | 0.422 | 0.418 | 0.453 | 0.439 |
| | Avg | **0.352** | 0.381 | 0.363 | 0.390 | 0.357 | 0.381 | 0.360 | 0.387 | 0.355 | 0.385 | 0.372 | 0.399 | 0.439 | 0.456 | 0.581 | 0.533 | 0.364 | 0.381 | 0.367 | 0.385 | 0.401 | 0.408 | 0.357 | **0.379** | 0.382 | 0.400 |
| ETTm2 | 96 | 0.170 | 0.257 | 0.172 | 0.261 | 0.165 | 0.256 | 0.179 | 0.268 | 0.166 | 0.255 | 0.179 | 0.271 | 0.277 | 0.349 | 0.285 | 0.361 | **0.164** | **0.253** | 0.189 | 0.292 | 0.217 | 0.290 | 0.165 | 0.260 | 0.182 | 0.275 |
| | 192 | **0.227** | **0.295** | 0.252 | 0.312 | 0.229 | 0.298 | 0.243 | 0.306 | 0.228 | 0.299 | 0.243 | 0.313 | 0.297 | 0.357 | 0.308 | 0.366 | 0.227 | 0.297 | 0.250 | 0.333 | 0.251 | 0.317 | 0.238 | 0.319 | 0.254 | 0.321 |
| | 336 | 0.278 | 0.327 | 0.285 | 0.335 | **0.274** | **0.328** | 0.315 | 0.352 | 0.280 | 0.331 | 0.295 | 0.345 | 0.349 | 0.386 | 0.347 | 0.387 | 0.275 | 0.329 | 0.309 | 0.372 | 0.312 | 0.358 | 0.309 | 0.370 | 0.298 | 0.353 |
| | 720 | **0.365** | 0.384 | 0.384 | 0.399 | 0.367 | **0.383** | 0.389 | 0.396 | 0.373 | 0.389 | 0.377 | 0.397 | 0.419 | 0.425 | 0.424 | 0.429 | 0.368 | 0.386 | 0.412 | 0.435 | 0.481 | 0.448 | 0.404 | 0.423 | 0.392 | 0.407 |
| | Avg | 0.260 | 0.316 | 0.273 | 0.327 | **0.259** | **0.316** | 0.282 | 0.331 | 0.262 | 0.319 | 0.273 | 0.332 | 0.336 | 0.379 | 0.341 | 0.386 | **0.259** | **0.316** | 0.290 | 0.358 | 0.315 | 0.353 | 0.279 | 0.343 | 0.282 | 0.339 |
| Weather | 96 | **0.149** | **0.198** | 0.154 | 0.205 | 0.155 | 0.209 | 0.164 | 0.215 | 0.150 | 0.201 | 0.160 | 0.210 | 0.245 | 0.304 | 0.267 | 0.322 | 0.174 | 0.227 | 0.176 | 0.228 | 0.161 | 0.216 | 0.172 | 0.232 | 0.161 | 0.218 |
| | 192 | **0.192** | **0.238** | 0.218 | 0.249 | 0.207 | 0.255 | 0.208 | 0.253 | 0.194 | 0.244 | 0.203 | 0.251 | 0.295 | 0.347 | 0.291 | 0.335 | 0.216 | 0.261 | 0.219 | 0.263 | 0.234 | 0.278 | 0.215 | 0.272 | 0.211 | 0.261 |
| | 336 | 0.244 | **0.278** | 0.263 | 0.283 | 0.263 | 0.294 | 0.256 | 0.289 | **0.243** | 0.279 | 0.252 | 0.287 | 0.338 | 0.377 | 0.322 | 0.354 | 0.263 | 0.294 | 0.266 | 0.298 | 0.286 | 0.316 | 0.264 | 0.317 | 0.259 | 0.296 |
| | 720 | **0.317** | **0.331** | 0.334 | 0.326 | 0.355 | 0.352 | 0.328 | 0.340 | 0.319 | 0.333 | 0.325 | 0.339 | 0.372 | 0.387 | 0.372 | 0.386 | 0.329 | 0.340 | 0.333 | 0.345 | 0.373 | 0.368 | 0.324 | 0.364 | 0.285 | 0.315 |
| | Avg | **0.225** | **0.261** | 0.242 | 0.266 | 0.245 | 0.277 | 0.239 | 0.274 | 0.226 | 0.264 | 0.235 | 0.272 | 0.313 | 0.354 | 0.313 | 0.349 | 0.246 | 0.281 | 0.249 | 0.284 | 0.264 | 0.295 | 0.244 | 0.296 | 0.229 | 0.272 |
| Solar-Energy | 96 | **0.173** | **0.241** | 0.215 | 0.286 | 0.175 | 0.243 | 0.225 | 0.296 | 0.197 | 0.291 | 0.187 | 0.246 | 0.328 | 0.425 | 0.946 | 0.696 | 0.223 | 0.297 | 0.228 | 0.300 | 0.217 | 0.311 | 0.216 | 0.316 | 0.206 | 0.306 |
| | 192 | **0.187** | **0.248** | 0.220 | 0.301 | 0.209 | 0.262 | 0.234 | 0.315 | 0.214 | 0.319 | 0.209 | 0.269 | 0.404 | 0.474 | 0.869 | 0.702 | 0.251 | 0.312 | 0.259 | 0.345 | 0.273 | 0.338 | 0.244 | 0.334 | 0.229 | 0.328 |
| | 336 | **0.193** | **0.260** | 0.232 | 0.311 | 0.209 | 0.272 | 0.236 | 0.318 | 0.235 | 0.328 | 0.222 | 0.281 | 0.431 | 0.498 | 0.834 | 0.694 | 0.265 | 0.338 | 0.273 | 0.353 | 0.266 | 0.368 | 0.259 | 0.346 | 0.235 | 0.325 |
| | 720 | **0.204** | **0.272** | 0.250 | 0.329 | 0.227 | 0.281 | 0.251 | 0.334 | 0.225 | 0.329 | 0.231 | 0.291 | 0.563 | 0.559 | 0.783 | 0.700 | 0.284 | 0.345 | 0.294 | 0.346 | 0.251 | 0.336 | 0.283 | 0.350 | 0.247 | 0.321 |
| | Avg | **0.189** | **0.255** | 0.229 | 0.307 | 0.205 | 0.265 | 0.237 | 0.316 | 0.218 | 0.317 | 0.212 | 0.272 | 0.432 | 0.489 | 0.858 | 0.698 | 0.256 | 0.323 | 0.264 | 0.336 | 0.252 | 0.338 | 0.251 | 0.337 | 0.229 | 0.320 |
| ECL | 96 | **0.132** | **0.231** | 0.137 | 0.232 | 0.133 | 0.233 | 0.137 | 0.232 | 0.133 | 0.232 | 0.140 | 0.240 | 0.199 | 0.318 | 0.137 | 0.246 | 0.151 | 0.251 | 0.161 | 0.266 | 0.192 | 0.300 | 0.151 | 0.250 | 0.162 | 0.269 |
| | 192 | 0.150 | 0.250 | 0.159 | 0.257 | 0.157 | 0.256 | 0.157 | 0.255 | **0.148** | **0.246** | 0.158 | 0.258 | 0.224 | 0.343 | 0.203 | 0.308 | 0.163 | 0.262 | 0.169 | 0.273 | 0.202 | 0.306 | 0.165 | 0.263 | 0.178 | 0.283 |
| | 336 | **0.167** | 0.267 | 0.169 | 0.268 | 0.171 | 0.269 | 0.169 | **0.265** | 0.175 | 0.275 | 0.249 | 0.364 | 0.236 | 0.338 | 0.179 | 0.277 | 0.181 | 0.284 | 0.262 | 0.350 | 0.180 | 0.280 | 0.188 | 0.294 |  |  |
| | 720 | 0.203 | 0.299 | 0.209 | 0.305 | **0.200** | **0.292** | 0.205 | 0.297 | 0.202 | 0.295 | 0.215 | 0.310 | 0.251 | 0.347 | 0.264 | 0.343 | 0.218 | 0.308 | 0.210 | 0.308 | 0.257 | 0.349 | 0.215 | 0.313 | 0.205 | 0.306 |
| | Avg | **0.163** | 0.262 | 0.170 | 0.266 | 0.165 | 0.262 | 0.168 | 0.263 | **0.163** | **0.260** | 0.172 | 0.271 | 0.231 | 0.347 | 0.205 | 0.309 | 0.178 | 0.275 | 0.180 | 0.283 | 0.228 | 0.326 | 0.178 | 0.277 | 0.183 | 0.288 |
| Traffic | 96 | **0.362** | **0.262** | 0.410 | 0.286 | 0.380 | 0.270 | 0.396 | 0.281 | 0.385 | 0.277 | 0.393 | 0.298 | 0.714 | 0.434 | 0.489 | 0.318 | 0.433 | 0.305 | 0.489 | 0.364 | 0.556 | 0.305 | 0.428 | 0.304 | 0.495 | 0.375 |
| | 192 | **0.385** | **0.268** | 0.413 | 0.291 | 0.398 | 0.277 | 0.411 | 0.289 | 0.401 | 0.283 | 0.414 | 0.312 | 0.625 | 0.420 | 0.811 | 0.471 | 0.441 | 0.307 | 0.493 | 0.370 | 0.557 | 0.304 | 0.437 | 0.308 | 0.509 | 0.386 |
| | 336 | **0.397** | **0.275** | 0.439 | 0.315 | 0.409 | 0.289 | 0.420 | 0.295 | 0.410 | 0.287 | 0.432 | 0.322 | 0.690 | 0.428 | 0.736 | 0.440 | 0.453 | 0.313 | 0.512 | 0.383 | 0.590 | 0.312 | 0.448 | 0.314 | 0.526 | 0.393 |
| | 720 | **0.434** | **0.298** | 0.464 | 0.333 | 0.437 | 0.301 | 0.451 | 0.312 | 0.443 | 0.304 | 0.472 | 0.347 | 0.755 | 0.455 | 0.900 | 0.523 | 0.483 | 0.331 | 0.539 | 0.404 | 0.617 | 0.323 | 0.480 | 0.333 | 0.565 | 0.411 |
| | Avg | **0.395** | **0.276** | 0.432 | 0.306 | 0.406 | 0.284 | 0.420 | 0.294 | 0.410 | 0.288 | 0.428 | 0.320 | 0.696 | 0.434 | 0.734 | 0.438 | 0.453 | 0.314 | 0.508 | 0.380 | 0.580 | 0.311 | 0.448 | 0.315 | 0.524 | 0.391 |
| 1st Count | | **24** | **22** | 0 | 0 | 5 | 5 | 0 | 0 | 3 | 3 | 0 | 0 | 0 | 0 | 0 | 0 | 2 | 2 | 0 | 0 | 0 | 0 | 0 | 1 | 0 | 0 |

Table 10: Full results of the **long-term forecasting tasks** with input length 768. We compare extensive competitive models under different prediction lengths. The input sequence length is set to 768 for all baselines. *Avg* means the average results from all four prediction lengths.

| Models | | DeformableTST (Ours) MSE | MAE | Pathformer [6] MSE | MAE | CARD [51] MSE | MAE | GPT4TS [58] MSE | MAE | PatchTST [35] MSE | MAE | iTransformer [22] MSE | MAE | FEDformer [57] MSE | MAE | Autoformer [47] MSE | MAE | RLinear [18] MSE | MAE | TiDE [9] MSE | MAE | TimesNet [46] MSE | MAE | DLinear [52] MSE | MAE | SCINet [20] MSE | MAE |
|---|---|---|---|---|---|---|---|---|---|---|---|---|---|---|---|---|---|---|---|---|---|---|---|---|---|---|---|
| ETTh1 | 96 | 0.367 | 0.404 | 0.389 | 0.417 | 0.377 | 0.410 | 0.454 | 0.449 | 0.379 | 0.410 | 0.401 | 0.428 | 0.481 | 0.486 | 0.374 | 0.436 | 0.372 | 0.400 | 0.446 | 0.457 | 0.510 | 0.483 | 0.372 | 0.401 | 0.427 | 0.447 |
| | 192 | 0.409 | 0.430 | 0.423 | 0.432 | 0.410 | 0.428 | 0.512 | 0.485 | 0.416 | 0.433 | 0.433 | 0.452 | 0.489 | 0.500 | 0.403 | 0.460 | 0.420 | 0.433 | 0.483 | 0.478 | 0.537 | 0.505 | 0.411 | 0.425 | 0.472 | 0.475 |
| | 336 | 0.406 | 0.437 | 0.449 | 0.450 | 0.430 | 0.443 | 0.525 | 0.489 | 0.440 | 0.451 | 0.475 | 0.481 | 0.647 | 0.563 | 0.516 | 0.514 | 0.438 | 0.441 | 0.508 | 0.496 | 0.691 | 0.561 | 0.438 | 0.450 | 0.525 | 0.509 |
| | 720 | 0.458 | 0.481 | 0.497 | 0.493 | 0.470 | 0.487 | 0.623 | 0.532 | 0.469 | 0.483 | 0.567 | 0.549 | 0.733 | 0.610 | 0.572 | 0.562 | 0.469 | 0.487 | 0.518 | 0.519 | 0.680 | 0.556 | 0.486 | 0.508 | 0.548 | 0.531 |
| | Avg | 0.410 | 0.438 | 0.440 | 0.448 | 0.422 | 0.442 | 0.529 | 0.489 | 0.426 | 0.444 | 0.469 | 0.478 | 0.588 | 0.540 | 0.466 | 0.493 | 0.425 | 0.440 | 0.489 | 0.488 | 0.605 | 0.526 | 0.427 | 0.446 | 0.493 | 0.491 |
| ETTh2 | 96 | 0.269 | 0.334 | 0.284 | 0.349 | 0.279 | 0.341 | 0.340 | 0.386 | 0.275 | 0.339 | 0.307 | 0.363 | 0.446 | 0.478 | 0.413 | 0.471 | 0.267 | 0.333 | 0.365 | 0.427 | 0.431 | 0.446 | 0.325 | 0.386 | 0.333 | 0.386 |
| | 192 | 0.325 | 0.372 | 0.346 | 0.393 | 0.350 | 0.384 | 0.420 | 0.429 | 0.336 | 0.378 | 0.441 | 0.442 | 0.449 | 0.480 | 0.459 | 0.503 | 0.332 | 0.378 | 0.436 | 0.471 | 0.444 | 0.471 | 0.432 | 0.450 | 0.375 | 0.411 |
| | 336 | 0.324 | 0.380 | 0.379 | 0.424 | 0.367 | 0.403 | 0.436 | 0.443 | 0.357 | 0.401 | 0.452 | 0.456 | 0.413 | 0.455 | 0.420 | 0.461 | 0.357 | 0.405 | 0.459 | 0.495 | 0.513 | 0.499 | 0.535 | 0.508 | 0.399 | 0.432 |
| | 720 | 0.412 | 0.446 | 0.422 | 0.456 | 0.392 | 0.430 | 0.470 | 0.476 | 0.397 | 0.436 | 0.434 | 0.462 | 0.511 | 0.528 | 0.500 | 0.516 | 0.424 | 0.454 | 0.494 | 0.529 | 0.520 | 0.494 | 0.947 | 0.689 | 0.433 | 0.460 |
| | Avg | 0.333 | 0.383 | 0.358 | 0.405 | 0.347 | 0.390 | 0.417 | 0.434 | 0.341 | 0.389 | 0.409 | 0.431 | 0.455 | 0.485 | 0.448 | 0.488 | 0.345 | 0.393 | 0.439 | 0.480 | 0.477 | 0.483 | 0.560 | 0.508 | 0.385 | 0.422 |
| ETTm1 | 96 | 0.291 | 0.347 | 0.301 | 0.353 | 0.296 | 0.343 | 0.306 | 0.359 | 0.295 | 0.351 | 0.316 | 0.370 | 0.444 | 0.465 | 0.466 | 0.463 | 0.312 | 0.355 | 0.317 | 0.360 | 0.378 | 0.397 | 0.307 | 0.351 | 0.334 | 0.377 |
| | 192 | 0.325 | 0.372 | 0.351 | 0.381 | 0.339 | 0.371 | 0.339 | 0.377 | 0.329 | 0.373 | 0.347 | 0.387 | 0.546 | 0.500 | 0.535 | 0.506 | 0.344 | 0.374 | 0.344 | 0.374 | 0.414 | 0.415 | 0.337 | 0.368 | 0.363 | 0.392 |
| | 336 | 0.359 | 0.390 | 0.382 | 0.403 | 0.368 | 0.388 | 0.370 | 0.395 | 0.364 | 0.395 | 0.388 | 0.412 | 0.471 | 0.478 | 0.614 | 0.548 | 0.366 | 0.387 | 0.373 | 0.391 | 0.443 | 0.429 | 0.364 | 0.384 | 0.392 | 0.409 |
| | 720 | 0.418 | 0.423 | 0.443 | 0.438 | 0.417 | 0.420 | 0.432 | 0.429 | 0.420 | 0.432 | 0.460 | 0.456 | 0.465 | 0.474 | 0.616 | 0.557 | 0.414 | 0.413 | 0.422 | 0.418 | 0.510 | 0.474 | 0.413 | 0.414 | 0.453 | 0.443 |
| | Avg | 0.348 | 0.383 | 0.369 | 0.394 | 0.355 | 0.381 | 0.362 | 0.390 | 0.352 | 0.388 | 0.378 | 0.406 | 0.482 | 0.479 | 0.558 | 0.519 | 0.359 | 0.382 | 0.364 | 0.386 | 0.436 | 0.429 | 0.355 | 0.379 | 0.386 | 0.405 |
| ETTm2 | 96 | 0.169 | 0.258 | 0.171 | 0.262 | 0.164 | 0.255 | 0.181 | 0.274 | 0.182 | 0.272 | 0.185 | 0.278 | 0.305 | 0.362 | 0.331 | 0.384 | 0.161 | 0.252 | 0.186 | 0.291 | 0.244 | 0.314 | 0.161 | 0.253 | 0.186 | 0.278 |
| | 192 | 0.229 | 0.299 | 0.239 | 0.312 | 0.226 | 0.298 | 0.249 | 0.311 | 0.247 | 0.314 | 0.239 | 0.317 | 0.344 | 0.386 | 0.351 | 0.397 | 0.227 | 0.299 | 0.245 | 0.333 | 0.287 | 0.342 | 0.222 | 0.302 | 0.245 | 0.322 |
| | 336 | 0.280 | 0.333 | 0.293 | 0.349 | 0.272 | 0.328 | 0.326 | 0.346 | 0.303 | 0.354 | 0.298 | 0.355 | 0.385 | 0.418 | 0.383 | 0.422 | 0.276 | 0.330 | 0.304 | 0.373 | 0.326 | 0.370 | 0.296 | 0.361 | 0.302 | 0.358 |
| | 720 | 0.349 | 0.384 | 0.387 | 0.409 | 0.390 | 0.399 | 0.397 | 0.392 | 0.371 | 0.398 | 0.401 | 0.416 | 0.420 | 0.437 | 0.426 | 0.440 | 0.353 | 0.386 | 0.392 | 0.430 | 0.450 | 0.442 | 0.400 | 0.425 | 0.386 | 0.410 |
| | Avg | 0.257 | 0.319 | 0.273 | 0.333 | 0.263 | 0.320 | 0.288 | 0.331 | 0.276 | 0.335 | 0.281 | 0.341 | 0.364 | 0.401 | 0.373 | 0.411 | 0.254 | 0.317 | 0.282 | 0.357 | 0.327 | 0.367 | 0.270 | 0.335 | 0.280 | 0.342 |
| Weather | 96 | 0.146 | 0.198 | 0.149 | 0.200 | 0.152 | 0.208 | 0.166 | 0.220 | 0.148 | 0.202 | 0.165 | 0.217 | 0.293 | 0.345 | 0.297 | 0.349 | 0.170 | 0.224 | 0.171 | 0.225 | 0.173 | 0.228 | 0.167 | 0.226 | 0.171 | 0.234 |
| | 192 | 0.191 | 0.239 | 0.197 | 0.242 | 0.227 | 0.269 | 0.210 | 0.257 | 0.194 | 0.241 | 0.205 | 0.253 | 0.306 | 0.350 | 0.313 | 0.354 | 0.213 | 0.259 | 0.214 | 0.261 | 0.258 | 0.296 | 0.211 | 0.267 | 0.211 | 0.262 |
| | 336 | 0.241 | 0.280 | 0.247 | 0.286 | 0.262 | 0.298 | 0.255 | 0.291 | 0.243 | 0.282 | 0.259 | 0.296 | 0.343 | 0.373 | 0.336 | 0.368 | 0.257 | 0.292 | 0.258 | 0.295 | 0.311 | 0.334 | 0.256 | 0.306 | 0.266 | 0.304 |
| | 720 | 0.310 | 0.331 | 0.315 | 0.336 | 0.362 | 0.356 | 0.322 | 0.338 | 0.310 | 0.328 | 0.321 | 0.342 | 0.369 | 0.392 | 0.374 | 0.392 | 0.320 | 0.337 | 0.321 | 0.340 | 0.416 | 0.404 | 0.315 | 0.353 | 0.322 | 0.345 |
| | Avg | 0.222 | 0.262 | 0.227 | 0.266 | 0.251 | 0.283 | 0.238 | 0.277 | 0.224 | 0.263 | 0.238 | 0.277 | 0.328 | 0.365 | 0.330 | 0.366 | 0.240 | 0.278 | 0.241 | 0.280 | 0.290 | 0.316 | 0.237 | 0.288 | 0.243 | 0.286 |
| Solar-Energy | 96 | 0.165 | 0.238 | 0.192 | 0.263 | 0.177 | 0.246 | 0.223 | 0.289 | 0.191 | 0.273 | 0.174 | 0.242 | 0.327 | 0.431 | 0.878 | 0.686 | 0.191 | 0.268 | 0.197 | 0.272 | 0.219 | 0.279 | 0.190 | 0.273 | 0.181 | 0.277 |
| | 192 | 0.184 | 0.254 | 0.198 | 0.270 | 0.210 | 0.268 | 0.238 | 0.314 | 0.211 | 0.282 | 0.194 | 0.259 | 0.313 | 0.410 | 0.836 | 0.674 | 0.209 | 0.277 | 0.216 | 0.293 | 0.228 | 0.297 | 0.211 | 0.291 | 0.201 | 0.288 |
| | 336 | 0.191 | 0.263 | 0.226 | 0.283 | 0.219 | 0.277 | 0.243 | 0.316 | 0.221 | 0.293 | 0.207 | 0.272 | 0.613 | 0.606 | 0.901 | 0.731 | 0.224 | 0.285 | 0.233 | 0.317 | 0.239 | 0.312 | 0.227 | 0.303 | 0.227 | 0.312 |
| | 720 | 0.199 | 0.262 | 0.220 | 0.270 | 0.233 | 0.294 | 0.275 | 0.343 | 0.235 | 0.297 | 0.217 | 0.279 | 0.579 | 0.591 | 0.809 | 0.710 | 0.228 | 0.288 | 0.237 | 0.320 | 0.238 | 0.313 | 0.234 | 0.316 | 0.234 | 0.316 |
| | Avg | 0.185 | 0.254 | 0.209 | 0.272 | 0.210 | 0.271 | 0.245 | 0.316 | 0.215 | 0.286 | 0.198 | 0.263 | 0.458 | 0.510 | 0.856 | 0.700 | 0.213 | 0.280 | 0.221 | 0.301 | 0.231 | 0.300 | 0.216 | 0.296 | 0.211 | 0.298 |
| ECL | 96 | 0.132 | 0.234 | 0.134 | 0.236 | 0.133 | 0.230 | 0.134 | 0.229 | 0.130 | 0.235 | 0.142 | 0.241 | 0.219 | 0.336 | 0.141 | 0.248 | 0.142 | 0.242 | 0.143 | 0.250 | 0.203 | 0.310 | 0.145 | 0.243 | 0.154 | 0.261 |
| | 192 | 0.148 | 0.248 | 0.156 | 0.256 | 0.157 | 0.256 | 0.153 | 0.250 | 0.153 | 0.249 | 0.160 | 0.258 | 0.229 | 0.347 | 0.240 | 0.331 | 0.156 | 0.254 | 0.154 | 0.258 | 0.205 | 0.313 | 0.159 | 0.257 | 0.167 | 0.274 |
| | 336 | 0.165 | 0.266 | 0.179 | 0.271 | 0.181 | 0.283 | 0.176 | 0.272 | 0.168 | 0.269 | 0.179 | 0.277 | 0.243 | 0.359 | 0.242 | 0.334 | 0.172 | 0.269 | 0.176 | 0.275 | 0.228 | 0.331 | 0.174 | 0.274 | 0.177 | 0.285 |
| | 720 | 0.197 | 0.296 | 0.209 | 0.307 | 0.207 | 0.303 | 0.206 | 0.296 | 0.205 | 0.298 | 0.221 | 0.312 | 0.250 | 0.363 | 0.269 | 0.351 | 0.212 | 0.301 | 0.213 | 0.303 | 0.241 | 0.342 | 0.208 | 0.307 | 0.202 | 0.306 |
| | Avg | 0.161 | 0.261 | 0.170 | 0.268 | 0.170 | 0.268 | 0.167 | 0.262 | 0.164 | 0.263 | 0.176 | 0.272 | 0.235 | 0.351 | 0.223 | 0.316 | 0.170 | 0.267 | 0.171 | 0.272 | 0.219 | 0.324 | 0.171 | 0.270 | 0.175 | 0.282 |
| Traffic | 96 | 0.355 | 0.261 | 0.389 | 0.277 | 0.369 | 0.267 | 0.371 | 0.265 | 0.373 | 0.267 | 0.380 | 0.291 | 0.782 | 0.470 | 0.606 | 0.385 | 0.402 | 0.285 | 0.451 | 0.344 | 0.565 | 0.303 | 0.400 | 0.287 | 0.456 | 0.352 |
| | 192 | 0.380 | 0.271 | 0.396 | 0.283 | 0.381 | 0.270 | 0.387 | 0.271 | 0.384 | 0.269 | 0.400 | 0.306 | 0.608 | 0.391 | 0.861 | 0.501 | 0.411 | 0.289 | 0.459 | 0.346 | 0.579 | 0.308 | 0.411 | 0.291 | 0.469 | 0.358 |
| | 336 | 0.393 | 0.281 | 0.417 | 0.307 | 0.402 | 0.283 | 0.398 | 0.279 | 0.399 | 0.275 | 0.420 | 0.317 | 0.624 | 0.396 | 0.905 | 0.529 | 0.425 | 0.296 | 0.470 | 0.353 | 0.573 | 0.304 | 0.425 | 0.298 | 0.483 | 0.365 |
| | 720 | 0.434 | 0.300 | 0.447 | 0.319 | 0.437 | 0.299 | 0.436 | 0.303 | 0.439 | 0.295 | 0.466 | 0.344 | 0.985 | 0.579 | 0.883 | 0.513 | 0.464 | 0.317 | 0.491 | 0.362 | 0.601 | 0.318 | 0.465 | 0.322 | 0.526 | 0.386 |
| | Avg | 0.391 | 0.278 | 0.412 | 0.297 | 0.397 | 0.280 | 0.398 | 0.280 | 0.399 | 0.277 | 0.417 | 0.315 | 0.750 | 0.459 | 0.814 | 0.482 | 0.426 | 0.297 | 0.468 | 0.351 | 0.579 | 0.308 | 0.425 | 0.300 | 0.484 | 0.365 |
| 1st Count | | 25 | 19 | 0 | 0 | 3 | 5 | 0 | 1 | 2 | 1 | 0 | 0 | 0 | 0 | 0 | 0 | 2 | 4 | 0 | 0 | 0 | 0 | 2 | 3 | 0 | 0 |

Table 11: Full results of the **multivariate short-term forecasting tasks**. We compare extensive competitive models under different prediction lengths. The input length is 2 times of the prediction length. *Avg* means the average results from all three prediction lengths.

| Models | | DeformableTST (Ours) | | Pathformer [6] | | CARD [51] | | GPT4TS [58] | | PatchTST [35] | | iTransformer [22] | | FEDformer [57] | | Autoformer [47] | | RLinear [18] | | TiDE [9] | | TimesNet [46] | | DLinear [52] | | SCINet [20] | |
|---|---|---|---|---|---|---|---|---|---|---|---|---|---|---|---|---|---|---|---|---|---|---|---|---|---|---|---|
| Metric | | MSE | MAE | MSE | MAE | MSE | MAE | MSE | MAE | MSE | MAE | MSE | MAE | MSE | MAE | MSE | MAE | MSE | MAE | MSE | MAE | MSE | MAE | MSE | MAE | MSE | MAE |
| ETTh1 | 6 | 0.507 | 0.428 | 0.608 | 0.481 | 0.728 | 0.521 | 0.670 | 0.495 | 0.759 | 0.535 | 0.589 | 0.472 | 0.622 | 0.500 | 0.609 | 0.494 | 1.049 | 0.636 | 1.090 | 0.663 | 0.606 | 0.479 | 0.870 | 0.624 | 0.592 | 0.482 |
| | 12 | 0.306 | 0.358 | 0.386 | 0.399 | 0.377 | 0.393 | 0.382 | 0.395 | 0.454 | 0.437 | 0.371 | 0.393 | 0.381 | 0.410 | 0.434 | 0.436 | 0.611 | 0.505 | 0.523 | 0.469 | 0.349 | 0.383 | 0.486 | 0.461 | 0.418 | 0.434 |
| | 18 | 0.305 | 0.358 | 0.373 | 0.392 | 0.365 | 0.386 | 0.372 | 0.390 | 0.440 | 0.430 | 0.346 | 0.372 | 0.401 | 0.425 | 0.439 | 0.446 | 0.570 | 0.489 | 0.483 | 0.453 | 0.366 | 0.396 | 0.463 | 0.446 | 0.422 | 0.435 |
| | Avg | 0.373 | 0.381 | 0.456 | 0.424 | 0.490 | 0.433 | 0.475 | 0.427 | 0.551 | 0.467 | 0.435 | 0.412 | 0.468 | 0.445 | 0.494 | 0.459 | 0.743 | 0.543 | 0.699 | 0.528 | 0.440 | 0.419 | 0.606 | 0.510 | 0.477 | 0.450 |
| ETTh2 | 6 | 0.132 | 0.230 | 0.145 | 0.246 | 0.150 | 0.252 | 0.152 | 0.254 | 0.156 | 0.259 | 0.151 | 0.253 | 0.149 | 0.262 | 0.139 | 0.257 | 0.203 | 0.302 | 0.180 | 0.286 | 0.154 | 0.254 | 0.194 | 0.326 | 0.176 | 0.301 |
| | 12 | 0.139 | 0.238 | 0.160 | 0.259 | 0.167 | 0.266 | 0.166 | 0.266 | 0.172 | 0.270 | 0.159 | 0.257 | 0.172 | 0.275 | 0.172 | 0.277 | 0.202 | 0.295 | 0.194 | 0.289 | 0.169 | 0.265 | 0.204 | 0.320 | 0.198 | 0.323 |
| | 18 | 0.155 | 0.250 | 0.171 | 0.265 | 0.172 | 0.268 | 0.173 | 0.268 | 0.189 | 0.279 | 0.181 | 0.275 | 0.206 | 0.299 | 0.210 | 0.301 | 0.228 | 0.311 | 0.202 | 0.293 | 0.189 | 0.280 | 0.207 | 0.312 | 0.258 | 0.371 |
| | Avg | 0.142 | 0.239 | 0.159 | 0.257 | 0.163 | 0.262 | 0.164 | 0.263 | 0.172 | 0.269 | 0.164 | 0.262 | 0.176 | 0.279 | 0.174 | 0.278 | 0.211 | 0.303 | 0.192 | 0.289 | 0.171 | 0.266 | 0.202 | 0.319 | 0.211 | 0.332 |
| ILI | 6 | 1.338 | 0.671 | 2.421 | 0.955 | 2.930 | 1.110 | 2.390 | 0.965 | 3.349 | 1.197 | 1.973 | 0.851 | 3.188 | 1.172 | 2.922 | 1.128 | 3.744 | 1.252 | 4.333 | 1.349 | 1.440 | 0.694 | 3.359 | 1.236 | 5.186 | 1.495 |
| | 12 | 2.397 | 0.924 | 3.289 | 1.221 | 4.842 | 1.591 | 4.141 | 1.399 | 4.465 | 1.513 | 2.761 | 1.064 | 4.863 | 1.650 | 5.557 | 1.741 | 5.915 | 1.739 | 6.737 | 1.886 | 2.830 | 1.022 | 5.450 | 1.732 | 4.659 | 1.509 |
| | 18 | 1.567 | 0.800 | 3.680 | 1.305 | 3.843 | 1.413 | 3.565 | 1.317 | 4.837 | 1.594 | 2.797 | 1.112 | 5.171 | 1.705 | 5.020 | 1.700 | 6.128 | 1.843 | 5.262 | 1.709 | 3.363 | 1.073 | 4.976 | 1.689 | 5.350 | 1.674 |
| | Avg | 1.767 | 0.798 | 3.130 | 1.160 | 3.872 | 1.371 | 3.365 | 1.227 | 4.217 | 1.435 | 2.510 | 1.009 | 4.407 | 1.509 | 4.500 | 1.523 | 5.262 | 1.611 | 5.444 | 1.648 | 2.544 | 0.930 | 4.595 | 1.552 | 5.065 | 1.559 |
| Exchange | 6 | 0.008 | 0.051 | 0.009 | 0.061 | 0.010 | 0.062 | 0.008 | 0.056 | 0.009 | 0.064 | 0.008 | 0.053 | 0.018 | 0.098 | 0.016 | 0.091 | 0.013 | 0.076 | 0.012 | 0.071 | 0.009 | 0.060 | 0.049 | 0.166 | 0.043 | 0.156 |
| | 12 | 0.013 | 0.072 | 0.016 | 0.081 | 0.016 | 0.086 | 0.015 | 0.079 | 0.016 | 0.087 | 0.014 | 0.075 | 0.026 | 0.114 | 0.027 | 0.120 | 0.022 | 0.103 | 0.018 | 0.091 | 0.016 | 0.082 | 0.034 | 0.138 | 0.056 | 0.175 |
| | 18 | 0.018 | 0.087 | 0.019 | 0.089 | 0.022 | 0.101 | 0.021 | 0.096 | 0.023 | 0.105 | 0.020 | 0.092 | 0.040 | 0.146 | 0.036 | 0.139 | 0.030 | 0.121 | 0.028 | 0.116 | 0.023 | 0.101 | 0.041 | 0.156 | 0.069 | 0.188 |
| | Avg | 0.013 | 0.070 | 0.015 | 0.077 | 0.016 | 0.083 | 0.015 | 0.077 | 0.016 | 0.085 | 0.014 | 0.073 | 0.028 | 0.119 | 0.026 | 0.117 | 0.022 | 0.100 | 0.019 | 0.093 | 0.016 | 0.081 | 0.041 | 0.153 | 0.056 | 0.173 |
| PEMS03 | 6 | 0.072 | 0.179 | 0.090 | 0.193 | 0.074 | 0.181 | 0.086 | 0.193 | 0.085 | 0.198 | 0.063 | 0.171 | 0.078 | 0.189 | 0.079 | 0.191 | 0.096 | 0.208 | 0.108 | 0.224 | 0.079 | 0.185 | 0.100 | 0.218 | 0.069 | 0.179 |
| | 12 | 0.100 | 0.208 | 0.122 | 0.223 | 0.101 | 0.210 | 0.123 | 0.231 | 0.131 | 0.247 | 0.089 | 0.198 | 0.110 | 0.230 | 0.147 | 0.280 | 0.147 | 0.255 | 0.167 | 0.276 | 0.114 | 0.224 | 0.131 | 0.255 | 0.105 | 0.225 |
| | 18 | 0.133 | 0.240 | 0.173 | 0.266 | 0.138 | 0.251 | 0.179 | 0.279 | 0.188 | 0.294 | 0.127 | 0.235 | 0.147 | 0.271 | 0.173 | 0.307 | 0.219 | 0.310 | 0.241 | 0.328 | 0.156 | 0.261 | 0.176 | 0.296 | 0.137 | 0.247 |
| | Avg | 0.102 | 0.209 | 0.128 | 0.227 | 0.104 | 0.214 | 0.129 | 0.234 | 0.135 | 0.246 | 0.093 | 0.201 | 0.112 | 0.230 | 0.133 | 0.259 | 0.154 | 0.258 | 0.172 | 0.276 | 0.116 | 0.223 | 0.136 | 0.256 | 0.104 | 0.217 |
| PEMS04 | 6 | 0.085 | 0.192 | 0.107 | 0.208 | 0.089 | 0.194 | 0.098 | 0.208 | 0.097 | 0.210 | 0.079 | 0.183 | 0.100 | 0.215 | 0.100 | 0.216 | 0.123 | 0.245 | 0.135 | 0.258 | 0.098 | 0.206 | 0.120 | 0.246 | 0.079 | 0.184 |
| | 12 | 0.113 | 0.222 | 0.144 | 0.243 | 0.118 | 0.227 | 0.136 | 0.246 | 0.144 | 0.261 | 0.111 | 0.219 | 0.134 | 0.257 | 0.160 | 0.291 | 0.178 | 0.289 | 0.208 | 0.319 | 0.138 | 0.250 | 0.169 | 0.298 | 0.106 | 0.221 |
| | 18 | 0.150 | 0.254 | 0.192 | 0.283 | 0.163 | 0.269 | 0.193 | 0.293 | 0.207 | 0.318 | 0.153 | 0.259 | 0.178 | 0.300 | 0.252 | 0.376 | 0.253 | 0.344 | 0.290 | 0.375 | 0.197 | 0.300 | 0.195 | 0.322 | 0.146 | 0.247 |
| | Avg | 0.116 | 0.223 | 0.148 | 0.245 | 0.123 | 0.230 | 0.142 | 0.249 | 0.149 | 0.263 | 0.114 | 0.220 | 0.137 | 0.257 | 0.171 | 0.294 | 0.185 | 0.293 | 0.211 | 0.317 | 0.144 | 0.252 | 0.161 | 0.289 | 0.110 | 0.217 |
| PEMS07 | 6 | 0.063 | 0.162 | 0.085 | 0.186 | 0.062 | 0.165 | 0.077 | 0.184 | 0.077 | 0.188 | 0.059 | 0.159 | 0.072 | 0.181 | 0.076 | 0.190 | 0.090 | 0.205 | 0.102 | 0.221 | 0.076 | 0.182 | 0.092 | 0.213 | 0.066 | 0.169 |
| | 12 | 0.087 | 0.191 | 0.120 | 0.222 | 0.087 | 0.195 | 0.114 | 0.225 | 0.122 | 0.243 | 0.082 | 0.188 | 0.101 | 0.219 | 0.118 | 0.247 | 0.140 | 0.250 | 0.160 | 0.270 | 0.118 | 0.230 | 0.122 | 0.250 | 0.087 | 0.200 |
| | 18 | 0.121 | 0.223 | 0.174 | 0.267 | 0.127 | 0.233 | 0.174 | 0.276 | 0.187 | 0.303 | 0.122 | 0.230 | 0.135 | 0.261 | 0.150 | 0.279 | 0.212 | 0.309 | 0.241 | 0.332 | 0.168 | 0.275 | 0.169 | 0.294 | 0.131 | 0.243 |
| | Avg | 0.090 | 0.192 | 0.126 | 0.225 | 0.092 | 0.198 | 0.122 | 0.228 | 0.129 | 0.245 | 0.088 | 0.192 | 0.103 | 0.220 | 0.115 | 0.239 | 0.147 | 0.255 | 0.168 | 0.274 | 0.121 | 0.229 | 0.128 | 0.252 | 0.095 | 0.204 |
| PEMS08 | 6 | 0.078 | 0.180 | 0.104 | 0.207 | 0.083 | 0.188 | 0.091 | 0.203 | 0.092 | 0.207 | 0.080 | 0.187 | 0.097 | 0.208 | 0.097 | 0.209 | 0.109 | 0.228 | 0.125 | 0.247 | 0.092 | 0.198 | 0.116 | 0.242 | 0.100 | 0.200 |
| | 12 | 0.104 | 0.208 | 0.140 | 0.238 | 0.113 | 0.217 | 0.132 | 0.240 | 0.142 | 0.259 | 0.110 | 0.217 | 0.132 | 0.247 | 0.165 | 0.289 | 0.171 | 0.280 | 0.187 | 0.300 | 0.135 | 0.242 | 0.164 | 0.293 | 0.126 | 0.238 |
| | 18 | 0.136 | 0.237 | 0.191 | 0.277 | 0.147 | 0.253 | 0.189 | 0.288 | 0.206 | 0.314 | 0.145 | 0.250 | 0.179 | 0.290 | 0.239 | 0.349 | 0.245 | 0.337 | 0.287 | 0.340 | 0.185 | 0.286 | 0.193 | 0.314 | 0.143 | 0.249 |
| | Avg | 0.106 | 0.208 | 0.145 | 0.241 | 0.114 | 0.219 | 0.137 | 0.244 | 0.147 | 0.260 | 0.112 | 0.218 | 0.136 | 0.248 | 0.167 | 0.282 | 0.175 | 0.282 | 0.200 | 0.296 | 0.137 | 0.242 | 0.158 | 0.283 | 0.123 | 0.229 |
| 1st Count | | 16 | 16 | 0 | 0 | 0 | 1 | 1 | 0 | 0 | 0 | 7 | 7 | 0 | 0 | 0 | 0 | 0 | 0 | 0 | 0 | 0 | 0 | 0 | 0 | 4 | 1 |

Table 12: Full results for the **univariate short-term forecasting** tasks in M4 dataset. We report SMAPE, MASE, OWA for M4 datasets as metrics. Lower metrics indicate better performance. *Wighted Average* means the results are wighted averaged from several M4 subdatasets under different sample intervals. ∗. in the Transformers indicates the name of ∗former. The original paper of N-BEATS [37] adopts a special ensemble method to promote the performance. For fair comparisons, we remove the ensemble and only compare the pure forecasting models.

| Models | | DeformableTST (Ours) | Path. [6] | CARD [51] | GPT4TS [58] | Cross. [53] | PatchTST [35] | iTransformer [22] | FED. [57] | Auto. [47] | RLinear [18] | TiDE [9] | TimesNet [46] | DLinear [52] | SCINet [20] | N-HiTS [4] | N-BEATS [37] |
|---|---|---|---|---|---|---|---|---|---|---|---|---|---|---|---|---|---|
| Yearly | SMAPE | 13.194 | 13.473 | 13.302 | 13.538 | 13.392 | 13.445 | 13.461 | 13.728 | 13.974 | 16.151 | 17.019 | 13.387 | 16.965 | 13.764 | 13.418 | 13.436 |
| | MASE | 2.955 | 3.005 | 3.016 | 3.041 | 3.001 | 3.021 | 3.045 | 3.048 | 3.134 | 3.680 | 3.945 | 2.996 | 4.283 | 3.103 | 3.045 | 3.043 |
| | OWA | 0.775 | 0.790 | 0.786 | 0.797 | 0.787 | 0.791 | 0.795 | 0.803 | 0.822 | 0.957 | 1.017 | 0.786 | 1.058 | 0.811 | 0.793 | 0.794 |
| Quarterly | SMAPE | 9.971 | 10.233 | 10.031 | 10.325 | 16.317 | 10.187 | 10.071 | 10.792 | 11.338 | 11.741 | 12.164 | 10.100 | 12.145 | 10.946 | 10.202 | 10.124 |
| | MASE | 1.163 | 1.203 | 1.176 | 1.218 | 2.197 | 1.196 | 1.182 | 1.283 | 1.365 | 1.456 | 1.510 | 1.182 | 1.520 | 1.293 | 1.194 | 1.169 |
| | OWA | 0.877 | 0.903 | 0.884 | 0.913 | 1.542 | 0.898 | 0.888 | 0.958 | 1.012 | 1.064 | 1.103 | 0.890 | 1.106 | 0.969 | 0.899 | 0.886 |
| Monthly | SMAPE | 12.592 | 12.895 | 12.670 | 12.860 | 12.924 | 12.856 | 12.737 | 14.260 | 13.958 | 13.599 | 13.616 | 12.670 | 13.514 | 13.541 | 12.791 | 12.677 |
| | MASE | 0.931 | 0.955 | 0.933 | 0.951 | 0.966 | 0.956 | 0.935 | 1.102 | 1.103 | 1.056 | 1.056 | 0.933 | 1.037 | 1.024 | 0.969 | 0.937 |
| | OWA | 0.874 | 0.896 | 0.878 | 0.893 | 0.902 | 0.895 | 0.881 | 1.012 | 1.002 | 0.968 | 0.968 | 0.878 | 0.956 | 0.951 | 0.899 | 0.880 |
| Others | SMAPE | 4.324 | 5.136 | 5.330 | 4.861 | 5.511 | 4.877 | 5.033 | 4.954 | 5.485 | 6.747 | 6.825 | 4.891 | 6.709 | 8.138 | 5.061 | 4.925 |
| | MASE | 2.993 | 3.427 | 3.261 | 3.320 | 3.733 | 3.280 | 3.284 | 3.264 | 3.865 | 4.652 | 4.809 | 3.302 | 4.953 | 4.997 | 3.216 | 3.391 |
| | OWA | 0.927 | 1.081 | 1.075 | 1.035 | 1.168 | 1.030 | 1.047 | 1.036 | 1.187 | 1.443 | 1.477 | 1.035 | 1.487 | 1.644 | 1.040 | 1.053 |
| Weighted Average | SMAPE | 11.688 | 12.001 | 11.815 | 12.008 | 13.475 | 11.952 | 11.878 | 12.840 | 12.909 | 13.398 | 13.711 | 11.829 | 13.639 | 12.699 | 11.927 | 11.851 |
| | MASE | 1.555 | 1.610 | 1.587 | 1.614 | 1.868 | 1.604 | 1.597 | 1.701 | 1.771 | 1.935 | 2.017 | 1.585 | 2.095 | 1.765 | 1.613 | 1.599 |
| | OWA | 0.838 | 0.863 | 0.850 | 0.865 | 0.985 | 0.860 | 0.855 | 0.918 | 0.939 | 1.000 | 1.033 | 0.851 | 1.051 | 0.930 | 0.861 | 0.855 |

Table 13: Full results for the **univariate short-term forecasting** tasks in other datasets. We report the SMAPE in this Table as metric and a lower metric indicates better performance. ∗. in the Transformers indicates the name of ∗former. The original paper of N-BEATS [37] adopts a special ensemble method to promote the performance. For fair comparisons, we remove the ensemble and only compare the pure forecasting models.

| Models | | DeformableTST (Ours) | Path. [6] | CARD [51] | GPT4TS [58] | Cross. [53] | PatchTST [35] | iTransformer [22] | FED. [57] | Auto. [47] | RLinear [18] | TiDE [9] | TimesNet [46] | DLinear [52] | SCINet [20] | N-HiTS [4] | N-BEATS [37] |
|---|---|---|---|---|---|---|---|---|---|---|---|---|---|---|---|---|---|
| M1 | Yearly | **15.902** | 20.305 | 21.769 | 21.208 | 26.310 | 22.408 | 29.190 | 21.718 | 27.242 | 21.491 | 25.671 | 16.023 | 27.472 | 21.145 | 27.404 | 21.021 |
| | Quarterly | **14.232** | 16.955 | 15.666 | 16.782 | 15.806 | 16.338 | 16.288 | 20.823 | 17.249 | 30.849 | 29.427 | 22.875 | 25.456 | 32.920 | 27.035 | 17.089 |
| | Monthly | **15.616** | 18.793 | 16.569 | 21.322 | 18.049 | 17.241 | 19.401 | 17.626 | 18.708 | 30.788 | 26.259 | 18.480 | 20.337 | 26.665 | 21.960 | 21.320 |
| M3 | Yearly | **15.315** | 24.509 | 20.867 | 21.846 | 18.488 | 18.623 | 22.998 | 17.300 | 17.386 | 55.073 | 20.301 | 23.989 | 24.359 | 33.380 | 23.544 | 17.204 |
| | Quarterly | **7.365** | 10.933 | 8.052 | 12.579 | 8.069 | 7.991 | 8.642 | 11.751 | 11.503 | 27.079 | 18.749 | 11.649 | 16.963 | 11.576 | 18.907 | 12.189 |
| | Monthly | **14.928** | 18.193 | 21.103 | 22.804 | 18.278 | 16.562 | 18.060 | 25.910 | 19.309 | 30.241 | 26.082 | 21.350 | 25.341 | 25.883 | 23.808 | 19.809 |
| | Other | **9.378** | 17.353 | 15.500 | 19.099 | 12.100 | 15.707 | 18.439 | 20.989 | 22.525 | 24.769 | 18.370 | 18.091 | 15.690 | 26.064 | 17.216 | 22.090 |
| Tourism | Quarterly | **17.968** | 27.921 | 19.182 | 31.216 | 19.517 | 19.314 | 20.116 | 38.606 | 39.887 | 39.079 | 48.609 | 28.052 | 34.630 | 35.208 | 33.909 | 43.887 |
| | Monthly | **25.037** | 34.769 | 27.515 | 38.819 | 27.718 | 27.570 | 28.743 | 35.983 | 31.698 | 40.432 | 40.399 | 38.505 | 40.095 | 40.269 | 37.959 | 41.778 |
| NN5 | Weekly | **14.372** | 18.186 | 22.491 | 16.012 | 17.672 | 17.717 | 20.449 | 17.072 | 19.650 | 22.868 | 23.424 | 22.355 | 23.781 | 26.486 | 16.645 | 23.282 |
| Hospital | Monthly | **19.088** | 21.551 | 21.392 | 22.587 | 20.907 | 22.123 | 20.785 | 28.646 | 25.749 | 26.184 | 29.103 | 21.182 | 23.015 | 28.807 | 25.309 | 23.594 |
| KDD Cup | Hourly | **50.694** | 56.740 | 61.653 | 54.713 | 58.472 | 59.449 | 60.615 | 59.013 | 57.617 | 62.922 | 63.447 | 56.618 | 59.523 | 63.358 | 54.855 | 62.305 |

# O  Showcases

To provide an intuitive comparison among different models, we provide showcases to the long-term forecasting tasks under two representative cases (the time series is in declining stage and the time series is in rising stage). The results are in Figure 12-13. Among the various models, our DeformableTST predicts the most precise future series variations and exhibits superior performance.

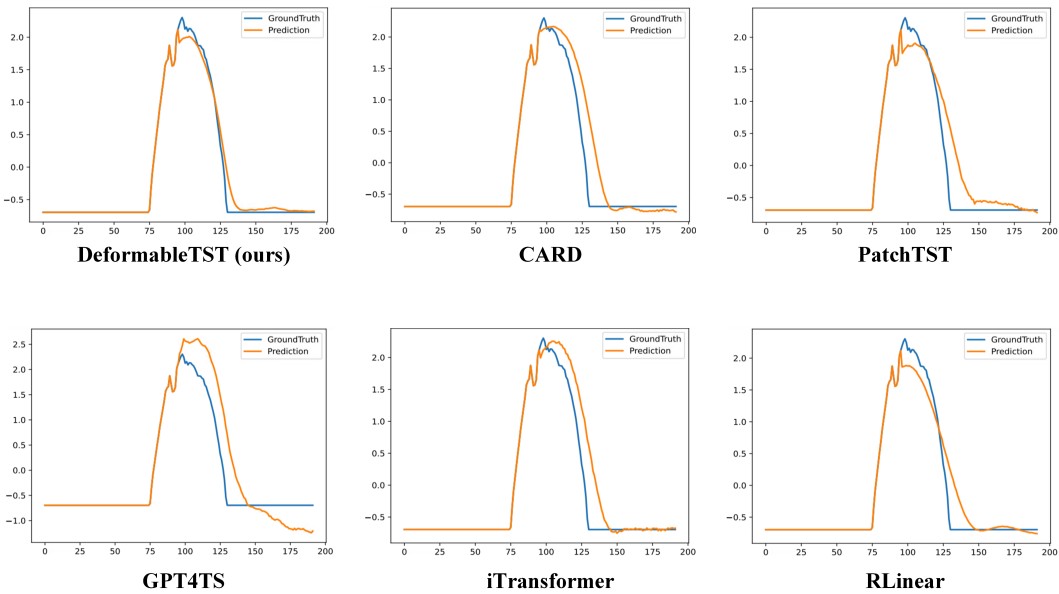

Figure 12: Visualization of input-96-predict-96 results on the Solar dataset. The time series is in declining stage. The blue lines stand for the ground truth and the orange lines stand for predicted values.

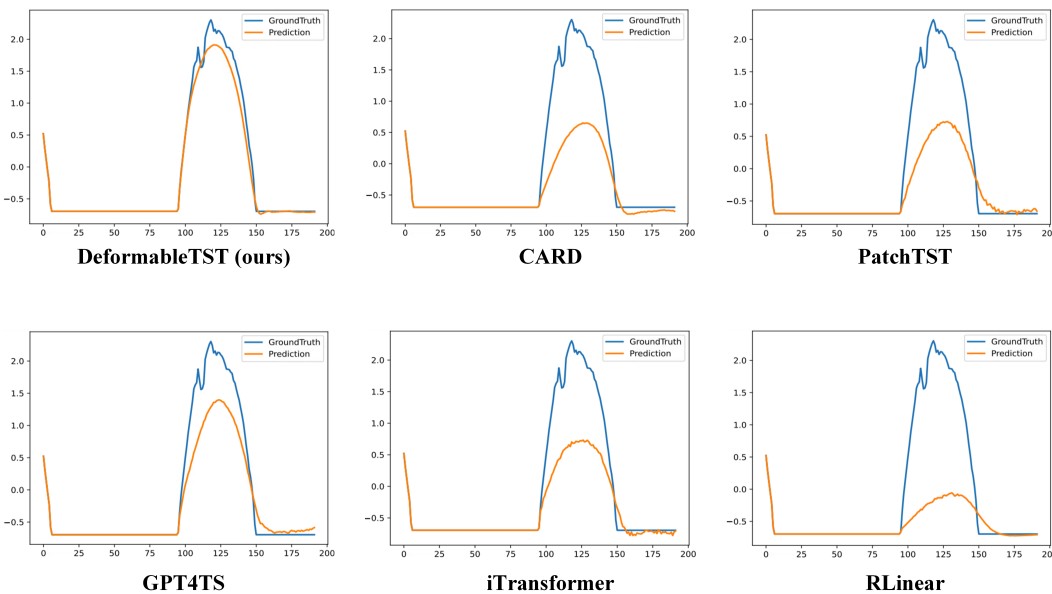

Figure 13: Visualization of input-96-predict-96 results on the Solar dataset. The time series is in rising stage. The blue lines stand for the ground truth and the orange lines stand for predicted values.

# P   Comparison with More Baselines and Experiments on More Datasets

## P.1   Compared with Sageformer

We compare our model with Sageformer [54], the latest Graph-Transformer model. Since Sageformer will produce NaN outputs in some short-term forecasting tasks, we mainly conduct comparisons in long-term forecasting. The results are shown in Table 14. Our DeformableTST achieves consistently better performance than the latest Graph-Transformer method, further demonstrating our performance superiority.

Table 14: Comparison with Sageformer in long-term forecasting tasks. A lower MSE or MAE indicates a better performance. Results are averaged from three input lengths $I \in \{96, 384, 768\}$ and four prediction lengths $T \in \{96, 192, 336, 720\}$. The best results are in **bold**. Full results of Sageformer [54] are provided in Table 15.

| Dataset | ETTh1 | | ETTh2 | | ETTm1 | | ETTm2 | | Weather | | Solar | | ECL | | Traffic | |
|---|---|---|---|---|---|---|---|---|---|---|---|---|---|---|---|---|
| Metric | MSE | MAE | MSE | MAE | MSE | MAE | MSE | MAE | MSE | MAE | MSE | MAE | MSE | MAE | MSE | MAE |
| **DeformableTST (Ours)** | **0.413** | **0.430** | **0.336** | **0.381** | **0.358** | **0.386** | **0.267** | **0.321** | **0.233** | **0.266** | **0.199** | **0.255** | **0.169** | **0.267** | **0.410** | **0.280** |
| Sageformer [54] | 0.427 | 0.438 | 0.368 | 0.405 | 0.371 | 0.394 | 0.275 | 0.327 | 0.238 | 0.272 | 0.227 | 0.285 | 0.174 | 0.273 | 0.418 | 0.287 |

## P.2   Experiments on Stock Market Dataset

We conduct multivariate short-term forecasting experiments on Stock Market Dataset [36]. As shown in Table 16, our DeformableTST still outperforms other competitors, validating that DeformableTST can work on stock market data.

# Q   Experiments on Synthetic Dataset

We conduct experiments on synthetic dataset with some typical cases of attention distributions to prove our model can handle both uniform and clustered attention distribution. The details and results are provided in Figure 14.

As shown in Figure 14, our method can accurately predict the future data in all cases. And ERFs can operate as anticipated, successfully matching the distributions of key information. In details, in the case of globally uniform attention, the brighter points in ERF are also distributed globally, which means the model can find the important time points across the whole series. In other cases, the brighter points in ERF tend to concentrate in localized areas of key information, proving the effectiveness of our method in scenarios where key information is clustered within specific time window. These results validate that our method can adeptly manage both uniform and clustered attention distributions.

# R   Model Robustness to Patching

We conduct ablation study to show the effect of patching on our method (under input-384 and input-768 settings). As shown in Figure 6 and Figure 16, our model is robust to the choice of different patch sizes on input length 384 and input length 768.

Meanwhile, as shown in Figure 15, the performance of other Patch-based Transformer competitors (e.g., PatchTST [35] and CARD [51]) will decrease obviously and fell out of the good rankings if without patching. This is a significant performance decrease, especially considering the intense competition in time series forecasting. By contrast, our method works well without patching, which further verifies our robustness to the use of patching and shows that our model can successfully get rid of the over-reliance of patching.

Table 15: Full results of Sageformer [54] in long-term forecasting tasks. *Avg* means the average results from all four prediction lengths.

| Models | | Sageformer (Input-96) | | Sageformer (Input-384) | | Sageformer (Input-768) | |
|---|---|---|---|---|---|---|---|
| Metric | | MSE | MAE | MSE | MAE | MSE | MAE |
| ETTh1 | 96 | 0.385 | 0.402 | 0.371 | 0.401 | 0.377 | 0.412 |
| | 192 | 0.431 | 0.428 | 0.418 | 0.429 | 0.413 | 0.429 |
| | 336 | 0.458 | 0.445 | 0.428 | 0.436 | 0.433 | 0.444 |
| | 720 | 0.476 | 0.469 | 0.461 | 0.470 | 0.473 | 0.485 |
| | Avg | 0.438 | 0.436 | 0.420 | 0.434 | 0.424 | 0.443 |
| ETTh2 | 96 | 0.300 | 0.346 | 0.284 | 0.343 | 0.289 | 0.347 |
| | 192 | 0.391 | 0.408 | 0.347 | 0.384 | 0.354 | 0.393 |
| | 336 | 0.418 | 0.426 | 0.372 | 0.419 | 0.386 | 0.430 |
| | 720 | 0.428 | 0.448 | 0.418 | 0.443 | 0.425 | 0.471 |
| | Avg | 0.384 | 0.407 | 0.355 | 0.397 | 0.364 | 0.410 |
| ETTm1 | 96 | 0.336 | 0.368 | 0.294 | 0.345 | 0.300 | 0.353 |
| | 192 | 0.376 | 0.395 | 0.339 | 0.375 | 0.337 | 0.374 |
| | 336 | 0.399 | 0.410 | 0.381 | 0.400 | 0.369 | 0.391 |
| | 720 | 0.464 | 0.447 | 0.434 | 0.436 | 0.418 | 0.432 |
| | Avg | 0.394 | 0.405 | 0.362 | 0.389 | 0.356 | 0.388 |
| ETTm2 | 96 | 0.177 | 0.259 | 0.172 | 0.260 | 0.170 | 0.255 |
| | 192 | 0.247 | 0.304 | 0.241 | 0.307 | 0.232 | 0.304 |
| | 336 | 0.308 | 0.346 | 0.284 | 0.336 | 0.308 | 0.352 |
| | 720 | 0.412 | 0.406 | 0.379 | 0.398 | 0.374 | 0.391 |
| | Avg | 0.286 | 0.329 | 0.269 | 0.325 | 0.271 | 0.326 |
| Weather | 96 | 0.163 | 0.207 | 0.153 | 0.207 | 0.149 | 0.198 |
| | 192 | 0.222 | 0.258 | 0.196 | 0.242 | 0.197 | 0.248 |
| | 336 | 0.272 | 0.296 | 0.247 | 0.285 | 0.253 | 0.293 |
| | 720 | 0.347 | 0.347 | 0.326 | 0.341 | 0.323 | 0.340 |
| | Avg | 0.251 | 0.277 | 0.231 | 0.269 | 0.231 | 0.270 |
| Solar-Energy | 96 | 0.231 | 0.286 | 0.190 | 0.254 | 0.179 | 0.243 |
| | 192 | 0.265 | 0.305 | 0.208 | 0.269 | 0.195 | 0.277 |
| | 336 | 0.288 | 0.313 | 0.213 | 0.286 | 0.212 | 0.287 |
| | 720 | 0.292 | 0.327 | 0.226 | 0.286 | 0.223 | 0.281 |
| | Avg | 0.269 | 0.308 | 0.209 | 0.274 | 0.202 | 0.272 |
| ECL | 96 | 0.156 | 0.251 | 0.141 | 0.244 | 0.138 | 0.243 |
| | 192 | 0.171 | 0.263 | 0.158 | 0.259 | 0.154 | 0.254 |
| | 336 | 0.188 | 0.285 | 0.174 | 0.275 | 0.171 | 0.273 |
| | 720 | 0.226 | 0.317 | 0.215 | 0.309 | 0.200 | 0.300 |
| | Avg | 0.185 | 0.279 | 0.172 | 0.272 | 0.166 | 0.268 |
| Traffic | 96 | 0.418 | 0.271 | 0.385 | 0.275 | 0.371 | 0.268 |
| | 192 | 0.434 | 0.281 | 0.397 | 0.279 | 0.385 | 0.273 |
| | 336 | 0.446 | 0.289 | 0.414 | 0.295 | 0.399 | 0.278 |
| | 720 | 0.480 | 0.305 | 0.443 | 0.308 | 0.442 | 0.320 |
| | Avg | 0.445 | 0.287 | 0.410 | 0.289 | 0.399 | 0.285 |

Table 16: Multivariate short-term forecasting results on Stock Market. We compare extensive competitive models under different prediction lengths. The input length is 2 times of the prediction length. *Avg* means the average results from all three prediction lengths. The best results are in **bold**.

| Models | | DeformableTST (Ours) | | Pathformer [6] | | CARD [51] | | GPT4TS [58] | | PatchTST [35] | | iTransformer [22] | | FEDformer [57] | | Autoformer [47] | | RLinear [18] | | TiDE [9] | | TimesNet [46] | |
|---|---|---|---|---|---|---|---|---|---|---|---|---|---|---|---|---|---|---|---|---|---|---|---|
| Metric | | MSE | MAE | MSE | MAE | MSE | MAE | MSE | MAE | MSE | MAE | MSE | MAE | MSE | MAE | MSE | MAE | MSE | MAE | MSE | MAE | MSE | MAE |
| Stock | 6 | **0.110** | **0.132** | 0.127 | 0.143 | 0.121 | 0.138 | 0.121 | 0.142 | 0.127 | 0.152 | 0.121 | 0.139 | 0.132 | 0.178 | 0.126 | 0.159 | 0.157 | 0.168 | 0.149 | 0.167 | 0.121 | 0.138 |
| | 12 | **0.121** | **0.152** | 0.130 | 0.157 | 0.134 | 0.161 | 0.144 | 0.166 | 0.138 | 0.174 | 0.137 | 0.162 | 0.143 | 0.189 | 0.143 | 0.186 | 0.161 | 0.187 | 0.164 | 0.192 | 0.136 | 0.163 |
| | 18 | **0.136** | **0.170** | 0.146 | 0.178 | 0.143 | 0.177 | 0.155 | 0.179 | 0.146 | 0.186 | 0.145 | 0.175 | 0.157 | 0.217 | 0.152 | 0.208 | 0.171 | 0.208 | 0.167 | 0.200 | 0.141 | 0.173 |
| | Avg | **0.122** | **0.151** | 0.134 | 0.159 | 0.133 | 0.159 | 0.140 | 0.162 | 0.137 | 0.171 | 0.134 | 0.159 | 0.144 | 0.195 | 0.140 | 0.184 | 0.163 | 0.188 | 0.160 | 0.186 | 0.133 | 0.158 |

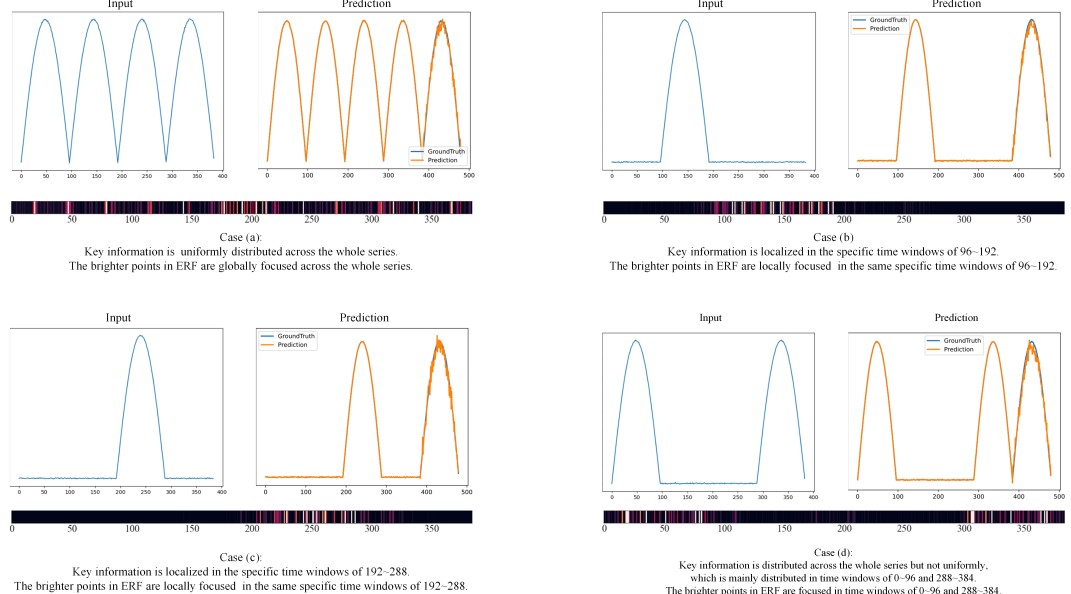

Figure 14: We conduct experiments with some typical cases of attention distributions. The synthetic data and experiment setups are as follows. **Setup (1) for globally uniform attention as shown in Case (a)**: The input consists of 4 semi-sinusoidal signals with noise. The task is to predict 1 semi-sinusoidal signal. Thus, the future data evenly relates to the historical data. The length of each signal is 96. So this is an input-384-predict-96 task. **Setup (2) for clustered attention as shown in Case (b) and (c)**: Simlilar to setup (1), but in the length-384 input, only 1 semi-sinusoidal signal is remained while others are masked as 0. Thus, the future data is related only to the local window of remained signal. Masks can be constant or varying across samples to simulate scenarios that the localized areas are constant or varying across samples. **Setup (3) for global but not uniform attention as shown in Case (d)**: This setup is similar to setup (2) but more semi-sinusoidal signals are remained, resulting in global attention distribution but not uniform.

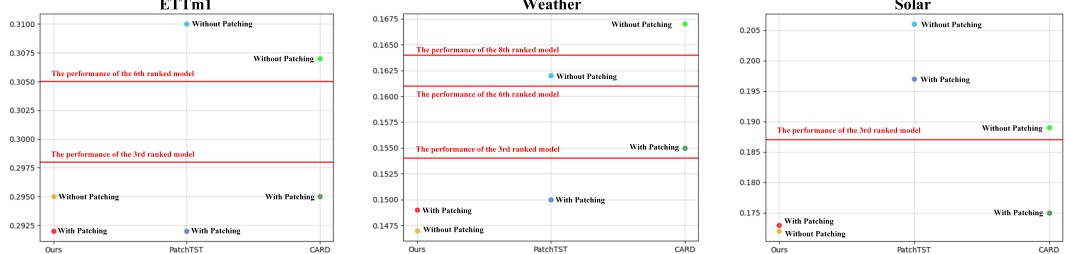

Figure 15: The impact of patching on latest patch-based Transformer forecasters (PatchTST and CARD). After the removal of patching, the performance of PatchTST and CARD will decrease obviously and fell out of the good rankings, while our DeformableTST is robust to patching and maintains the consistent excellent performance, consistently ranked in the top-3. We conduct experiments under input-384-predict-96 settings. The rankings on each dataset are calculated from Table 9 of Appendix N.

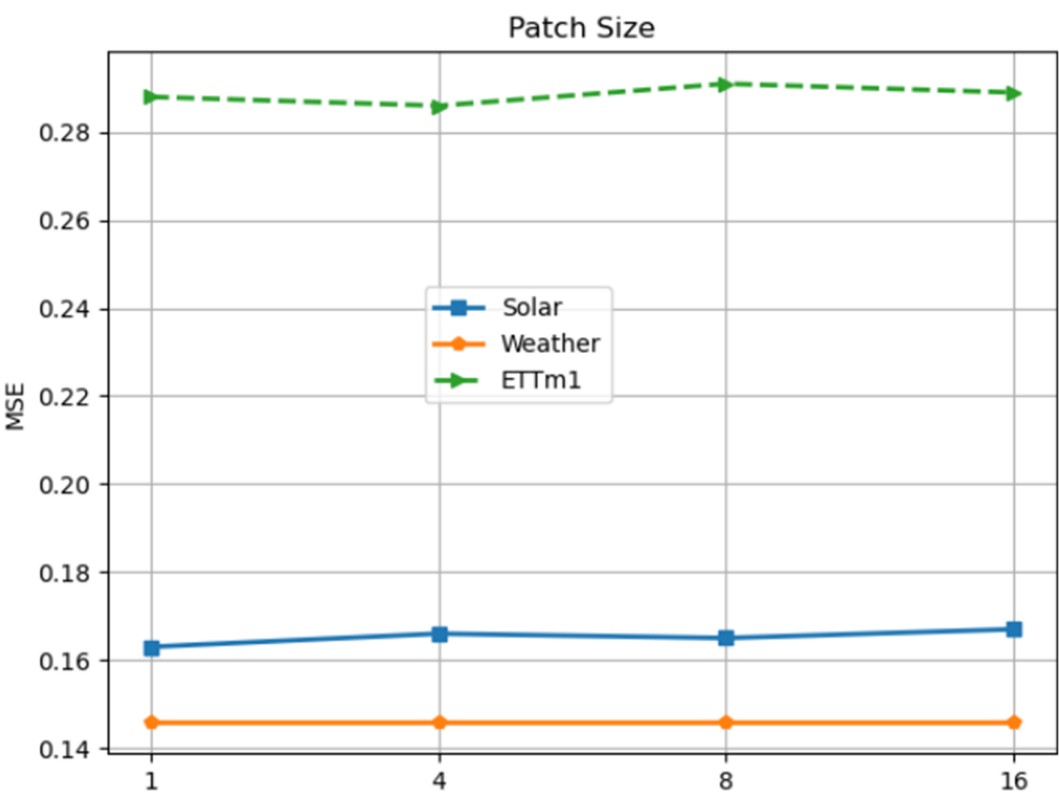

Figure 16: Robustness of our DeformableTST to patch size under input-768 settings.

