# OpenReview forum: "DeformableTST: Transformer for Time Series Forecasting without Over-reliance on Patching"
_NeurIPS.cc/2024/Conference — NeurIPS 2024 poster_

### Official Review · Reviewer_Qoam · 2024-07-08

**Soundness:** 3
**Presentation:** 3
**Contribution:** 3
**Rating:** 7
**Confidence:** 5

**Summary:**

The paper highlights the limitations of current time series forecasting models based on patching techniques. Existing models rely on patching to handle long sequences, but this approach is not suitable for all forecasting tasks. To overcome these limitations, DeformableTST introduces a Deformable Attention mechanism that dynamically selects important time points. This mechanism uses learnable offsets to sample significant time points from the input sequence and calculates attention only on these selected points, thereby reducing computational costs and improving performance.

**Strengths:**

1. The paper proposes a method to address the limitations of current advancements that rely on patching techniques, which seems like a reasonable problem definition.
2. The proposed deformable attention mechanism is an advanced form of sparse attention that reduces computational costs by selecting only significant time points using learnable offsets. This is particularly advantageous for handling long sequences.
3. By dynamically selecting important information based on the characteristics of the data rather than relying on patching techniques, the model can be flexibly applied to various time series forecasting tasks.

**Weaknesses:**

New attention mechanisms have been proposed extensively from the past to the present. This paper seems to lack novelty in that regard.

**Questions:**

1. For highly complex time series data, Deformable Attention alone may not be sufficient, and additional preprocessing or postprocessing might be necessary. I am curious about how the authors plan to address this.
2. While the paper is reasonable and convincing, it appears somewhat lacking in terms of novelty.

**Limitations:**

1. For highly complex time series data, Deformable Attention alone may not be sufficient, and additional preprocessing or postprocessing might be necessary. I am curious about how the authors plan to address this.
2. While the paper is reasonable and convincing, it appears somewhat lacking in terms of novelty.

---

> ### Author Rebuttal · Authors · 2024-08-07
>
> Many thanks to Reviewer Qoam for providing thorough insightful comments.
>
> > W1 & Q2: Concerns about the novelty. While the paper is reasonable and convincing, it appears somewhat lacking in terms of novelty, since new attention mechanism has been proposed from past to the presently.
>
> We'd like to highlight our novelty in the following points:
> + **Our novelty is more than designing a new attention mechanism**:
>     + In this paper, we expose, analyze and solve the problem of over-reliance on patching in latest Transformer-based forecasters, **which is of great significance to the time series community.** And this significant problem is **less-explored in previous works.**
>     + Our proposed methods **is the first to** deeply explore and specially target to this novel research problem, **which highlights our novelty and contribution**.
>     + Meanwhile, our findings can **bring new perspective** and prompt people to rethink the relationship between attention and patching, which benefits the future work to design more powerful Transformer-based forecasters with wider applicability.
>     + Therefore, **in addition to proposing a state-of-the-art method for time series forecasting, the exposure and exploration of this novel problem is also a main novelty of our paper.**
>
> + Although existing many new attention mechanisms, **our proposed method still shows great distinctions from others**:
>     + **Different from latest Transformer-based models**: Most of the latest Transformer-based models highly rely on patching and thus have limited applicability.
>     Different from them, our DeformableTST can successfully reduce the reliance on patching and broaden the applicability of Transformer-based models. Specifically, it can flexibly adapt to multiple input lengths and achieve excellent performance in tasks unsuitable for patching,
> **which is a great improvement than previous Transformer-based models and further highlights our novelty and contribution**.
>     + **Different from prior-based sparse attentions**: As a novel data-driven sparse attention, our method differs from the previous prior-based sparse attentions by its excellent flexibility.
>     Due to the diverse pattern in different time series, the priors used in previous sparse attentions are hard to match all kinds of inputs, resulting in their inferior performance.
>     By contrast, our method can learn from the input time series dynamically, therefore is more flexible to the diverse property in different time series, leading to better performance.
>
> + The exploration on more new attentions is of values:
>     + In our humble opinions, since the vanilla full attention is deficient in time series forecasting,
> exploration on more new attentions is of great significance. In details, we propose DeformableTST as a **novel methods for time series forecasting with better performance and wider applicability**, which is of great practical values in a wider range of real-world applications.
>     + We hope above clarifications help to distinguish our method from previous works and clarify our contributions, thus highlighting our novelty.
>
>
> > Q1: Additional preprocessing or postprocessing for highly complex time series data.
> + Preprocessing method : Using an additive seasonal-trend decomposition method to decompose time series into long-term trend, seasonal, and residual components.
> The effectiveness of this preprocessing is verified in some latest LLM for time series models through the compelling performance improvement[1][2].
> And different from early decomposition method using moving average[3][4], these latest decomposition methods are based on STL decomposition[5].
>
> + Postprocessing: Using ensemble. In the most representative forecasting competitions like M1, M3 and M4, most participants employ ensemble methods to enhance performance[6].
>
> + Postprocessing: Using Refinement. According to [7], we can use diffusion models to refine the prediction results and bring better performance in highly complex time series data. And [7] also proves that this postprocessing method is computationally efficient.
>
> > Reference
>
> [1] Defu Cao, et al. "TEMPO: Prompt-based Generative Pre-trained Transformer for Time Series Forecasting."
>
> [2] Zijie Pan, et al. "S2IP-LLM: Semantic Space Informed Prompt Learning with LLM for Time Series Forecasting."
>
> [3] Haixu Wu, et al. "Autoformer: Decomposition Transformers with Auto-Correlation for Long-Term Series Forecasting."
>
> [4] Ailing Zeng, et al. "Are Transformers Effective for Time Series Forecasting?"
>
> [5] Robert Cleveland, et al. "Stl: A seasonal-trend decomposition."
>
> [6] Boris Oreshkin, et al. "N-BEATS: Neural basis expansion analysis for interpretable time series forecasting."
>
> [7] Marcel Kollovieh, et al. "Predict, Refine, Synthesize: Self-Guiding Diffusion Models for Probabilistic Time Series Forecasting."

---

> > ### Comment · Reviewer_Qoam · 2024-08-12
> >
> > Thanks for your response, I am updating my score.

---

> > > ### Author Response · Authors · 2024-08-12
> > > **Thanks for Your Response and Raising the Score**
> > >
> > > We would like to thank Reviewer Qoam again for providing the insightful pre-rebuttal review and valuable feedback,
> > > which help us a lot in the rebuttal and paper revision.
> > >
> > > And we would also like to thank you for raising the score and recommending our paper!

---

### Official Review · Reviewer_reZp · 2024-07-10

**Soundness:** 3
**Presentation:** 3
**Contribution:** 3
**Rating:** 7
**Confidence:** 4

**Summary:**

The paper presents a novel approach for time series forecasting that relies less on patching. The authors incorporate deformable attention capturing important temporal information and a hierarchical structure that reduces memory consumption. The authors verify the effectiveness of their framework across multiple benchmarks: short-term, long-term, univariate, and multivariate.
The paper demonstrates state-of-the-art forecasting performance, reducing the dependence on patching.

**Strengths:**

1. The paper is well-written, making it easy to read and understand.

2. Authors provide good motivation for their design choices, as well as reasoning and ablations of the different hyperparameters for their method, e.g., hierarchical structure, patching and the number of important time points.

3. The paper conducts comprehensive experiments in both short-term forecasting and long-term forecasting settings. The proposed method outperforms in the majority of the cases.

**Weaknesses:**

I don't see any major weaknesses for this paper. However, adding adjustments and justifications to some of the claims (see questions below) would make the paper more convincing.

**Questions:**

1. In the introduction it is stated that:
> Patched-based Transformers have to work with a very long input length and a very large patch size to achieve ideal performance.

Can you please cite the works that support this claim?

> PatchTST proposes that attention mechanism can work better in temporal modeling with the help of large size patching technique.

PatchTST suggests that patching can improve the long-term forecasting accuracy by using patch sizes between {8, 16} for input length of 336. The patch lengths used in your work are 4 and 8 and are referred to as small sized patches (under the input-768 settings and the input-384 settings).  Do you consider patch sizes of 8 and 16 to be large?

> In such condition, the advanced Transformer-based models suffer from severe performance degradation due to the lack of patching, limiting their applicability to a wider range of forecasting tasks.

Does this statement refer to patch based Transformer models? Can you please cite the works that support this claim?

2. The main text uses the term "important points" often and yet its meaning (e.g., time point in the similar changing stage, the inflexion point, the extremal point and so on) is mentioned in the appendix. Can you add an explanation for the term in the main text (figure 2 or figure 3 can be used for illustration)?

3. One of the main claims of this work (including the title) is that Transformer-based models are too reliant on patching.
Some of the latest Transformer-based models adopt the use of patching to reduce memory usage and improve performance. These methods leverage  (or taylor their methods to harness) the advantages of patching rather than relying on it. Will you be willing to change this claim?

4. Have you conducted an ablation study showing the effect of patching (on input lengths 384 and 768) on you method? These result can strengthen the claim of the paper.

5.  > In conclusion, full attention must work with patching to achieve ideal performance for it highly relies on the guidance of patching to focus on important time points.

To my understanding, this conclusion is based on the results of figure 1. The effective receptive field offers a nice visualization but not convincing enough to support the claim above. How did you validate the importance of the brighter points? Does having less important points is necessarily a good objective?

**Limitations:**

The authors adequately addressed the limitations.

---

> ### Author Rebuttal · Authors · 2024-08-07
>
> Many thanks to Reviewer reZp for the detailed and insightful review.
>
> > Q1: Supports for our claims
>
> Due to the page limitation, please refer to **Global Response**.
>
> > Q2: Add an explanation for "important points" in the main text.
> + We will explain the concept of "important points" in Section 1 before using it and label different types of "important points" in Figure 2 (b) of main text for illustration.
> + Please refer to **Q5 1** for the detailed explanation.
>
> > Q3: About the claim like "rely on".
>
> We are willing to revise our paper into a better shape. But we hope we can have more discussions about this claim.
>
> In my humble opinion, both "leverage the advantages of patching" and "rely on patching" tend to emphasize that patching plays an important role in patch-based Transformers, but "rely on" goes deeper. And followings are the reasons why we use a deeper word like "rely on":
> + It's a fact that patching has become a **must-have** technique for most latest Transformer-based models.
> + The impact of patching on performance may be greater than we suppose.  As shown in **Figure 2 in global response PDF**, if without patching,  the performance of PatchTST and CARD will decrease obviously and fell out of the good rankings. This is a significant performance decrease, especially considering the intense competition in time series forecasting. **Considering the great impact of patching on the performance in patch-based Transformers, we use "rely on" to emphasize it**.
>
> > Q4: Ablation study showing the effect of patching (on input lengths 384 and 768) on you method.
> + As shown in **Figure 6 of Appendix D**, our model is robust to patch sizes on input length 384.
> + As shown in **Figure 2 of PDF in global response**, our method works well without patching, while other competitors suffer from obvious performance decrease.
> + As shown in **Figure 3 of PDF in global response**, our model is robust to patch sizes on input length 768.
>
> > Q5 1: How to validate the importance of the brighter points.
> + We are sorry that some concepts may be misleading and sincerely thanks for your reminder.
> + We provide explanations of "important time points" and "focused time points" to avoid confusion between these two concepts:
>     + **The important time points refer to the time points that make contribution to a better performance**. Based on the findings in [1][2][3][4], the time points in time series are very redundant or even noisy. Thus, only a small number of points are needed to represent the properties of the time series and make contribution to better performance. These points are considered to be important and are mainly the time points that reflect the property of time series, such as time point in the similar changing stage, the inflexion point, the extremal point and so on.
>     + **The brighter points here means the focused time points by the models**. It just means that the model tends to focus on these time points when extracting temporal representation. The focused time points don't equal to the important time points in all cases.  Only when the focused time points are exactly the important time points, the model can focus on important information and learn useful representation to achieve a better performance.
> + In conclusion, we validate the importance of a time points based on the performance. And a brighter point only means this point is focused by model when extracting temporal representation. It doesn't always mean a higher importance.
> + **About the conclusion in Figure 1**:
>     + In left figure where almost all time points are brighter, the model focuses on all time points when extracting temporal representations but leads to a worse performance (MSE 0.385). Since the time points are very redundant or even noisy, focusing on the trivial part of them will influence the predictions. Thus, we state that "attention has not learned to distinguish the importance of each time point, leading to trivial representation" in line 47-49.
>     + In right figure where few time points are brighter, the model focuses on some selected time points and achieve better performance (MSE 0.367) after patching. Thus we can suppose that the model focuses on the important time points. And the pattern of ERF is also divided by patches, so we state that "attention focus on a small number of important time points based on the patch partition" in line 49-50.
>     + Above comparision of two figures proves our conclusion that "attention highly relies on the guidance of patching to focus on important time points" in line 51-52.
> + To avoid misleading, we will clarify above concepts in Section 1 and re-introduce Figure 1 with more detailed description.
>
> > Q5 2: Does having less important points is necessarily a good objective?
> + Based on our above explanation, we suppose this question should be "Does having less focused points is necessarily a good objective", because important time points are the inherent properties of time series, while the focused time points are the characteristic of the model that we can determine.
> + In our opinion, having less focused points is a good objective.
> The time points in time series are very redundant or even noisy. And it is inappropriate to focus on all of them because focusing on the trivial and noisy part of them will influence the predictions.
> + But too few focused points is not appropriated. If the model can only focus on very few time points, it may miss some important time points, leading to performance degradation.
> + As shown in **Figure 6 in Appendix D**, only sampling 6 points as important time points can achieve ideal performance, which prove that it's unnecessary to focus on all time points. Thus focusing on less time points is a good objective. But in some datasets, sampling 24 points as important time points performs better than only sampling 6 points, which proves that too few focused points is not appropriated.
>
> >Reference
>
> [1] PatchTST
>
> [2] Triformer
>
> [3] Informer
>
> [4] FiLM

---

> > ### Author Response · Authors · 2024-08-10
> > **Detailed Version of Reference and More Discussion about Q1**
> >
> > Dear Reviewer reZp,
> >
> > Due to the page limitation, we provide the detailed version of Reference here.
> >
> > And we also provide a copy of our response to Q1 here (which is originally provided in the **Global Response**) and further provide more discussion about Q1. We hope it can better help to address your concern.
> >
> > > Detailed Reference for this Specific Response
> >
> > [1] PatchTST: Yuqi Nie, et al. "A Time Series is Worth 64 Words: Long-term Forecasting with Transformers."
> >
> > [2] Triformer: Razvan-Gabriel Cirstea, et al. "Triformer: Triangular, Variable-Specific Attentions for Long Sequence Multivariate Time Series Forecasting."
> >
> > [3] Informer: Haoyi Zhou, et al. "Informer: Beyond Efficient Transformer for Long Sequence Time-Series Forecasting."
> >
> > [4] FiLM: Tian Zhou, et al. "FiLM: Frequency improved Legendre Memory Model for Long-term Time Series Forecasting."
> >
> > > A Copy of Our Response to Q1 in the **Global Response**
> >
> > >> For line 34-35
> > + About claims on long input length:
> >     + [1][2][3] find that using input lengths longer than 96 can provide ideal performance. But they do not compare their performance under short input lengths like 96.
> >     + [4] compares these models under input-96 settings and observes significant decrease in their performance, indicating these patch-based Transformer forecasters need a long input to achieve ideal performance.
> > + About claims on large patch size:
> >     + [1][2] study the impact of large patch size on long-term forecasting performance and find that performance will only improve **when a large patch size (at least more than 8)** is used.
> >
> > >> Do you consider patch sizes of 8 and 16 to be large?
> > + Considering the diversity of input lengths in real-world time series, patch sizes of 8 and more than 8 can be considered to be large. In a wider range of time series tasks besides long-term forecasting, the series length may be less than 10 (which mainly happens on short-term tasks with yearly or quarterly sampling frequency). In such condition, patch sizes of 8 and 16 are very large.
> > + In long-term forecasting, based on the findings in [1][2] that the performance will only improve when a large patch size (**at least more than 8**) is used, we use 8 as a cutoff for the patch sizes.
> > + The word "large patch size" refers not only to PatchTST, but also to the trend of increasing patch sizes in subsequent works (e.g., a very large patch size of 32 in [5]).
> >
> > >> For line 38-39
> > + Yes, this statement refers to patch-based Transformers.
> > + The works that support this claim: [3] conducts experiment on M4 but underperforms the classic baselines like N-BEATs. [6] also conducts experiment on M4 but its performance is not as compelling as it is in long-term tasks.
> >
> > >> Detailed Reference for Global Response
> >
> > [1] PatchTST: Yuqi Nie, et al. "A Time Series is Worth 64 Words: Long-term Forecasting with Transformers."
> >
> > [2] Crossformer: Yunhao Zhang, et al. "Crossformer: Transformer Utilizing Cross-Dimension Dependency for Multivariate Time Series Forecasting."
> >
> > [3] GPT4TS: Tian Zhou, et al. "One Fits All: Power General Time Series Analysis by Pretrained LM."
> >
> > [4] iTransformer: Yong Liu, et al. "iTransformer: Inverted Transformers Are Effective for Time Series Forecasting."
> >
> > [5] Pathformer: Peng Chen, et al. "Pathformer: Multi-scale Transformers with Adaptive Pathways for Time Series Forecasting."
> >
> > [6] CARD: Xue Wang, et al. "CARD: Channel Aligned Robust Blend Transformer for Time Series Forecasting."
> >
> > > More Discussion about Q1
> >
> > In addition to above papers we provided in **Global Response**, our experimental results further support our claims:
> > + For the claims on input length in **line 34-35**, we conduct unified experiments to evaluate models' performance under various input lengths. As shown in **Figure 4 (right) in the main text**, the patch-based Transformers (e.g., PatchTST and CARD) only outperform the non-patch ones (e.g., iTransformer) **when the input is longer than 384**, indicating these patch-based Transformer forecasters need a long input to achieve ideal performance.
> > + For the claims on patching in **line 34-35**, we newly conduct more experiments. As shown in **Figure 2 in PDF of global response**, if without patching, the performance of PatchTST and CARD will decrease obviously and fell out of the good rankings, indicating that previous patch-based Transformer forecasters indeed need to work with patching.
> > + For the claims on short-term forecasting performance in **line 38-39**, we conduct experiments on a wider range of short-term forecasting tasks to thoroughly evaluate previous patch-based Transformer models. As shown in **Section 4.2 of the main text**, we find that they fail in many cases of short-term forecasting.
> >
> > **These findings from our experiments are consistent with the above papers, which further support our claims**.
> >
> > Thanks again for your valuable suggestions. We will cite the above papers in the corresponding places to support our claims.
> >
> > Sincerely,
> >
> > Authors

---

> > > ### Comment · Reviewer_reZp · 2024-08-11
> > >
> > > Thank you for the response and the additional experiments. As my concerns have been addressed, I raised the score of the paper.

---

> > > > ### Author Response · Authors · 2024-08-11
> > > > **Thanks for Your Response and Raising the Score**
> > > >
> > > > We would like to thank Reviewer reZp again for providing the insightful review and valuable suggestions,
> > > > which enable us to make an effective response and help us a lot to improve our paper.
> > > >
> > > > And we would also like to thank you for raising the score and recommending our paper!

---

### Official Review · Reviewer_fiTw · 2024-07-12

**Soundness:** 2
**Presentation:** 3
**Contribution:** 3
**Rating:** 5
**Confidence:** 4

**Summary:**

The paper introduces a new transformer architecture for time series forecasting that does not necessarily depend on patching, resulting in consistent performance improvements over all baselines. Although the approach primarily extends existing work, its simplicity and potential for widespread adoption in various time series forecasting applications is notable, provided the experimental results are robust and well-justified.

**Strengths:**

1. **Clean Methodology**: The methodology is elegantly simple and eliminates the need for manual hyperparameter tuning required by patch-based methods. This adaptability to various sequence scales with minimal complexity is a significant advantage. Additionally, as demonstrated in Section D, the method exhibits low sensitivity to parameter variations, which underscores its potential for broad applicability.
2. **Clear Presentation of Proposed Method**: The authors effectively identify the core problem and critically review prior works, highlighting limitations stemming from their dependence on patching. The presentation of the methodological framework is clear and detailed.
3. **Comprehensive Experiments**: The experimental setup is thorough, employing well-known datasets relevant to the task. The choice of baselines is comprehensive, including both patch-based and non-patch-based transformer models. The experiments cover a range of scenarios, including different dataset temporal scales and analyses of time complexity and parameter sensitivity.

**Weaknesses:**

1. **Uniform Attention Concerns**: The proposed deformableTST model appears to use a uniform attention prior, adjustable via learnable offsets. Despite this flexibility, the effectiveness in scenarios where key information is clustered within specific time window remains uncertain. The paper asserts that this method can adeptly manage both uniform and clustered attention distributions, yet it lacks systematic experimental validation of this claim. Incorporating a synthetic dataset designed to simulate distinct scenarios could substantiate these assertions more convincingly. Example test cases include:
    1. Future data evenly relates to historical data, anticipating good performance from methods like PatchTST, which use a uniform attention prior.
    2. Future data is closely related only to a specific historical window $[t_0-a_i, t_0-a_i+\Delta t]$, with constant $\Delta t$ and $a_i$ for each sample.
    3. Similar to the second scenario, but with $a_i$ varying across samples.
    4. Possible other typical cases …

    Can the proposed method consistently lead in performance across these scenarios? Do the Effective Receptive Fields (ERFs) operate as anticipated for each case? I believe it is crucial that the results rigorously validate these aspects to strengthen the paper’s claims and ensure its conclusions are compelling and beyond reproach.

2. **Differentiation from previous Work**: The paper seems a application of 2D deformable attention [1] in vision transformer to 1D sequence forecasting. A more detailed comparison with [1] from the method design view could enhance the paper's contribution to the field.
3. **Implementation Details**: The authors' promise to release the source code is appreciated. While early access would be beneficial for thorough validation and further exploration of the method's promising capabilities, considering the consistently leading performance and simple design.

[1] Vision Transformer with Deformable Attention

[2] Deformable Convolutional Networks

**Questions:**

1. Does the method in this paper use a similar restriction as in [1], where $∆p ←− s \tanh (∆p)$ limits the range of attention offset, or does it allow offsets across the entire sequence?
2. Is the important time point sampling detailed in Algorithm 1 (Sec.3.2, Eq. 6-8)? It seems crucial to the method's design but is not explicitly mentioned in the algorithm description.
3. In I.1, the method considers only the two closest time points for output. Could this focus on local optima restrict the learning potential of the Deformable Attention module, especially if other time points might be more relevant?
4. In lines 710-712, the authors note that PatchTST cannot effectively learn centralized attention in localized areas. Could the authors provide illustrations of Effective Receptive Fields (ERFs) similar to those shown in Figure 8, but demonstrating how the proposed method manages such scenarios? This addition would be highly beneficial as the existing illustrations primarily depict the method's ability to distribute attention across entire series. A comparison showcasing the method's capability to focus attention locally would address a critical aspect of time series forecasting where concentrated information is crucial.
5. typo in line 714: focus

I am prepared to increase my score if all my concerns are adequately addressed.

**Limitations:**

Yes, the authors have mentioned the limitations and potential negative societal impacts. They do an excellent job empirically demonstrating performance improvements. The method clearly shows an advantage over prior works across all benchmarks. To enhance the paper, a deeper analysis explaining why the proposed method improves performance would be beneficial and provide valuable insights into its effectiveness.

---

> ### Author Rebuttal · Authors · 2024-08-07
>
> Many thanks to Reviewer fiTw for providing a detailed review and insightful questions.
> > W1 & Q4: Experiments on synthetic dataset to prove our model can handle both uniform and clustered attention distribution, especially the clustered ones (centralized attention in localized areas).
>
> + Thanks for the valuable suggestion, we conduct experiments with some typical cases of attention distributions.
> The details and results are provided in **Figure 1 of PDF in global response**.
>
> + Our method can accurately predict the future data in all cases. And ERFs can operate as anticipated, successfully matching the distributions of key information. In details, in the case of globally uniform attention, the brighter points in ERF are also distributed globally, which means the model can find the important time points across the whole series. In other cases, the brighter points in ERF tend to concentrate in localized areas of key information, **proving the effectiveness of our method in scenarios where key information is clustered within specific time window**. These results validate that our method can adeptly manage both uniform and clustered attention distributions.
>
> > W2: Differentiation from previous work.
>
> We'd like to highlight our difference from previous works as follows:
> + **Different model designs and implementations**:
> [1] is a computer vision method and focuses more on the locality of image. Therefore it uses the combination of three attentions (local attention, shift-window attention and deformable attention), while our method only use pure deformable attention.
> Meanwhile, [1] limits the range of attention offset to further enhance locality, while we do not use this restriction.
> + **Using similar idea for totally different purpose**:
>     + Our method is inspired by the deformable operations in CV like [1], but is proposed for different purpose.
> [1] adopts deformable attention to construct an efficient and flexible backbone for vision tasks.
>     + But in our paper, we use deformable attention to solve the problem of over-reliance on patching, which is a significant problem in time series domain.
> Based on our exploration, we find out the reason behind this problem is that previous attentions have poor ability to focus on the important time points and thus need to rely on the guidance of patching.
> Considering deformable attention enjoys better focusing ability by itself, we adopt it to solve this significant problem, **which is specific to our analysis**.
>     + Therefore, **although all adopting deformable attentions, we use it for totally different purpose and solve a completely different problem.** We hope this can clarify our difference from [1].
> + We'd also like to emphasize that the problem of over-reliance on patching is of great significance to the time series community.
> Our proposed method is the first to deeply explore and specially target to this less-explored research problem, which is of great significance and novelty. And our findings can prompt people to rethink the relationship between attention and patching, and thereby design more powerful Transformer-based forecasters with wider applicability. Therefore, **in addition to proposing a state-of-the-art method for time series forecasting, the exposure and exploration of this significant problem is also a main contribution of our paper.**
>
> > W3: Implementation details & code release.
>
> + As claimed in our **Reproducibility Statement**, we provide implementation details, model settings and pseudo-code in **Appendix A,B,C**. Details about tensor shape and model structure are also included in **Section 3**.
> + **As per our tradition, we guarantee to make the code public upon paper acceptance.**
>
> > Q1: Do you use a restriction to limit the range of offset?
>
> + No, we don't use any restrictions. Our strategy of not using restriction can **better manage both uniform and clustered attention**, which is a main concern of the reviewer.
> + In terms of clustered attention, despite our reference points are initially uniformly distributed through the whole series, our strategy makes it possible for all of them to converge towards the same localized area because it allows offsets across the entire series.
> + In terms of uniform attention, as shown in **Figure 8 of Appendix**, since our reference points are initially uniformly distributed through the whole series, our deformable attention can finally find the important time points in a global range.
> And Figure 8 also shows that, in case of need, our module can spontaneously learn a smaller offset value without additional restrictions, which verifies the soundness of our strategy that uses no restriction.
>
> > Q2: Description of important time point sampling is not in Algorithm 1.
> + Thanks for your concern. Algorithm 1 mainly introduces the overall structure of our models, not the sampling process.
> + For the details of sampling process, we describe it step by step in **Section 3.2 and Appendix I** through detailed text descriptions,
> including how to obtain references, calculate offsets and finish the sampling with linear interpolation.
> And we also provide **Figure 3 in main text** as a clear visual description.
>
> > Q3: Only use 2 closet points in I.1.
>
> + We sample each important time point based on linear interpolation. In linear interpolation, we can assume that the output are more relevant to its two closest points, which brings better efficiency and simplicity.
> + This assumption is also validated in 2D cases. The deformable methods in CV also accomplish their bi-linear interpolation only with the closest points [1][2].
> + In real-implementation, Deformable Attention module will sample 12 important time points. Since the references for the sampling process are distributed across the whole series, **it can avoid the focus on local optima and ensure the learning potential.**
>
> > Q5: typo
>
> Thanks for your reminder. We have fixed it.
>
> > The reference is the same as in the Official Review.

---

> > ### Comment · Reviewer_fiTw · 2024-08-10
> >
> > Thank you, authors, for taking the time to response directly to the major considerations I raised in my review. I raised my score.

---

> > > ### Author Response · Authors · 2024-08-10
> > > **Thanks for Your Response and Raising the Score**
> > >
> > > We would like to thank Reviewer fiTw again for providing a detailed valuable pre-rebuttal review.
> > > Your detailed suggestions help us a lot in the rebuttal and paper revision!
> > > And we guarantee to make the code public upon paper acceptance.
> > >
> > > And we would also like to thank you for raising the score and recommending our paper!
> > >
> > > If you have any further questions or concerns, please feel free to let us know.

---

### Official Review · Reviewer_eY9h · 2024-07-12

**Soundness:** 3
**Presentation:** 3
**Contribution:** 2
**Rating:** 6
**Confidence:** 4

**Summary:**

This paper presents a Time Series Forecasting method DeformableTST that makes patching replaceable in transformer with sparse attention, using a hierarchical architecture to avoid memory issues. Experiments are performed on 8 data sets and the proposed method is compared with variety of SOTA methods.

**Strengths:**

1. Making patching replaceable with sparse attention
2. The use of hierarchical architecture to overcome memory issues due to removal of patching
3. The use of small patch for very long time series
4. Comparisons with Variety of SOTA methods on 8 data sets
5. Ablation Study

**Weaknesses:**

1. Related works does not include Graph-Transformer methods e.g. STGNN
2. Graph-Transformer methods e.g. STGNN are not used as SOTA for comparisons.
3. Data Sets should include financial data e.g. Stock Market
4. Synthetic data should have been used to really test the proposed approach and claims.
5. Table 1 is not properly discussed, specifically, why CARD  and RLinear are better on some data sets
6. Table 2 should be discussed the same way for PEMS

I appreciate the description of model efficiency in terms of computational cost, but I believe the contribution is incremental and needs more work and the use of synthetic data to improve the impact.

**Questions:**

1. Why SOTA methods do not include Graph-Transformer methods e.g. STGNN?
2. Will DeformableTST work on stock market data?

**Limitations:**

None Listed

---

> ### Author Rebuttal · Authors · 2024-08-07
>
> Many thanks to Reviewer eY9h for the thorough and detailed comments.
>
> > Concern about contributions.
>
> We'd like to highlight our contributions in the following points:
> + Our method **differs from** previous Transformer-based forecasters for its **better applicability and less reliance on patching**.
> Different from previous works, our DeformableTST can successfully reduce the reliance on patching and broaden the applicability of Transformer-based models, **which is a great improvement than previous Transformer-based models**.
>
> + We propose a **novel methods for time series forecasting with better performance and wider applicability**. Our DeformableTST can flexibly adapt to multiple input lengths and achieve excellent performance in tasks unsuitable for patching. Therefore, our DeformableTST achieves consistent state-of-the-art performance in a broader range of time series tasks, showing great practical values in a wider range of real-world applications.
>
> + In addition to proposing a state-of-the-art method for time series forecasting, **the exposure and exploration of the novel problem of over-reliance on patching is also a main contribution of our paper, which is not incremental.**
>     + The problem of over-reliance on patching is of great significance to the time series community.
>     And our proposed methods is the first to deeply explore and specially target to this less-explored research problem,
>     **which highlights our novelty and contribution**.
>     + And our findings can bring new perspective to time series community and prompt people to rethink the relationship between attention and patching,
>     which benefits the future work to design more powerful Transformer-based forecasters with wider applicability.
>
> > W1 & W2: Including Graph-Transformer methods in related work and comparison.
> + Following your valuable suggestion, we add the latest Graph-Transformer Sageformer[1] as our baseline.
> The results are shown in **Table 1 of global response**. Our DeformableTST achieves consistently better performance than the latest Graph-Transformer method, further demonstrating our performance superiority.
> + Applying Sageformer on short term tasks leads to NaN output. We are working diligently to fix it. We will include the complete experimental results in the final version and include Sageformer in our related work.
>
> [1] SageFormer: Series-Aware Framework for Long-Term Multivariate Time-Series Forecasting.
> > Q1: Why SOTA do not include Graph-Transformer methods.
> + Since the latest state-of-the-art models in time series forecasting are mainly Transformer-based and Linear-based models,
> mainly using these two types of models as strong baselines can enhance the persuasiveness of experiments.
> And this is a protocal widely adopted by many published papers from premier conferences.
> **To provide a fair comparision, we follow this mainstream protocol**.
> + Thanks again for suggestions in improving our experiments. We newly add Sageformer as SOTA baseline to further enhance our persuasiveness.
>
> > W3: Data Sets should include financial data.
>
> Thanks for your concern, in addition to the 8 datasets mentioned by you, we have already conducted experiments on 32 datasets to ensure adequacy of our experiments, which adequately cover a wide range of real-world scenarios and **have already included the financial data (e.g., M3, M4 and Exchange)**.
> > **Q2:** Will DeformableTST work on stock market data?
> + Following your valuable suggestion, we further conduct experiments on Kaggle Stock Market dataset. As shown in **Table 2 of global response**, our DeformableTST still outperforms other competitors, validating that DeformableTST can work on stock market data.
>
> > W4 : Experiments on synthetic data.
> + Following your valuable suggestion, we conduct experiments on synthetic data and show the results in **Figure 1 of PDF in global response**.
> Please refer to the **Response to W1 & Q4 of Reviewer fiTw** for detailed analysis.
> + The results shows that our method works well for each specific scenario, which further verify the proposed approach and our claims.
>
> > W5 & W6 : Discussion about Table 1 & 2
> + We'd like to claim the adequacy of our discussion as follows:
>     + Following the guideline of paper checklist, our experimental discussions are mainly **served to analyse our method and validate our claims**.
> Therefore, the discussion in our paper mainly focus on our own model. And other baselines are comprehensively discussed by category (but not individually) to ensuring the persuasiveness.
>     + In Table 1, RLinear and CARD just achieve **similar performance** with us in **one of the datasets**.
> And **our model surpasses them by a larger margin in a wider range of tasks, gaining SOTA in most cases**.
> Especially considering that these two models are the most powerful Transformer- and linear-based models,
> the results in Table 1 convincingly verify our performance superiority.
>     + In Table 2, we emphasize PEMS as an example to illustrate our limitation of not considering multivariate correlation.
> For other datasets, we discuss them by category to demonstrate the model's ability to generalize across multiple scenarios.
> By categorizing them from different input lengths and different task difficulty,
> the discussion of Table 2 proves our claim that our method can reduce the reliance on patching and broaden the applicability
>
> > Concern about limitations.
> + Thanks for your concern, we have already addressed the limitations in **Section 4.2 (line 262-265)** and also listed it in **Appendix K**, which are:
>     + Our paper mainly focuses on how to better use attention in temporal modeling. It will be our future work to study how to further capture the multivariate correlation in our model.
>     + Our paper mainly focuses on time series forecasting tasks. We will further explore its potential in more time series analysis tasks and further develop its performance by large-scale pre-training in the future.

---

> > ### Comment · Reviewer_eY9h · 2024-08-08
> >
> > Thanks for your response, after reading your response and other reviews, I am updating my score.

---

> > > ### Author Response · Authors · 2024-08-09
> > > **Thanks for Your Response and Raising the Score**
> > >
> > > We would like to thank Reviewer eY9h again for providing the valuable review and insightful suggestions.
> > > Your constructive suggestions are very helpful for us to improve the paper into a better shape.
> > >
> > > And we would also like to thank you for raising the score and recommending our paper!

---

### Author Rebuttal · Authors · 2024-08-07

> **Global Response**

We sincerely thank all the reviewers for their insightful reviews and valuable comments, which are instructive for us to improve our paper further.

In this paper, we expose the significant problem of over-reliance on patching in latest Transformer-based forecasters.
Based on our exploration, we find out the reason behind is that previous attentions have poor ability to focus on the important time points and thus need to rely on the guidance of patching.
To solve this problem, we propose DeformableTST, which equipped with deformable attention that enjoys better focusing ability by itself.
Experimentally, our DeformableTST achieves the consistent state-of-the-art performance in a wider range of time series tasks, successfully reducing the reliance on patching and broadening the applicability of Transformer-based models.

The reviewers generally hold positive opinions of our paper,
in that **"we effectively identify the core problem"**, our problem definition is **"reasonable"**, our method is **"novel", "advanced", "elegantly simple", "of good motivation"** and "**"clearly shows an advantage over prior works across all benchmarks"**, our paper is **"well-written, easy to read and understand"**, our experiments are **"comprehensive and thorough"** and we **"do an excellent empirical job"**.

The reviewers also raised insightful and constructive concerns. We made every effort to address all the concerns by providing sufficient evidence and requested results. Here is the summary of the major revisions:
+ **Experiments on synthetic dataset (Reviewer fiTw, eY9h)**: We use synthetic dataset to simulate typical scenarios and conduct systematic experiments to prove our method can adeptly manage both uniform and clustered attention distributions.
+ **More baselines and datasets (Reviewer eY9h)**: By comparing our method and the latest graph-Transformer Sageformer and conducting new experiments on Stock Market, we further demonstrate our consistent performance superiority.
+ **More ablations (Reviewer reZp)**: We provide ablations to prove our robustness to patching.
+ **Adjustments and justification for some claims (Reviewer reZp)**: We cite related works, explain our results and provide further experiments to support our claims. And we aslo provide explanations on some concepts to make them clearer.
+ **Description on details of our methods (Reviewer fiTw)**: We illustrate the details of the sampling process and explains the soundness of our designs.
+ **Novelty and Contribution (Reviewer eY9h, Qoam)**: We highlight our difference with previous works in time series domains. Our method differs from previous Transformer-based forecasters **for its better applicability and less reliance on patching**. And we highlight **our contribution on exposing, exploring and solving the significant problem of over-reliance on patching**.
+ **Difference from CV methods (Reviewer fiTw)**: Our method differs from previous methods in CV for the different model designs and different purpose, which is specific to our analysis.

The valuable suggestions from reviewers are very helpful for us to revise our paper to a better shape. We'd be very happy to answer any further questions.

> **Tables and Figures**

All Tables are listed as follows. And Figures are provided in PDF.

> Table 1: Comparison of DeformableTST and Sageformer in long-term forecasting.

|Dataset MSE/MAE|ETTh1|ETTh2|ETTm1|ETTm2|Weather|Solar|ECL|Traffic|
|:---:|:---:|:---:|:---:|:---:|:---:|:---:|:---:|:---:|
|Ours|0.413/0.430|0.336/0.381|0.358/0.386|0.267/0.321|0.233/0.266|0.199/0.255|0.169/0.267|0.410/0.280|
|Sageformer|0.427/0.438|0.368/0.405|0.371/0.394|0.275/0.327|0.238/0.272|0.227/0.285|0.174/0.273|0.418/0.287|


> Table 2: Multi-variate short-term forecasting on Stock Market.

|Models|Ours|Path.|CARD|GPT4TS|PatchTST|iTrans.|Auto.|FED.|RLinear|TiDE|TimesNet|
|:---:|:---:|:---:|:---:|:---:|:---:|:---:|:---:|:---:|:---:|:---:|:---:|
|MSE|0.122|0.134|0.133|0.140|0.137|0.134|0.144|0.140|0.163|0.160|0.133|
|MAE|0.151|0.159|0.159|0.162|0.171|0.159|0.195|0.184|0.188|0.186|0.158|

> **For Q1 of Reviewer reZp**

> For line 38-39
+ About claims on long input length:
    + [1][2][3] find that using input lengths longer than 96 can provide ideal performance. But they do not compare their performance under short input lengths like 96.
    + [4] compares these models under input-96 settings and observes significant decrease in their performance, indicating these patch-based Transformer forecasters need a long input to achieve ideal performance.
+ About claims on large patch size:
    + [1][2] study the impact of large patch size on long-term forecasting performance and find that performance will only improve **when a large patch size (at least more than 8)** is used.

> Do you consider patch sizes of 8 and 16 to be large?
+ Considering the diversity of input lengths in real-world time series, patch sizes of 8 and more than 8 can be considered to be large.
In a wider range of time series tasks besides long-term forecasting,
the series length may be less than 10 (which mainly happens on short-term tasks with yearly or quarterly sampling frequency).
In such condition, patch sizes of 8 and 16 are very large.
+ In long-term forecasting, based on the findings in [1][2] that the performance will only improve when a large patch size (**at least more than 8**) is used,
we use 8 as a cutoff for the patch sizes.
+ The word "large patch size" refers not only to PatchTST, but also to the trend of increasing patch sizes in subsequent works (e.g., a very large patch size of 32 in [5]).

> For line 38-39
+ Yes, this statement refers to patch-based Transformers.
+ [3] conducts experiment on M4 but underperforms the classic baselines like N-BEATs. [6] also conducts experiment on M4 but its performance is not as compelling as it is in long-term tasks.

[1]PatchTST

[2]Crossformer

[3]GPT4TS

[4]iTransformer

[5]Pathformer

[6]CARD

---

### Decision · Program_Chairs · 2024-09-25

**Decision:**

Accept (poster)

**Comment:**

The reviewers unanimously voted for acceptance. They appreciated the technical originality, clarity and extensive experiments that support the efficacy of the proposed approach. After carefully reading the reviews and the rebuttal, this meta-reviewer concurs with the reviewers' recommendations.